# Scabertopin Derived from *Elephantopus scaber* L. Mediates Necroptosis by Inducing Reactive Oxygen Species Production in Bladder Cancer In Vitro

**DOI:** 10.3390/cancers14235976

**Published:** 2022-12-02

**Authors:** Yuanhui Gao, Zhenyu Nie, Hui Cao, Denggao Huang, Mei Chen, Yang Xiang, Xiaolong Yu, Shufang Zhang

**Affiliations:** 1Central Laboratory, Affiliated Haikou Hospital of Xiangya Medical College, Central South University, Haikou 570208, China; 2School of Materials Science and Engineering, Hainan University, Haikou 570228, China

**Keywords:** scabertopin, bladder cancer, reactive oxygen species, necroptosis, *Elephantopus scaber* L.

## Abstract

**Simple Summary:**

Scabertopin is one of the major sesquiterpene lactones from *Elephantopus scaber* L. Sesquiterpene lactones are thought to have fairly strong anti-cancer efficacy. However, there has not been any research report on the efficacy and mechanism of Scabertopin in the treatment of bladder cancer. The aim of this study is to evaluate the antitumor activity of scabertopin in bladder cancer and its potential molecular mechanism in vitro. In this study, we found that scabertopin can induce necroptosis in bladder cancer cells by promoting the production of mitochondrial reactive oxygen species, and also inhibit the migration and invasion ability of bladder cancer cells. At the same time, we also demonstrated that the half-inhibition rate of scabertopin on various bladder cancer cell lines is much lower than that on human ureteral epithelial immortalized cells. This shows that scabertopin is a safe and effective anti-bladder cancer drug.

**Abstract:**

Bladder cancer remains one of the most common malignant tumors that threatens human health worldwide. It imposes a heavy burden on patients and society due to the high medical costs associated with its easy metastasis and recurrence. Although several treatment options for bladder cancer are available, their clinical efficacy remains unsatisfactory. Therefore, actively exploring new drugs and their mechanisms of action for the clinical treatment of bladder cancer is very important. Scabertopin is one of the major sesquiterpene lactones found in *Elephantopus scaber* L. Sesquiterpene lactones are thought to have fairly strong anti-cancer efficacy. However, the anticancer effect of sesquiterpenoid scabertopin on bladder cancer and its mechanism are still unclear. The aim of this study is to evaluate the antitumor activity of scabertopin in bladder cancer and its potential molecular mechanism in vitro. Our results suggest that scabertopin can induce RIP1/RIP3-dependent necroptosis in bladder cancer cells by promoting the production of mitochondrial reactive oxygen species (ROS), inhibit the expression of MMP-9 by inhibiting the FAK/PI3K/Akt signaling pathway, and ultimately inhibit the migration and invasion ability of bladder cancer cells. At the same time, we also demonstrated that the half-inhibition concentration (IC50) of scabertopin on various bladder cancer cell lines (J82, T24, RT4 and 5637) is much lower than that on human ureteral epithelial immortalized cells (SV-HUC-1). The above observations indicate that scabertopin is a potential therapeutic agent for bladder cancer that acts by inducing necroptosis and inhibiting metastasis.

## 1. Introduction

Bladder cancer is one of the most common malignant tumors of the urinary system. The cancer statistics released in 2022 state that, in the United States, bladder cancer has an estimated incidence of 81,800 cases and a mortality rate of 17,100 cases [1]. These rates are slightly lower than those reported in 2021 [2]. In China, bladder cancer had an incidence of approximately 80,500 and a mortality rate of 32,900 in 2015 [3]. Uroepithelial cancer is the main pathological type of bladder cancer. Approximately 75% and 25% of the cases of this type of cancer are non-muscle-invasive bladder cancer (NMIBC) and muscle-invasive bladder cancer (MIBC), respectively. The current clinical treatment of bladder cancer remains based on surgery combined with postoperative radiotherapy [4,5]. Mitomycin C, epirubicin, adriamycin, methotrexate, vincristine, pirarubicin, and cisplatin are used as the first-line chemotherapeutic agents for bladder cancer in clinical practice [6]. However, the use of these drugs is greatly limited by their unsatisfactory efficacy, the widely reported drug tolerance of bladder cancer, and their adverse effects. Although the rapid development of molecular biology and genetics research has provided new therapeutic tools for the clinical development of immunotherapy, gene therapy, and targeted therapy, these approaches fail to exert an ideal treatment effect on bladder cancer because they use a large number of chemically synthesized drugs. This approach results in high treatment costs, drug side effects, hepatotoxicity, and drug resistance. Therefore, the search for new drugs with low toxicity and side effects has been a hot issue in tumor treatment.

In recent years, many natural compound products have played an important role in the treatment of tumors. Finding compounds with significant tumor-inhibiting activity from natural products is an important approach and means for the research and development of anticancer drugs [7]. Scabertopin is a sesquiterpene lactone compound that is mainly derived from the chemical constituents of *Elephantopus scaber* L. It is a kind of herbaceous plant belonging to the phylum Angiosperm, class Dicotyledonous, order Campanulaceae, family Asteraceae, and genus Elephantopus [8], which is widely distributed worldwide, particularly in East Asia, Southeast Asia, Africa, Australia, India, and South America, and has been reported to have various pharmacological properties, such as anti-bacterial, anti-diabetic, anti-inflammatory, and anti-cancer efficacy [9,10]. An increasing number of studies have shown that the antitumor activity of *Elephantopus scaber* L. is attributed to sesquiterpene lactones, which execute significant antitumor activities in nasopharyngeal [11,12], cervical [13], breast [14], colon [15,16], and hepatocellular carcinomas [17]. For example, in HCT116 and RKO cells, deoxyelephantopin (DET), another sesquiterpene lactone compound derived from *Elephantopus scaber* L. can significantly increase reactive oxygen species (ROS) levels, thereby activating the JNK signaling pathway and triggering cell death. In the mouse breast cancer cell line TS/A, DET can inhibit TNF-α-induced NF-κB activity and down-regulate the NF-κB regulated gene products of matrix metalloproteases (MMP)-2 and MMP-9, thereby inhibiting the migration and invasion in vitro and in vivo [14]. In addition, sesquiterpenoids exhibit different cytotoxicities toward tumor cells and normal cells, whereby they selectively target tumor cells and show little systemic toxicity [18,19]. However, the research on the anticancer effect and mechanism of scabertopin in bladder cancer has not been reported.

Necroptosis is a unique form of programmed cell death, which is characterized by caspase-independent activation and is morphologically accompanied by the swelling of organelles and the rupture of cell membranes, resulting in the spillage of the cellular contents into the surrounding cells and, consequently, triggering inflammatory response [20,21]. Various drugs and active sites of traditional Chinese medicine have been confirmed to exert their anticancer activity through necroptosis [22,23]. Receptor-interacting protein (RIP)1, RIP3, and mixed lineage kinase-like (MLKL) are three key factors in the regulation of necroptosis; RIP3 has been well-established to recruit MLKL and induce its phosphorylation. Phosphorylated-MLKL (p-MLKL) then undergoes oligomerization and translocates to the plasma membrane to execute cellular necrosis [24,25,26]. However, the pathway through which RIP1 undergoes autophosphorylation remains controversial. ROS have been recently reported to act as upstream cell signaling molecules to activate and drive necroptosis [27]. RIP1 can sense ROS by modifying three key cysteine residues (C257, C268, and C586) and, subsequently, induces their autophosphorylation at the S161 site, which in turn results in the recruitment of RIP3 to form a necrosome [28]. Previous studies have also shown a correlation between necroptosis and drug-induced ROS. For example, artesunate can inhibit the proliferation of renal cancer cell lines in vitro by inducing elevations in ROS levels and, thus, promoting RIP1-dependent necrotic apoptosis [29]. In addition, emodin promotes the occurrence of necrotic apoptosis in renal cell carcinoma (RCC) cell lines, thereby restricting tumor development by inducing the production of ROS that promote increases in the levels of RIP1 and MLKL phosphorylation [30]. In conclusion, the positive correlation of necroptosis with ROS implies that regulating ROS to induce necroptosis in tumor cells is a potential strategy for cancer therapy.

In this study, we aimed to evaluate the inhibitory effects and potential molecular mechanisms of scabertopin on bladder cancer cells in vitro. Scabertopin is one of the four sesquiterpene lactones with the highest content in *Elephantopus scaber* L. Since DET has shown a good anti-tumor effect in a variety of tumor cell lines, we speculated that scabertopin would also have similar efficacy. Therefore, we determined our research objective was to study the effect of scabertopin on bladder cancer cell lines and explore its way to promote the death of bladder cancer cell lines. Our results showed that scabertopin with antitumor properties inhibited the growth, migration, and invasion of the bladder cancer cell line J82. Further in-depth studies demonstrated that the above effects of scabertopin may be related to the promotion of ROS generation, the induction of focal adhesion kinase (FAK) phosphorylation, and the activation of necroptotic signaling pathways.

## 2. Materials and Methods

### 2.1. Main Reagents

Scabertopin (PS1863-0010) was purchased from Chengdu Push Biotechnology Co., Ltd. (Chengdu, China). DMEM basal medium (10569010) and trypsin-EDTA (25200056) were purchased from Gibco (Grand Island, NY, USA). RPMI 1640 (PM150110) was purchased from Procell Life Science & Technology Co., Ltd (Wuhan, China). Fetal bovine serum (FBS) (1907301) was purchased from Biological Industries (BI) (Kibbutz Beit Haemek, Israel). Cell Counting Kit-8 (CK04) was purchased from Dojindo Laboratories (Kumamoto, Japan). LIVE/DEAD cell viability assay kit (L3224) was purchased from Invitrogen (Carlsbad, CA, USA). The cell cycle assay kit (KGA512) and the apoptosis assay kit (KGA1018) were purchased from KeyGen BioTECH (Shanghai, China). ROS detection kit (S0033S) was purchased from Beyotime Biotechnology (Shanghai, China). B cell lymphoma-2 (Bcl-2, ab182858), Bcl-2-associated X (Bax, ab32503), phospho-PI3K (Y607, ab182651), AKT (ab179463), phospho-AKT (S472, S473, S474, ab192623), and GAPDH (ab181602) antibodies were purchased from Abcam (Cambridge, MA, USA). Gasdermin D (97558), GPX4 (59735), FAK (71433), phospho-FAK (Y397, 8556), MMP-9 (13667), PI3K (4257S), caspase-3 (14220), caspase-8 (9746), caspase-9 (9508), MLKL (14993), phospho-MLKL (S358, 91689), RIP1 (3493), phospho-RIP1 (Ser166, 65746), RIP3 (13526), and phospho-RIP3 (Ser227, 93654) antibodies were purchased from Cell Signaling Technology (Danvers, MA, USA).

### 2.2. Material Preparation

Scabertopin (10 mg) was dissolved in 1 ml of dimethyl sulfoxide DMSO (Beyotime Biotechnology, ST2335-100 mL) and configured into a master batch with a concentration of 40 μM such that the maximum concentration of DMSO in the cell culture medium was 0.143%. In the subsequent experiment, in order to exclude the effect of DMSO on cells, 0.143% DMSO was contained in all negative control group media as control. A total of 10 μL of scabertopin and deoxyelephantopin solution was applied evenly on KBr-pressed plates, evaporated and dehydrated using an infrared lamp, and analyzed using a Fourier-transform infrared (FTIR) spectroscopy and KBr-pressed plates. KBr (P116274-100 g) was purchased from Aladdin Biochemical Technology Co., Ltd. (Shanghai, China).

### 2.3. Infrared Spectroscopy

A Fourier-transform infrared tester (Nexus-670 produced by Nexus, Madison, WI, USA) was used to conduct IR spectrum test on the scabertopin powder: the wavelength range was 400–4000 cm^−1^, the number of scans was 32, and the resolution was 4 cm^−1^. The operation was as follows: grind KBr dried under vacuum for 8 h at 140 °C, take 10 mg and press into tablets to obtain background tablets, and grind another 10 mg of KBr and 1 mg of scabertopin powder in an agate mortar evenly. The IR spectroscopy was performed on the obtained background pieces (only KBr) and the samples (KBr and scabertopin powder).

### 2.4. UV Absorption Assay

The mixed solution of scabertopin and DMEM medium was tested using an UV-Vis spectrophotometer (UV9100B from Beijing LabTech Instrument Co., Ltd., Beijing, China), with a wavelength range of 200–360 nm and a resolution of 2 nm.

If the chemical structure of the substance were stable, its UV maximum absorption wavelength would not change. It could be seen from the UV spectrum that the UV maximum absorption peak of scabertopin is around 214 nm, which is consistent with the UV spectrum of the α,β-unsaturated carbonyl moiety. There was no change at 0, 24, and 48 h, and it could be inferred from the Beer–Lambert law that, when the length of the absorption cell, the light source, and the type of the substance to be tested are the same, the absorbance is strictly proportional to the substance concentration.
(1)$$Abs=\mathrm{lg}\left(\frac{1}{T}\right)=Kcl$$

*Abs*: Absorbance; *T*: transmittance; *K*: molar extinction coefficient, cm^2^·mol; *c*: molar concentration, mol/L; and *l*: the length of the absorption cell, cm.

### 2.5. Cell Lines and Culture

Human bladder cancer cell lines (J82 and T24) and human ureteral epithelial immortalized cells (SV-HUC-1) were purchased from the ATCC cell bank, and 5637 and RT4 cell lines were purchased from the China Centre for Type Culture Collection (Shanghai, China). All cell culture media and FBS (30044333) were obtained from Gibco. J82 cells were cultured in DMEM (10% FBS) medium, T24 and 5637 cells were cultured in RPMI 1640 (10% FBS), RT4 cells were cultured in McCoy’s 5A (Gibco, 16600082) (10% FBS) medium, and SV-HUC-1 cells were cultured in F-12K (Gibco, 21127022) (10% FBS) medium. All cell lines were grown in a cell culture incubator at 37 °C and 5% CO_2_ to maintain cell growth. The cells were revived and passaged two times for subsequent experiments.

### 2.6. CCK-8 Assay

The human bladder cancer cells (J82 and T24) and human ureteral epithelial immortalized cells (SV-HUC-1) were inoculated into 96-well plates (100 μL per well). Subsequently, after being added with different concentrations of drugs, the plates were incubated in an incubator at 37 °C and 5% CO_2_ for 24–48 h. A total of 10 μL of CCK-8 reagent (Dojindo Laboratories, CK04) was added to each well, and incubation was continued in the incubator for 1–4 h. Finally, absorbance was detected at 450 nm by using an enzyme marker (Bio-Rad, Hercules, CA, USA). The experimental data (mean ± standard deviation) were plotted in the form of histograms. Dose–response curves were generated using GraphPad Prism 7 (GraphPad Software, San Diego, CA, USA) software, and the absolute 50% inhibitory concentration (IC50) was determined.

### 2.7. LIVE/DEAD Cell Activity Assay

The J82 cells were inoculated at a density of 1 × 10^4^ cells/mL into 96-well plates and treated with gradient concentrations of scabertopin for 24–48 h. Subsequently, the cells were washed with phosphate buffered saline (PBS) in accordance with the instructions of the LIVE/DEAD cell vitality assay kit (Invitrogen), and each group was incubated for 30 min at room temperature with 5 μL of the supplied 4 mM calcein AM stock solution to the 10 mL ethidium dimers (EthD-1). The cells were observed and photographed with an inverted fluorescence microscope, which showed live cells in green (AM) and dead cells in red (EthD-1).

### 2.8. Scanning and Transmission Electron Microscopy

The human bladder cancer J82 cells were inoculated at a density of 2 × 10^4^ cells/mL on slides attached to a 24-well plate, and then treated with the drug at a concentration of 10 μM. After being cultured in the incubator for 24 h, the cells were taken out and washed with precooled PBS twice. The scabertopin-treated cells were fixed with 4% paraformaldehyde (Beyotime Biotechnology, P0099-500 mL) at room temperature and dehydrated with alcohol. Then, their morphology was observed using a scanning electron microscope (SEM) (Regulus8100, HITACHI, Tokyo, Japan).

The J82 cells were inoculated into 6-well plates and then treated with gradient concentrations of scabertopin for 24 h. The culture medium was discarded. The cells were treated with an electron microscope fixative (Servicebio Technology, Wuhan, China, G1102-100 mL) for 2–4 h at 4 °C, collected, and then centrifuged at low speed until green bean-sized clumps of cells could be seen at the bottom of the tubes. After further wrapping, postfixation, dehydration, permeabilization, embedding, sectioning, and staining, the cells were observed under a transmission electron microscope (TEM) and images were collected for analysis.

### 2.9. Measurement of Reactive Oxygen Species

Intracellular ROS levels were measured using an intracellular ROS assay kit (Solarbio Science & Technology, Beijing, China, CA1420). The J82 cells were treated with gradient concentrations of scabertopin for 24–48 h. In the rescue experiments, the cells were incubated with scabertopin for 2 h with 5 mM of ROS scavenger *N*-acetylcysteine (NAC) (MedChemExpress, Monmouth Junction, NJ, USA, HY-B0215). The cells were washed three times with PBS and treated with 10 µM of dichlorofluorescein diacetate probe for 20–30 min. Subsequently, the cells were washed three times with a serum-free medium, and their fluorescence intensity was measured using a fluorescence microplate reader (BioTek, Winooski, VT, USA, Synergy H1).

In order to determine the type of ROS, the J82 cells were treated with gradient concentrations of scabertopin and then loaded with dihydroethidium (DHE) probe (KeyGEN BioTECH, KGAF019) at a concentration of 25 μM and incubated at 37 °C for 60 min in the darkness. After the incubation, the cells were washed with a fresh culture medium and their fluorescence intensity was measured using a fluorescence microplate reader (BioTek, Winooski, VT, USA, Synergy H1). The cells were then imaged under an inverted fluorescence microscope.

### 2.10. Mitochondrial Membrane Potential Assay

The effect of scabertopin on mitochondrial membrane potential (ΔΨ) was detected using the Mitochondrial membrane potential assay kit JC-1 (Beyotime Biotechnology, C2006). The cells were treated with gradient concentrations of scabertopin, washed with PBS, added to a cell culture medium, mixed with the JC-1 solution, and incubated at 37 °C for 20 min according to the manufacturer’s instructions. The fluorescence intensity of the cells was measured using a fluorescence microplate reader (BioTek, Winooski, VT, USA, Synergy H1), and the cells were imaged under an inverted fluorescence microscope. Whereas green fluorescence is an indicator of depolarized mitochondria, intact mitochondria produce red fluorescence.

When ΔΨ is at higher levels, JC-1 aggregates in the matrix of mitochondria to form polymers and produce red fluorescence. When ΔΨ is at a lower level, JC-1 cannot aggregate in the mitochondrial matrix. At this time, JC-1 is monomer and can produce green fluorescence. The ratio of monomer/polymer represents the ratio of green fluorescence/red fluorescence, which can be used to measure the proportion of mitochondrial depolarization

### 2.11. GSH Assay

The GSH (reduced glutathione) and GSSG (oxidized glutathione disulfide) assay kit (Beyotime Biotechnology, S0053) is a simple and easy-to-use assay that can detect the contents of GSH and GSSG, respectively. Briefly, total GSH and GSSG levels were measured at a wavelength of 412 nm after the cells were treated with gradient concentrations of scabertopin. The level of reduced GSH was calculated according to the following formula: (2)GSH=total glutathione (GSH+GSSG)−2×GSSG

### 2.12. Wound Healing Assay

A total of 70 μL of J82 cells were inoculated at a density of 8 × 10^5^ cells/mL into each insert of a Culture-Insert 2 Well (ibidi, Grafelfing, Germany, 80206) in the middle of a dish. After the cells were attached, the insert was removed with forceps and the old medium was aspirated off. The cells were washed gently 1–2 times with PBS, treated with different concentrations of the drugs, placed in the incubator for further incubation, and removed at 0 and 24 h for fluorescent inverted microscopy (IX71, Olympus, Tokyo, Japan) to observe whether the peripheral cells had migrated to the central scratch area. The cells were photographed and recorded. The percentage of wound healing was analyzed using ImageJ software and calculated as the ratio of the initial scratch area minus the partially healed area that had healed at a certain time to the initial area, according to the following formula.
(3)$$initial\;area-\frac{area\;at\;a\;certain\;time\;point}{initial\;area}$$

### 2.13. Transwell Assay

The J82 cells treated with different concentrations of drugs for 24 h were collected and their density was adjusted to 5 × 10^4^ cells/well. The cells were inoculated into the upper chamber of a transwell plate and the lower chamber was supplemented with 10% FBS medium. The plate was placed in the incubator for 24 h, washed twice with PBS, and fixed with methanol. Then, the cells were treated with 0.1% Giemsa staining solution, washed three times with PBS, and allowed to air dry. The number of migrated cells was recorded by photography under multiple high-magnification fields using a microscope (Etaluma, Inc., San Diego, CA, USA, LS720) and by counting the number of migrated cells.

### 2.14. Cell cycle Assays

The J82 cells were treated with different concentrations of drugs for 24 h, digested with tryspin, collected, and washed 1–2 times with PBS. The cells were fixed by adding precooled 70% ethanol and then washed with PBS to remove the fixative. The cells were administered with a RNase/propidium iodide (PI) staining working solution and incubated for 30 min at room temperature under protection from light. The samples were subjected to flow cytometry (FACSCanto, Becton, Dickinson and Company, Franklin Lakes, NJ, USA), and Modfit software was used to analyze the results.

### 2.15. Cell Apoptosis Analysis

The J82 cells were treated with different concentrations of drugs for 24 h, digested by using EDTA-free trypsin, collected, washed and resuspended with PBS, and centrifuged to collect cell precipitates. The precipitates were resuspended again with a small amount of a binding buffer, mixed with an Annexin-V-FITC working solution, incubated for 5 min at room temperature under protection from light, and mixed with a PI reagent and PBS. The samples were analyzed using flow cytometry and the results were analyzed using Modfit software.

### 2.16. Western Blot Analysis

A total of 10 μL of ProteinSafe™ Phosphatase inhibitor cocktail (DI201, TransGen Biotech, Beijing, China) and ProteinSafe™ Protease inhibitor cocktail (DI101, TransGen Biotech, Beijing, China) was added to every 1 ml of ProteinExt^®^ Mammalian total protein extraction kit (TPEB) (DE101, TransGen Biotech, Beijing, China). The TPEB mixed reagents were made by mixing the above three reagents according to the protocols. The cells were mixed with the TPEB mixed reagents on ice after 24 h of scabertopin treatment. The lysates were collected and centrifuged at 14,000× *g* for 10 min at 4 °C. Protein concentrations were analyzed using a Bicinchoninic acid (BCA) kit (P0010, Beyotime Biotechnology). Equal amounts of protein samples were separated through electrophoresis on a 10% precast gel (M00664, GenScript Biotech, Nanjing, China). The proteins were then transferred to PVDF membranes. The membranes were blocked with 5% skimmed milk at 4 °C overnight. Afterward, the membranes were incubated for 2 h at room temperature with different primary antibodies, including Bax (1/10,000), Bcl-2 (1/10,000), caspase-9 (1/1000), caspase-3 (1/1000), caspase-8 (1/1000), GAPDH (1/10,000), MMP-9 (1/500), PI3K (1/1000), p-PI3K (1/1000), AKT (1/1000), p-AKT (1/1000), glutathione peroxidase 4 (GPX4, 1/1000), gasdermin-D (GSDMD, 1/1000), RIP (1/1000), p-RIP (1/1000), MLKL (1/1000), p-MLKL (1/1000), RIP3 (1/1000), and p-RIP3 (1/1000). The membranes were washed three times with TBST and incubated with the secondary antibodies (1/2000–1/20,000) for 1 h at room temperature. The target protein lane was imaged using an iBright 1500 (Invitrogen) with enhanced chemiluminescent substrates (Merck Millipore, Darmstadt, Germany, WBKLS0500). Original blots see Appendix A.

### 2.17. Statistical Analysis

Statistics and bar graphs were analyzed using the Xiantao Academic Online Tools (https://www.xiantao.love/products) (accessed on 30 August 2022), which is based on R and the ggplot2 R package. The means of two groups were considered significantly different if * *p* < 0.05, ** *p* < 0.01, and *** *p* < 0.001.

## 3. Results

### 3.1. Bladder Cancer Cells Are Sensitive to Scabertopin

The chemical structure of scabertopin is shown in Figure 1A. The two boxes show the α-methylene-γ-lactone and butenolide moieties that confer the drug activity. The FTIR spectrum of scabertopin is shown in Figure 1B, and the UV absorption peaks of scabertopin in the DMEM medium at 0, 24, and 48 h are shown in Figure 1C. According to the Beer–Lambert law, when the length of the absorption cell, the light source, and the type of the substance to be measured are the same, the absorbance is strictly proportional to the concentration of the substance. Since the intensity of the UV maximum absorption peaks of scabertopin at 0, 24, and 48 h were all around 0.55, the concentration of scabertopin did not change. In conclusion, it can be considered that scabertopin can maintain stability in a DMEM medium. The viability of scabertopin-treated bladder cancer cells (T24, J82, RT4, and 5637) and human ureteral epithelial immortalized cells (SV-HUC-1) was determined using the CCK-8 assay. The results showed that scabertopin significantly inhibited the viability of the human bladder cancer cells (J82, T24, RT4, and 5637) in a dose-dependent manner (Figure 1D). The 24 h IC50 of scabertopin for the bladder cancer cell lines was approximately 20 μM, and the 48 h IC50 was even lower (approximately 18 μM). However, the IC50 values of scabertopin for the SV-HUC-1 cells at 24 and 48 h were 59.42 and 55.84 μM, which were considerably higher than those for the bladder cancer cells (Figure 1D). Next, the J82 cell line was arbitrarily selected for further study. In the following section, if the treatment time of scabertopin is not explicitly mentioned, it defaults to 24 h.

### 3.2. Increased Mitochondrial ROS Levels in J82 Cells Treated with Scabertopin

We detected the cells treated with scabertopin for 24 and 48 h using a DCF fluorescent probe and found that scabertopin treatment could significantly increase the content of intracellular ROS (Figure 2A). To determine the mechanism of ROS production caused by scabertopin, we used a JC-1 assay to observe the changes of ΔΨ in the J82 cells treated with scabertopin. The results showed that ΔΨ decreased (red fluorescence decreased while green fluorescence increased) after scabertopin treatment (Figure 2C,D). In conclusion, the ΔΨ of the scabertopin treatment group decreased significantly compared to the control group.

The ROS in mitochondria are usually in the form of hydrogen peroxide (H_2_O_2_) and superoxide anions (O_2_^−^). However, H_2_O_2_ is hardly able to escape through the mitochondrial membrane and be detected in the cytoplasm, and, thus, superoxide anion produced in the mitochondria and transported to the cytoplasm has become a key signaling factor for mitochondrial ROS [31]. Therefore, we used a DHE fluorescent probe to detect the content of superoxide anion in the cells treated with scabertopin. The results showed that the level of superoxide anion in the scabertopin treatment group was significantly higher than that in the control group (Figure 2B).

In addition, GSH is an important antioxidant in cells, and we also detected the changes in intracellular GSH after scabertopin treatment (Figure 2E). The results showed that scabertopin treatment could deplete GSH, which is one of the reasons for the accumulation of intracellular ROS, and the efficacy of scabertopin depleting GSH could be blocked by NAC (Figure 2E).

### 3.3. Cell Death Induced by Scabertopin Treatment Was Not Apoptosis and Ferroptosis

We utilized a scanning electron microscope to observe the effects of scabertopin treatment on cell morphology. The results showed that the cells without scabertopin treatment had strong three-dimensionality and thick and long pseudopodia; after scabertopin treatment, the cell spreading area increased and the pseudopodia shortened and thinned out (Figure 3A). To preliminarily elucidate the potential mechanism of scabertopin cytotoxicity in bladder cancer cells, we analyzed the effect of scabertopin treatment on the potential disruption of cell cycle phases using flow cytometry. The results showed that scabertopin treatment induced a significant increase in the percentage of cells both in the S and G2/M phases in a concentration-dependent manner (Figure 3B). Meanwhile, we detected the effect of scabertopin on the apoptosis and necrosis of J82 cells using the AnnexinV-FITC/PI double-staining method. The transfer of the cell membrane phospholipid phosphatidylserine from the inner to the outer layer of the plasma membrane is one of the early features of apoptosis. A single positive for AnnexinV-FITC is considered a typical cell in early apoptosis, i.e., the fourth quadrant (Q4), whereas AnnexinV-FITC and PI double-positive cells are considered to be in the end stage of apoptosis, necrosis, or are already dead, i.e., the second quadrant (Q2). The results in Figure 3C showed that, under the treatment with gradient concentrations of scabertopin, the number of early apoptotic cells was relatively small. Under scabertopin treatment, the number of necrosis cells significantly increased in a dose-dependent manner. In addition, water-soluble tetrazolium salts and green-fluorescent calcein-AM were deployed to determine whether scabertopin had an effect on the viability of bladder cancer cells. The results showed that, in contrast to the control treatment, scabertopin increased cytotoxicity in a concentration- and time-dependent manner, thus resulting in cell death (Figure 3D). However, scavenging of ROS with NAC significantly rescued scabertopin-induced cell death (Figure 3E). To reveal the mechanism of scabertopin-induced J82 cell death, we verified apoptosis-related caspase proteins Bcl-2 and Bax (Figure 3F), on the one hand, and ferroptosis-related GPX4 and pyroptosis-related GSDMD, on the other hand, using WB (Figure 3G). The results showed that scabertopin-treated J82 cells did not die by apoptosis, ferroptosis, or pyroptosis.

### 3.4. Necroptosis Is a Type of Cell Death Induced by Scabertopin and can Be Inhibited by NAC

To further investigate the extent to which cell organelles were altered by scabertopin treatment and to further determine the manner in which scabertopin induces death in bladder cancer cells, we performed TEM. The results showed that, in contrast to the control group cells, the scabertopin-treated cells were swollen with vacuolated cytoplasm, had severely swollen mitochondria, had a lack of membrane blebbing, had partially dissolved or absent organelles, and showed perforation of cell membranes (Figure 4A). These morphological changes are consistent with the characteristics of necroptosis [32]. We examined the expression of necroptosis-related proteins using Western blotting, and the expressions of phosphorylated RIP1, RIP3 and MLKL significantly increased in the J82 cells treated with scabertopin (Figure 4B). NAC significantly inhibited the expressions of phospho-RIP1, phospho-RIP3, and phospho-MLKL (Figure 4C).

### 3.5. Scabertopin Inhibits the Migration and Invasion of Bladder Cancer Cells

We performed wound healing and transwell assays to characterize how scabertopin affected the migratory and invasive abilities of bladder cancer cells. To this end, 24 h treatment with scabertopin inhibited the wound healing ability of J82 cells in a dose-dependent manner (Figure 5A). Similarly, the results of the transwell assays also showed that, after scabertopin treatment, the invasive ability of cells significantly reduced. This effect was negatively correlated with the concentration of scabertopin (Figure 5B). In addition, we further detected the expressions of motor-related molecules p-FAK, FAK, p-PI3K, PI3K, p-AKT, AKT, and MMP-9 after scabertopin treatment by performing Western blot analysis. The results showed that the expressions of phospho-FAK (Tyr397), phospho-PI3K (Tyr607), phospho-AKT (Ser472, Ser473, Ser474), and MMP-9 decreased significantly after scabertopin treatment in a dose-dependent manner (Figure 5C). Therefore, we suggest that scabertopin can downregulate the expression of MMP-9 by inhibiting the activation of the FAK/PI3K/Akt signaling axis and, ultimately, inhibit the invasiveness of bladder cancer cells. Similarly, we sought to understand the role of ROS in this process. We found that the expressions of phospho-FAK, phospho-PI3K, phospho-Akt, and MMP-9 were significantly inhibited by NAC (Figure 5D). This indicated that ROS played a very important role in scabertopin-mediated invasion and metastasis of J82 cells. These results suggested that scabertopin may be a multifunctional inhibitor for the treatment of bladder cancer.

## 4. Discussion

Despite extensive advances in the treatment of bladder cancer, it remains one of the most recurring and life-threatening tumors. Natural herbs are an important source of potential anticancer compounds in the field of drug discovery and development [33]. Natural compounds themselves contain unique structurally diverse molecules with multiple targets, making them ideal candidates for drug discovery and development.

Scabertopin, a sesquiterpene compound extracted from *E. scaber L*, has attracted interest because of its promising antitumor effects. In the present study, the anti-proliferative ability of scabertopin against various bladder cancer cell lines and human ureteral epithelial immortalized cells (SV-HUC-1) was assessed using the CCK-8 assay. Our results showed that scabertopin significantly inhibited the viability of bladder cancer cells in a dose-dependent manner but had a weak effect on the viability of noncancerous SV-HUC-1 cells. The above results suggested that scabertopin may be a potentially useful agent for bladder tumor treatment. In addition, we also found that J82 cells treated with scabertopin displayed a decrease in ΔΨ and an increase in superoxide anion production. ROS are a different class of molecular oxygen derivatives produced during normal aerobic metabolism. They include peroxides, superoxides, singlet oxygen, and free radicals. ROS levels are higher in different types of cancer cells than in normal cells. However, further elevation of ROS levels increases the susceptibility of cancer cells to oxidative stress-induced cell death [34,35]. Natural active ingredients and their derivatives are one of the main sources of antitumor drugs [7], and studies have shown that natural products can exert antitumor effects by increasing ROS levels. For example, isoalantolactone induces the elevation of ROS levels in human pancreatic ductal epithelial carcinoma PANC-1 cells, arrests these cells in the S phase, thereby inhibiting cell proliferation and inducing apoptosis [36]. We used flow cytometry to detect the effect of scabertopin on the cell cycle of J82 cells. Our results demonstrated that scabertopin could induce cell cycle arrest at the S and G2/M phases in a concentration-dependent manner. In addition, ROS can act as signaling molecules and play a key role in the drug-induced inhibition of tumor invasion and metastasis. In the present study, we likewise observed that scabertopin significantly inhibited cell migration and invasive ability. Cell migration and invasion are key phenotypes that affect the metastasis of tumors. FAK is a cytoplasmic nonreceptor protein tyrosine kinase, a member of the adhesion patch complex family, and an important regulator that mediates cell adhesion to the extracellular matrix (ECM) [37]. The upregulation and hyperphosphorylation of FAK expression have been shown to increase the invasive capacity of several malignancies, including gastric and breast cancers, whereas the inhibition of FAK activity significantly reduces the migration capacity of breast cancer cells [38]. Tyr397 is the major phosphorylation site of FAK and phosphorylation of FAK leads to tumor metastasis and disease progression by promoting migration and invasion [39]. We found that scabertopin could inhibit the expression of MMP-9 in J82 cells by inhibiting the FAK/PI3K/Akt signaling pathway, and the inhibition could be rescued by NAC. MMP-9 is a member of the MMP family and a key enzyme necessary for the degradation of the ECM. The ECM is the first barrier that restricts tumor cells from undergoing migration. Activated MMP-9 can degrade the ECM and basement membrane components, allowing tumor cells to break through the primary site and become invasive and metastatic [40]. Therefore, we propose that scabertopin inhibits the pathway of FAK/PI3K/Akt/MMP-9 axis, which in turn inhibits cell migration and invasion.

In the present study, we found that scabertopin could induce the production and accumulation of ROS in J82 cells, which was identified as superoxide anion-dominated mitochondrial ROS (Figure 2). ROS is one of the important mechanisms that cause ferroptosis in cells, and the level of intracellular ROS accumulation is often positively correlated with the severity of ferroptosis [41]. However, in this study, although we found an increase in ROS, there was no difference in the expression of ferroptosis-related molecules (Figure 3). This phenomenon may be related to the type of ROS. They can cause cell death by damaging DNA, RNA, and lipid molecules [42]. During ferroptosis, the accumulation of lipid peroxides, especially phospholipid peroxides, is considered to be a landmark event of ferroptosis and also a prerequisite for ferroptosis [43]. Accordingly, some scholars refer to lipid peroxides that can specifically cause ferroptosis, such as arachidonoyl (AA)-phosphatidylethanolamine (PE) and adrenoyl (AdA)-PE, as ferroptosis-specific lipid peroxidation [44]. However, this is not rigid, because hydrogen peroxide in the presence of iron ions can be converted into hydroxyl radicals through the Fenton reaction, which in turn oxidizes lipids to form lipid peroxides [45]. While levels of iron, iron-containing proteins [46], and lipid peroxides [47] also promote necroptosis, GPX4 can prevent RIP3-dependent necroptosis in erythroid precursor cells by avoiding lipidic ROS accumulation [48]. Another major difference between ferroptosis and necroptosis is that the cellular morphology of ferroptosis is very unique. Unlike apoptosis or necroptosis, the morphological features of ferroptosis are mainly changes in mitochondrial structure without the shrinkage, rupture, and perforation of the plasma membrane [32]. In tumor cells, the original level of ROS is higher than that of ordinary cells, but abnormally high levels of ROS can also induce different forms of cell death [49]. The occurrence of necroptosis is closely related to the overproduction of ROS [44]. Although the mechanism of ROS in necroptosis is not fully understood, there is a lot of evidence that ROS play a crucial role in many drugs-induced necroptosis [50,51], which is accompanied by mitochondrial injury and decreased expression of MMPs [52]. The key molecules in necroptosis, RIP1 and RIP3, are most closely related to mitochondrial ROS. For example, mitochondrial ROS can mediate the autophosphorylation of RIP1, which subsequently induces necroptosis by recruiting and promoting the phosphorylation of RIP3, suggesting that mitochondrial ROS are the initiators of necroptosis [28]. On the other hand, it has been observed in hepatic stellate cells that the activation of RIP1/RIP3 not only promotes necroptosis, but also increases ROS production via a positive feedback loop [22]. Mediating necroptosis by drug-induced ROS has emerged as a potential approach for tumor therapy because the occurrence of necroptosis not only bypasses apoptosis [53,54], but also causes death in apoptosis-resistant cancer cells. Second, this mechanism of necroptosis induced by ROS can occur in most common cancers. This is consistent with the phenomena we observed, such as membrane perforation and mitochondrial shrinkage, in the scabertopin-treated J82 cells using TEM. Flow cytometry showed that the treatment of scabertopin could increase the number of late apoptotic, necrotic, and dead J82 cells in the Q2 region. However, the expression of apoptosis-related caspase protein, Bcl-2, and Bax displayed no significant difference, which was the same as for ferroptosis. However, necroptosis-related phospho-RIP1, phospho-RIP3, and phospho-MLKL were significantly upregulated by scabertopin treatment and could be inhibited by NAC. Therefore, we believe that scabertopin can promote the increase of mitochondrial ROS by causing a decrease in mitochondrial membrane potential, thereby stimulating the phosphorylation and activation of RIP1/RIP3/MLKL and, finally, triggering necroptosis (Figure 6).

In fact, previous studies have reported the phenomenon of sesquiterpenoid-induced ROS increase. For example, Verma et al. found that isodeoxyelephantopin and deoxyelephantopin can inhibit the activation of NF-κB by inducing the production of ROS and inhibit the growth of breast cancer [55]. Xanthatin, a sesquiterpenoid derived from Xanthium strumarium L, may induce the elevation of ROS, mitochondrial injury, and apoptosis in non-small cell lung cancer (NSCLC) [56]. The ability of sesquiterpenes to induce ROS generation may stem from the fact that they both have α-methylene-γ-lactone and [57] a cyclopentenone ring-like structure (butenolide in scabertopin) (Figure 1A) [58]. The most direct mechanism of sesquiterpenoid-induced ROS elevation may be related to its ability to change the mitochondrial ΔΨ [59]. In fact, there are a variety of ROS inducers that directly act on the mitochondrial voltage-dependent anion channel, resulting in a change in ΔΨ, a decrease in glycolysis, and an increase in ROS [60,61]. From this, we speculate that these unique structures of sesquiterpenes, namely α-methylene-γ-lactone and cyclopentenone ring-like structures, may affect the changes in mitochondrial ΔΨ and cause the increase in ROS. Scabertopin not only has the capacity to trigger this mechanism owing to its unique structure but also meets the rules of five in terms of pharmacokinetics. Since the molecular weight (MW) of SA is 358.4 Da, the logarithm of lipid water partition coefficient (LogP) is 2.6, the hydrogen bond donor count is 0, the hydrogen bond acceptor count is 6, and the rotatable bond count is 3 [62]. These characteristics of SA well meet the Lipinski rules, which requires the MW to be less than 500 Da, the LogP to be between −2 and 5, the hydrogen bond donor count to be less than 5, and the count of hydrogen bond acceptor and the rotatable bond to be less than 10 [63]. In our study, we found that scabertopin significantly reduced ΔΨ, promoted ROS generation, and increased intracellular ROS accumulation by depleting GSH in J82 cells. ROS-induced necroptosis is likely to be associated with decreased GSH [64]. The decrease in GSH, the most important antioxidant-reduction factor in cells, is the result of ROS-induced GSH oxidation, and ROS production is further enhanced by depleting GSH [65]. NAC can supplement GSH by donating cysteine, thereby exerting anti-ROS effect [66]. NAC-treated cells also promote mitochondrial integrity through multiple mechanisms, maintain mitochondrial function, and reduce ROS production, thereby protecting cells from necroptosis [67].

However, we did not evaluate the therapeutic efficacy of scabertopin in vivo, and its pharmacokinetics and pharmacodynamics in vivo remain unclear. Therefore, further studies are needed to provide additional evidence for the use of scabertopin as a chemotherapeutic agent or adjuvant in the chemotherapy of bladder cancer.

## 5. Conclusions

We demonstrate that scabertopin can deplete GSH in bladder cancer cells and cause ROS elevation and accumulation by reducing ΔΨ. It also inhibits bladder cancer cell migration and invasion by targeting the FAK/PI3K/Akt/MMP-9 axis. Furthermore, scabertopin can mediate necroptosis of bladder cancer cells by activating the RIP1/RIP3/MLKL pathway through phosphorylation and inhibit the proliferation and viability of bladder cancer cells. Using the ROS scavenger NAC can not only reduce cell death caused by scabertopin but also rescue J82 cells from scabertopin-mediated necroptosis and limit their invasive and migratory ability by inhibiting the phosphorylation activation of the two abovementioned signaling pathways. In conclusion, our study provides new insights into the anti-bladder cancer mechanisms of scabertopin, suggesting that scabertopin may be a potential alternative drug for bladder cancer therapy.

## Figures and Tables

**Figure 1 cancers-14-05976-f001:**
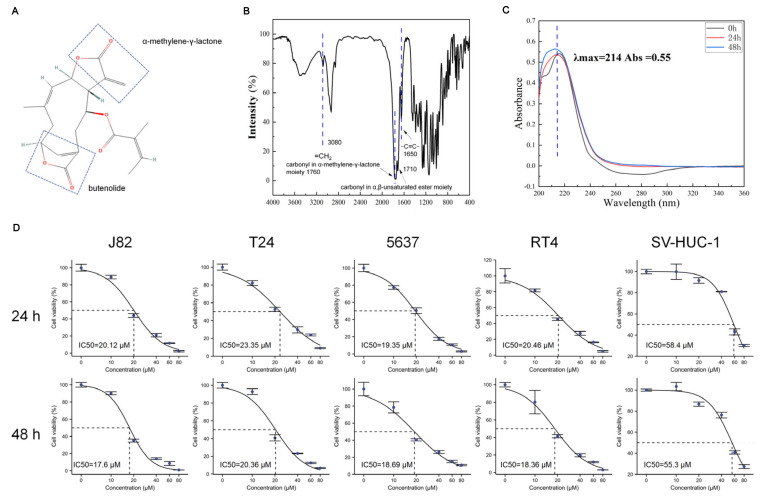
Chemical structure, stability, and efficacy of scabertopin against bladder cancer cells. (**A**). Chemical structure scabertopin. The two boxes show the structure of the molecule with the drug activity of α-methylene-γ-lactone and butenolide. (**B**). Infrared spectrum of scabertopin. The stretching vibration of =CH_2_ exists at 3080 cm^−1^, and the carbonyl stretching vibration of α-methylene-γ-lactone structure exists at 1760 cm^−1^. The peak at 1710 cm^−1^ is the carbonyl stretching vibration of another non-lactone α,β-unsaturated ester, and the C=C stretching vibration exists at 1650 cm^−1^. (**C**). The UV absorption peak of scabertopin in the DMEM medium at 0, 24, and 48 h. (**D**). 24 and 48 h IC50 of bladder cancer cell lines (J82, T24, 5637, and RT4) and human ureteral epithelial immortalized cells (SV-HUC-1) (n = 4).

**Figure 2 cancers-14-05976-f002:**
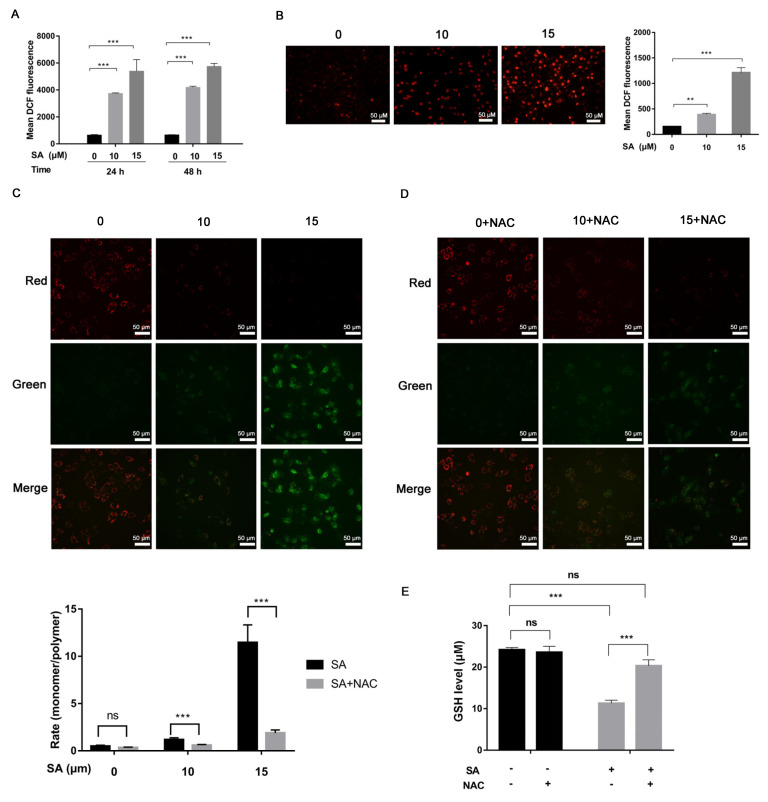
Increased mitochondrial ROS and reduced mitochondrial membrane potential in J82 cells treated with scabertopin. (**A**) Scabertopin induces ROS production in a dose- and time-dependent manner. (**B**) The DHE probe assay shows that superoxide anion is positively correlated with the scabertopin dose. (**C**) The JC-1 assay shows that, after 24 h treatment, scabertopin decreases the mitochondrial membrane potential in J82 cells in a dose-dependent manner. (**C**) The JC-1 assay shows that NAC treatment can restore the mitochondrial membrane potential reduced by 24 h treatment of scabertopin to some extent. (**D**) The DHE probe assay shows that superoxide anion is positively correlated with the scabertopin dose. (**E**) A 24 h treatment with 10 μM of scabertopin is efficient in depleting GSH through a dose-dependent mechanism in J82 cells, which can be reversed with NAC treatment. A total of 5 μM of NAC was used in the experiments. Data represent the mean ± standard error of mean (s.e.m.) of the three independent experiments. ** *p* < 0.01, *** *p* < 0.001, ns: no significance vs. 0 μM scabertopin-treated group. NAC: *N*-acetylcysteine; SA: scabertopin.

**Figure 3 cancers-14-05976-f003:**
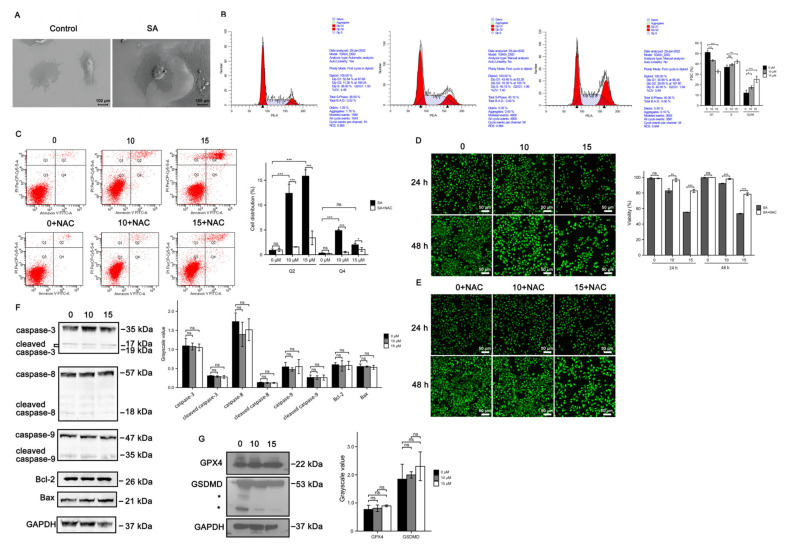
Cell death induced by scabertopin treatment can be rescued by NAC. (**A**) SEM images show that, in J82 cells treated for 24 h with 10 μM of scabertopin, the cell spreading area increases and the cell pseudopods shorten and thin out. The 24 h scabertopin treatment induces cell cycle arrest (**B**) and death (**C**) in J82 cells in a concentration-dependent manner (the black arrowheads in Figure 3B indicate the values of the peaks). The live/death cell assay shows scabertopin inhibits the viability of J82 cells after 24 and 48 h of treatment (**D**), and this effect can be reduced by NAC treatment (**E**,**F**). The expression levels of the apoptosis-related proteins caspase-9, caspase-8, caspase-3, Bax, and Bcl-2 in J82 cells do not significantly change after 24 h of scabertopin treatment. (**G**). Scabertopin does not significantly alter the expression of the ferroptosis- and pyroptosis-related proteins GPX4 and GSDMD, respectively, in J82 cells after 24 h treatment. A total of 5 μM of NAC was used in the experiments, * means nonspecific bands. Data represent the mean ± s.e.m. of the three independent experiments. * *p* < 0.05; ** *p* < 0.01; *** *p* < 0.001, ns: no significance vs. 0 μM scabertopin-treated group. The grayscale values are normalized to GAPDH. Bax: Bcl-2-associated X; Bcl-2: B cell lymphoma-2; GPX4: glutathione peroxidase 4; GSDMD: Gasdermin-D; NAC: *N*-acetylcysteine; PE-A: phycoerythrin area; PerCP-Cy5-5-A: Peridinin-Chlorophyll-Protein Complex-Cyanine5.5 area; SA: scabertopin.

**Figure 4 cancers-14-05976-f004:**
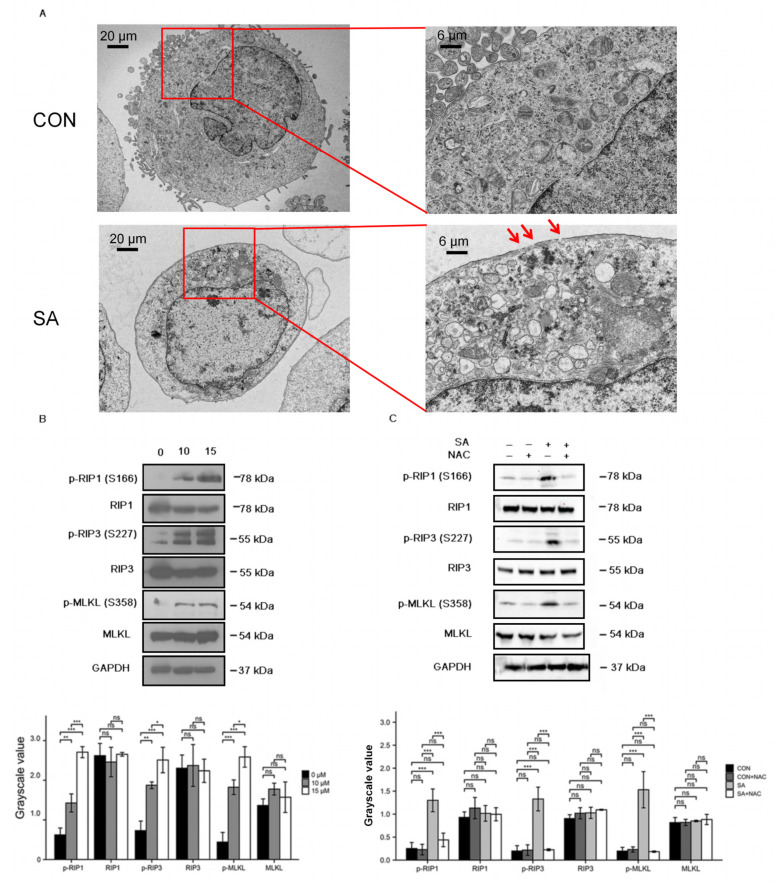
Scabertopin-treated J82 cells undergo necroptosis, which can be rescued by NAC. (**A**) TEM images are shown for the control group (CON) and the J82 cells treated with 10 μM of scabertopin (SA) for 24 h. The image on the right is an enlarged image of the left image, and the red arrow points to the perforation on the cell membrane during necroptosis; at least three independent samples were observed in each group. (**B**) Scabertopin can significantly increase the phosphorylation of the necroptosis-related proteins RIP1, RIP3, and MLKL (n = 3) after 24 h of treatment; (**C**) Scabertopin-induced phosphorylation of RIP1, RIP3, and MLKL can be reversed by NAC in the group treated with 10 μM of scabertopin for 24 h. A total of 5 μM of NAC was used in the experiments. The grayscale values are normalized to GAPDH. Data represent the mean ± s.e.m. of the three independent experiments. * *p* < 0.05, ** *p* < 0.01, *** *p* < 0.001, ns: no significance vs. 0 μM scabertopin-treated group. MLKL: mixed lineage kinase like; NAC: N-acetylcysteine; p-: phosphorylated; RIP1: receptor-interacting protein.

**Figure 5 cancers-14-05976-f005:**
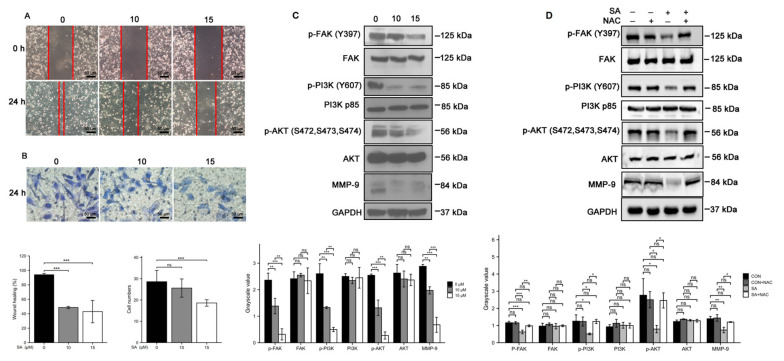
Scabertopin treatment inhibits the migratory and invasive ability of J82 cells by inhibiting the FAK/PI3K/Akt/MMP-9 signaling pathway, which can be rescued by NAC. Scabertopin significantly inhibits the migration ability (difference in area reduction between red lines) (n = 3) (**A**) and invasion of (n = 5) (**B**) of J82 cells treated with gradient concentrations of scabertopin for 24 h. (**C**) Phospho-FAK, phospho-PI3K, phospho-Akt, and MMP-9 expression levels significantly decrease in J82 cells after 24 h of scabertopin treatment. (**D**) Scabertopin-induced phosphorylation of p-FAK, p-PI3K, p-Akt, and MMP-9 expression can be reversed by NAC. A total of 5 μM of NAC was used in the experiments. The grayscale values are normalized to GAPDH. Data are presented as the mean ± s.e.m. of the three independent experiments (n = 3). * *p* < 0.05, ** *p* < 0.01, *** *p* < 0.001, ns: no significance vs. 0 μM scabertopin-treated group. AKT: Akt-protein kinase B; CON: control group; FAK: focal adhesion kinase; MMP-9: matrix metalloprotease-9; NAC: N-acetylcysteine; p-: phosphorylated; PI3K: phosphoinositide 3-kinase; SA: scabertopin.

**Figure 6 cancers-14-05976-f006:**
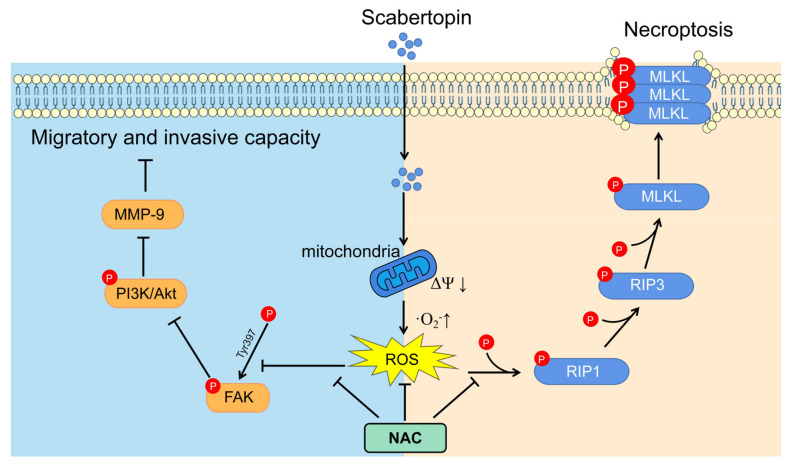
Mechanism of scabertopine inhibiting invasion and promoting necroptosis in human bladder cancer J82 cells. Scabertopin can reduce mitochondrial membrane potential and stimulate mitochondrial ROS production, thereby activating RIP1/RIP3/MLKL phosphorylation, mediating J82 cell necroptosis, and inhibiting the FAK/PI3K/Akt/MMP-9 signaling pathway. This in turn inhibits the migration and invasive potential of J82 cells. Akt: Akt-protein kinase B; FAK: focal adhesion kinase; MLKL: mixed lineage kinase like; MMP-9: matrix metalloprotease 9; NAC: *N*-acetylcysteine; PI3K: phosphoinositide 3-kinase; p-: phosphorylated; RIP: receptor-interacting protein.

## Data Availability

The data can be shared upon request.

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
