# Peer review of "Scabertopin Derived from Elephantopus scaber L. Mediates Necroptosis by Inducing Reactive Oxygen Species Production in Bladder Cancer In Vitro"

_cancers, 2022, doi:10.3390/cancers14235976_

Round 1
Reviewer 1 Report
Manuscript is interesting and it deserves to be published after some minor changes that I mentioned below:
Manuscript presents the effect of scabertopin isolated from Elephantopus scaber on necroptosis in bladder cancer cells. In my opinion the title of the manuscript should be changed a little bit because it is a little confusing. At first readers may have thought that experiments were done in vivo, not in vitro, because authors did not include that information in the title. All the analysis were performed in cancer cell lines, not in bladder cancer. So it is misunderstanding.
An abstract is comprehensive and it contains all the necessary information. In keywords the species name:"Elephantopus" should start with capital letter. The Introduction part of the manuscript is interesting, however tha aim of the study shoul be slightly modified and explain in more details. Did authors mean that they would like to check if studied compound acts as a cancer cells inhibitor and on what molecular pathways? I think that they should explain possibilities of application of this compound as a possible anticancer drug.
Materials and methods part of the manuscript should be modified:
What was the final concentration of a solvent used for the solution of scabertopin? Was it DMSO? The final concentration of DMSO used as a solvent in cell culture medium should not exceed 1%.
Did authors mean RPMI1640 when they have written "1640 medium"?
Where from did authors obtain two cell lines: 5637 and RT4? They mentioned only that J82, T24 and SH-HUC-1 they obtained from ATCC and others? Where from?
Results part of the manuscript:
In figure 3 there is a lack of "A" in the description of fig.3.
Discussion is interesting and comprehensive.
Author Response
Dear Reviewers and Editors
Thank you very much for your valuable and insightful comments and suggestions which have greatly improved the quality of our manuscript. On behalf of all the authors, I would like to thank you for your efforts. We have carefully revised the manuscript according to your comments: all revisions are shown in the manuscript with tracked changes and distinguishable by different colors (green for revision based on reviewer 1's comments and yellow for reviewer 2). "
We now address both reviewers’ comments point counterpoint below:
You can also download the attachment.
To Reviewer 1
Manuscript is interesting and it deserves to be published after some minor changes that I mentioned below:
Q1. Manuscript presents the effect of scabertopin isolated from Elephantopus scaber on necroptosis in bladder cancer cells. In my opinion the title of the manuscript should be changed a little bit because it is a little confusing. At first readers may have thought that experiments were done in vivo, not in vitro, because authors did not include that information in the title. All the analysis were performed in cancer cell lines, not in bladder cancer. So it is misunderstanding.
Answer 1: Thank you for picking this up. According to your suggestions, we have updated the title to “Scabertopin derived from Elephantopus scaber L. mediates necroptosis by inducing reactive oxygen species production in bladder cancer in vitro ”
Q2. An abstract is comprehensive and it contains all the necessary information. In keywords the species name:"Elephantopus" should start with capital letter. The Introduction part of the manuscript is interesting, however tha aim of the study shoul be slightly modified and explain in more details. Did authors mean that they would like to check if studied compound acts as a cancer cells inhibitor and on what molecular pathways? I think that they should explain possibilities of application of this compound as a possible anticancer drug.
Answer 2: Thank you for your acknowledgement of the part of abstract and introduction in our manuscript. We have included the keywords “elephantopus scaber L” to “Elephantopus scaber L”. In addition, we have taken your suggestion and explained in more detail in the aim of the study section. “In this study, we aimed to evaluate the inhibitory effects and potential molecular mechanisms of scabertopin on bladder cancer cells in vitro. Scabertopin is one of the four sesquiterpene lactones with the highest content in Elephantopus scaber L. Since DET has shown a good anti-tumor effect in a variety of tumor cell lines, we speculate that scabertopin also has similar efficacy. Therefore, we determined our research object as the effect of scabertopin on bladder cancer cell lines, and explore its way to promote the death of bladder cancer cell lines. ”
Materials and methods part of the manuscript should be modified:
Q3. What was the final concentration of a solvent used for the solution of scabertopin? Was it DMSO? The final concentration of DMSO used as a solvent in cell culture medium should not exceed 1%.
Answer 3: Scabertopin is dissolved in DMSO and the final concentration of DMSO is 0.143%. We regret that this was not stated in detail and clearly, so we have added appropriate additions to the original text to explain this.
Scabertopin (10mg) was dissolved in 1 ml DMSO and configured into a master batch with a concentration of 40 μM (The maximum concentration of DMSO in the cell culture medium is 0.143% ).
Q4. Did authors mean RPMI1640 when they have written "1640 medium"?
Answer 4: We are very sorry for the incorrect use of the name RPMI 1640 and we have fixed all such errors in the manuscript.
RPMI 1640 (PM150110) was purchased from Procell Life Science&Technology Co.,Ltd(Wuhan, China).
T24 and 5637 cells were cultured in RPMI 1640 (10% FBS);
Q5. Where from did authors obtain two cell lines: 5637 and RT4? They mentioned only that J82, T24 and SH-HUC-1 they obtained from ATCC and others? Where from?
Answer 5: The source of 5637 and RT4 cell lines has been added: 5637 and RT4 cell lines were purchased from the China Centre for Type Culture Collection (Shanghai, China).
Results part of the manuscript:
Q6. In figure 3 there is a lack of "A" in the description of fig.3.
Answer 6: Added.

Reviewer 2 Report
Major points:
1) Figures 1A–D; 3B,C,F,G; 4A–C appear too pixelated. Please enlarge and/or increase pixel density.
2) The resolution of Figures 2B–D; 3A,D,E; 4A; 5A,B is poor. Please enlarge and/or increase pixel density so that individual cellular features can be clearly distinguished.
3) The resolution of Western blots presented in Figures 3F,G; 4B,C; 5C,D is poor. Please enlarge and/or increase pixel density so that individual bands can be clearly seen.
Minor points:
1) Please format "2" in "2 School of Materials Science and Engineering, Hainan University, Haikou, Hainan 570228, P.R.China" using superscript (page 1).
2) Please align "School of Materials Science and Engineering, Hainan University, Haikou, Hainan 570228, P.R.China" horizontally to precisely match the position of other headings in the Affiliations section.
3) Please change "P.R.China" to "P. R. China" (page 1).
4) Please replace "promoting the production of mitochondrial ROS" with "promoting the production of mitochondrial reactive oxygen species (ROS)" (page 1).
5) Please replace "inhibited the migration and invasion" with "inhibit the migration and invasion" (page 1).
6) Please change "necroptosis; elephantopus scaber L" to "necroptosis; Elephantopus scaber L" (page 1).
7) Please replace "2015[3]" with "2015 [3]" (page 1).
8) Please change "radiotherapy[4,5]" to "radiotherapy [4,5]" (page 1).
9) Please replace "down-regulate" with "down-regulate" (pages 2, 10).
10) Please change "[14].In addition" to "[14]. In addition" (page 2).
11) Please replace "toxicity[18,19]" with "toxicity [18,19]" (page 2).
12) Please change "reported at all" to "reported" (page 2).
13) Please replace "consequently trigger inflammation" with "consequent inflammatory response" (page 2).
14) Please change "the abovementioned process[22,23]" to "necroptosis [22,23]" (page 2).
15) Please replace "necroptosis[27]" with "necroptosis [27]" (page 2).
16) Please change "key residues of cysteine" to "key cysteine residues" (page 2).
17) Please replace "adhesion patch kinase" with "focal adhesion kinase" (page 3).
18) The catalog numbers "#8121526" and "#1798320" mentioned in "Scabertopin and deoxyelephantopin was purchased from Chengdu Pusi Biotechnology Co., Ltd. DMEM basal medium (#8121526) and trypsin solution (#1798320) were purchased from Gibco" (page 3) do not seem to be correct as they seem to represent lot numbers. Please fix.
19) Please change "Scabertopin and deoxyelephantopin was" to "Scabertopin and deoxyelephantopin were" (page 3).
20) Please double-check that the correct company name is "Chengdu Pusi Biotechnology" and not "Chengdu Push Biotechnology" mentioned in "Scabertopin and deoxyelephantopin was purchased from Chengdu Pusi Biotechnology Co., Ltd. DMEM basal medium (#8121526) and trypsin solution (#1798320) were purchased from Gibco" (page 3) and "Scabertopin was purchased from Chengdu Pusi Biotechnology Co.,Ltd. Scabertopin was dissolved in 1ml DMSO and configured into a master batch with a concentration of 10mg/ml (40 μM/L)" (page 3).
21) Please provide city and state name for the headquarters of "Chengdu Pusi Biotechnology Co., Ltd." (page 3), "Gibco" (page 3), Procell (page 3), "BI Company" (page 3), "Dojindo Corporation" (page 3), "Invitrogen" (page 3), "KeyGen BioTECH" (page 3), "Beyotime Biotechnology Company" (page 3), "Abcam" (page 3), "Cell Signaling" (page 3), "Beijing Labtech Instrument Co., Ltd." (page 3), "BD" (page 6).
22) Please replace "#8121526" with "8121526" (page 3).
23) Please change "#1798320" to "1798320" (page 3).
24) Please replace "#PM150110" with "PM150110" (page 3).
25) Please change "#1907301" to "1907301" (page 3).
26) Please define abbreviation for "BI" (page 3), "FTIR" (page 3), "DHE" (page 5), "GSH" (page 5), "GSSG" (page 5), "PI" (page 6), "TPEB" (page 6), "BCA" (page 6).
27) Please replace "BI Company" with "BI" (page 3).
28) The catalog number "#SA616" mentioned in "Cell Counting Kit-8 (#SA616) was purchased from Dojindo Corporation" does not seem to be correct (page 5). Please fix.
29) Please change "#SA616" to "SA616" (page 3).
30) Please replace "Dojindo Corporation" with "Dojindo Laboratories" or "Dojindo Molecular Technologies" (page 3).
31) The catalog number "#1915835" mentioned in "LIVE/DEAD cell viability assay kit (#1915835) was purchased from Invitrogen" does not seem to be correct (page 5). Please fix.
32) Please change "#1915835" to "1915835" (page 3).
33) The catalog numbers "#20201111" and "#20210114" mentioned in "The cell cycle assay kit (#20201111) and the apoptosis assay kit (#20210114) were purchased from KeyGen BioTECH" do not seem to be correct (page 5). Please fix.
34) Please replace "#20201111" with "20201111" (page 3).
35) Please change "#20210114" to "20210114" (page 3).
36) Please replace "#S0033S" with "S0033S" (page 3).
37) Please change "Beyotime Biotechnology Company" to "Beyotime Biotechnology" (page 3).
38) Please replace "BCL-2-associated" with "Bcl-2-associated" (page 3).
39) Please change "(AKT1-3, ab179463)" to "(AKT1-3, ab179463)," (page 3).
40) Please replace "#71433" with "71433" (page 3).
41) Please replace "Phospho-FAK(Y397, #8556)" with "phospho-FAK (Y397, 8556)" (page 3).
42) Please change "#13667" to "13667" (page 3).
43) Please replace "#4257S" with "4257S" (page 3).
44) Please change "Caspase-3 (#14220)" to "caspase-3 (14220)" (page 3).
45) Please replace "Caspase-8 (#9746)" with "caspase-8 (9746)" (page 3).
46) Please change "Caspase-9 (#9508)" to "caspase-9 (9508)" (page 3).
47) Please replace "#14993" with "14993" (page 3).
48) Please change "(S358,#91689)" to "(S358, 91689)" (page 3).
49) Please replace "#3493" with "3493" (page 3).
50) Please change "#65746" to "65746" (page 3).
51) Please change "RIP3 (#13526) and Phospho-RIP3 (Ser227, #93654)" to "RIP3 (13526), and phospho-RIP3 (Ser227, 93654)" (page 3).
52) Please replace "Cell Signaling company" with "Cell Signaling Technology" (page 3).
53) Please change "Scabertopin was purchased from Chengdu Pusi Biotechnology Co.,Ltd. Scabertopin" to "Scabertopin" (page 3).
54) Please replace "1ml DMSO" with "1 ml DMSO" (page 3).
55) Please change "10mg/ml (40 μM/L)" to "10 mg/ml (40 μM/l)" (page 3).
56) Please provide the manufacturer and its city and state headquarters for "potassium bromide pressed tablets" mentioned in "Scabertopin and deoxyelephantopin solution was taken 10 μl, applied evenly on potassium bromide pressed tablets, evaporated and dehydrated by infra-red lamp, and analyzed by FTIR spectroscopy using KBr pressed plates" (page 3).
57) Please replace "Scabertopin and deoxyelephantopin solution was taken 10 μl, applied evenly on potassium bromide pressed tablets" with something like "10 μl of scabertopin and deoxyelephantopin solution was applied evenly on KBr tablets" (page 3).
58) Please change "KBr pressed plates" to "KBr-pressed plates" (page 3).
59) Please provide city name for the headquarters of "Nexus" (page 3).
60) Please replace "infrared spectrum" with "IR spectrum" (page 3).
61) Please change "400-4000cm-1" to "400–4000 cm-1" (page 3).
62) Please format "-1" in "400–4000cm-1" using superscript (page 3).
63) Please replace "32times" with "32" (page 3).
64) Please change "4cm-1" to "4 cm-1" (page 3).
65) Please format "-1" in "4cm-1" using superscript (page 3).
66) Please replace "The operation is as follows: grind potassium bromide" with "The operation was as follows: grind KBr" (page 3).
67) Please change "10mg potassium bromide and 1mg" to "10 mg potassium bromide and 1 mg" (page 3).
68) Please replace "agate mortar and evenly" with "agate mortar evenly" (page 3).
69) Please change "Infrared spectroscopy" to "IR spectroscopy" (page 3).
70) Please replace "(UV9100B from Beijing Labtech Instrument Co., Ltd.), with" with "(UV9100B from Beijing LabTech Instruments Co., Ltd.) with" (page 3).
71) Please change "200-360nm and a resolution of 2nm" to "200–360nm and a resolution of 2 nm" (page 3).
72) Please replace "ultraviolet maximum" with "UV maximum" (page 3 2x).
73) Please change "ultraviolet spectrum" to "UV spectrum" (page 3 2x).
74) Please replace "214nm" with "214 nm" (page 3).
75) Please change "α,β-unsaturated carbonyl structure" to "α,β-unsaturated carbonyl moiety" (page 3).
76) Please replace "There is no change in 0h, 24h and 48h" with "There was no change in 0, 24, and 48 h" (page 3).
77) Please change "Rambeau Beer's law: When" to "Beer-Lambert law that when" (page 3).
78) Please format "2" in "cm2·mol" using superscript (page 4).
79) Please replace "mol/L" with "mol/l" (page 4).
80) Please provide catalog number for "FBS" mentioned in "All cell culture media and fetal bovine serum (FBS) were obtained from Gibco" (page 4).
81) Please change "Dojindo, #CK04" to "Dojindo Laboratories, CK04" or "Dojindo Molecular Technologies, CK04" (page 4).
82) Please change "J82 was" to "J82 cells were" (page 4).
83) Please replace "RT4 was" with "RT4 cells were" (page 4).
84) Please format "2" in "CO2" using subscript (page 4).
85) Please change "BIO-RAD" to "Bio-Rad" (page 4).
86) Please provide city name for the headquarters of "BIO-RAD" (page 4).
87) Please replace "mean ±standard" with "mean ± standard" (page 4).
88) Please change "generated by using" to "generated using" (page 4).
89) Please provide city name for the headquarters of "GraphPad Software" (page 4).
90) Please replace "1 × 104" with "1×104" (page 4).
91) Please provide the manufacturer and its city and state headquarters for "LIVE/DEAD Cell Vitality Assay Kit" mentioned in "Subsequently, the cells were washed with PBS in accordance with the instructions of the LIVE/DEAD Cell Vitality Assay Kit, and each group was incubated for 30 min at room temperature with an appropriate amount of calcein (AM) and ethidium dimers (EthD-1)" (page 4).
92) It is not clear what the authors mean by "appropriate amount of calcein (AM) and ethidium dimers (EthD-1)" in "Subsequently, the cells were washed with PBS in accordance with the instructions of the LIVE/DEAD Cell Vitality Assay Kit, and each group was incubated for 30 min at room temperature with an appro-priate amount of calcein (AM) and ethidium dimers (EthD-1)" (page 4)?
93) Please replace "Cell Vitality Assay Kit" with "cell vitality assay kit" (page 4).
94) Please change "24-well plate then treated with the drug at the concentration of 10μM" to "24-well plate, then treated with the drug at the concentration of 10 μM" (page 4).
95) Please replace "After being cultured in the incubator for 24h" with "After being cultured in the incubator for 24 h" (page 4).
96) Please provide manufacturer and its city headquarters for "SU8100" mentioned in "Then, their morphology was observed by using a scanning electron microscope (SU8100, Japan)" (page 4).
97) Please change "J82 cells were inoculated into six-well" to "J82 cells were inoculated into 6-well" (page 4).
98) Please replace "2.9. Measurement of ROS" with "2.9. Measurement of reactive oxygen species" (page 5).
99) Please change "gradient concentrations of scabertopin for 24-48h" to "gradient concentrations of scabertopin for 24–48h" (page 5).
100) Please replace "cells were incubated with Scabertopin for 2 hours" with "cells were incubated with scabertopin for 2 h" (page 5).
101) Please provide the molar concentration for NAC mentioned in "In rescue experiments, cells were incubated with Scabertopin for 2 hours with 1:200 ROS scavenger N-acetylcysteine (NAC) (MCE#HY-B0215)" (page 5).
102) Please provide city and state name for the headquarters for "MCE" mentioned in "In rescue experiments, cells were incubated with Scabertopin for 2 hours with 1:200 ROS scavenger N-acetylcysteine (NAC) (MCE#HY-B0215)" (page 5).
103) Please change "MCE#HY-B0215" to "MCE, HY-B0215" (page 5).
104) "The cells were washed three times with PBS and treated with 10 μM dichlorofluorescein diacetate in situ loading probe for 20–30 min" (page 5) does not seem to be grammatically correct with respect to "in situ loading probe for 20–30 min". Please rephrase.
105) Please provide city name for the headquarters of "BiOTek" mentioned in "Subsequently, the cells were washed three times with serum-free medium, and their fluorescence intensity was measured with a fluorescence microplate reader (BiOTek, USA)" (page 5).
106) Please change "BiOTek" to "BioTek" (page 5).
107) Please replace "at a concentration of 25μM" with "at a concentration of 25 μM" (page 5).
108) Please change "fresh culture medium was used to washed, and the fluorescence intensity of the cells" to "cells were washed with fresh culture medium, the fluorescence intensity" (page 5).
109) Please replace "the fluorescence intensity of the cells was measured by a fluorescence microplate reader (BiOTek, USA), and" with "the fluorescence intensity of the cells was measured by a fluorescence microplate reader (BioTek, USA). Cells were" (page 5).
110) Please change "KeyGEN. #KGAF019" to "KeyGen BioTECH, KGAF019" (page 5).
111) Please replace "Mitochondrial membrane potential assay kit JC-1 kit" with "Mitochondrial membrane potential assay kit with JC-1" (page 5).
112) Please replace "Beyotime Biotechnology #C2006" with "Beyotime Biotechnology, C2006" (page 5).
113) It is not exactly clear what the authors mean by "200× JC-1" in "The cells were treat with gradient concentrations of scabertopin, and then washed with PBS, added to cell culture medium and mixed with 200× JC-1, put into 37°C incubator for 20min" (page 5)?
114) Please change "and then washed with PBS" to "then washed with PBS" or "washed with PBS" (page 5).
115) Please replace "put into 37°C incubator for 20min" with "incubated at 37 °C for 20 min" (page 5).
116) Please change "The fluorescence intensity of the cells was measured by the fluorescence microplate reader (BIOTek, USA)," to "The fluorescence intensity of the cells was measured by the fluorescence microplate reader (BioTek)" (page 5).
117) Please change "GSH and GSSG Assay Kit (Beyotime Biotechnology #S0053) is a simple and easy-to-use assay that can detect the content of GSH (reduced glutathione) and GSSG (oxidized glutathione disulfide)" to "GSH (reduced glutathione) and GSSG (oxidized glutathione disulfide) assay kit (Beyotime Biotechnology, S0053) is a simple and easy-to-use assay that can detect the content of GSH and GSSG" (page 5).
118) Please replace "Total Glutathione (GSH+GSSG)- 2×GSSG" with "total glutathione (GSH + GSSG) - 2×GSSG" (page 5).
119) Please format the formula "GSH = Total Glutathione (GSH+GSSG)- 2×GSSG" (page 5) using the same style as for the "Rambeau Beer's law" (page 4).
120) It is not exactly clear what the authors mean by "insert area" in "J82 cells were inoculated at the density of 8 × 105 cells/mL into the insert area in the middle of a dish" (page 5)?
121) Please change "8 × 105 cells/mL" to "8×105 cells/ml" (page 5).
122) It is not clear what the authors mean by "cells were walled" in "After the cells were walled, the insert was removed with forceps, and the old medium was aspirated off" (page 5)?
123) Please replace "removed with forceps, and" with "removed with forceps and" (page 5).
124) Please change "with PBS 1–2 times" to "1–2 times with PBS" (page 5).
125) Please replace "Olympus, IX71" with "IX71, Olympus" (page 5).
126) Please provide city name for the headquarters of "Olympus" mentioned in "The cells were washed gently with PBS 1–2 times, treated with different concentrations of the drugs, placed in the incubator for further incubation, and removed at 0 and 24 h for fluorescent inverted microscopy (Olympus, IX71, Japan) to observe whether the peripheral cells had migrated to the central scratch area" (page 5).
127) Please format the formula "(initial area – area at a certain time point/initial area)" (page 5) using the same style as for the "Rambeau Beer's law" (page 4).
128) Please change "that had been treated" to "treated" (page 5).
129) Please replace "drugs for 24h were collected, and" with "drugs for 24 h were collected and" (page 5).
130) Please change "5×104cells/well" to "5×104 cells/well" (page 5).
131) Please replace "Transwell plate, and" with "transwell plate and" (page 5).
132) Please change "lower chamber was added" to "lower chamber was supplemented" (page 5).
133) Please replace "0.1%Giemsa staining solution" with "0.1% Giemsa staining solution" (page 5).
134) Please change "different concentrations of drugs for 24h" to "different concentrations of drugs for 24 h" (page 6).
135) Please replace "1-2 times with PBS" with "1–2 times with PBS" (page 6).
136) Please change "ethanol, then" to "ethanol and then" (page 6).
137) Please replace "cells were added with" with "cells were administered with" (page 6).
138) Please change "(BD FACSCanto), and" to "(BD FACSCanto) and" (page 6).
139) It is not clear what the authors mean by "Bing Buffer" in "The precipitates were resuspended again with a small amount of Bing Buffer, mixed with Annexin-V-FITC mixed working solution, incubated for 5 min at room temperature under protection from light, and mixed with PI reagent and PBS" (page 6)?
140) Please replace "Bing Buffer" with "bing buffer" (page 6).
141) Please change "cytometry, and the" to "cytometry and the" (page 6).
142) Please specify the composition of the "TPEB buffer" and the "protease inhibitor" and "phosphatase inhibitor" mixtures mentioned in "The cells were mixed with TPEB lysis buffer (1% protease inhibitor and 1% phospha-tase inhibitor mixture) on ice after 24 h of scabertopin treatment" (page 6).
143) Please provide the manufacturer and its city and state headquarters for "BCA kit" mentioned in "The lysates were collected and centrifuged at 14,000×g for 10 min at 4 °C. Protein concentrations were analyzed by using a BCA kit" (page 6).
144) Please provide the manufacturer and its city and state headquarters for "10% precast gel" mentioned in "Equal amounts of protein samples were separated through electrophoresis on a 10% precast gel" (page 6).
145) Please replace "different primary antibodies, including" with "different primary antibodies including" (page 6).
146) Please change "(dilution ratio: 1/10000), Bcl-2 (dilution ratio: 1/10000), Caspase-9 (dilution ratio: 1/1000), Caspase-3 (dilution ratio: 1/1000), Caspase-8 (dilution ratio: 1/1000), GAPDH ( dilution ratio: 1/10000), MMP-9 (dilution ratio: 1/500), PI3K (dilution ratio: 1/1000), p-PI3K (dilution ratio: 1/1000), AKT(dilution ratio: 1/1000), p-AKT(dilution ratio: 1/1000), glutathione peroxidase 4 (GPX4, dilution ratio: 1/1000), Gasdermin-D (GSDMD, dilution ratio: 1/1000), RIP (dilution ratio: 1/1000), p-RIP (dilution ratio: 1/1000), MLKL (dilution ratio: 1/1000), p-MLKL (dilution ratio: 1/1000), RIP3 (dilution ratio: 1/1000), and p-RIP3 (dilution ratio: 1/1000)" to "(1/10000), Bcl-2 (1/10000), caspase-9 (1/1000), caspase-3 (1/1000), caspase-8 (1/1000), GAPDH (1/10000), MMP-9 (1/500), PI3K (1/1000), p-PI3K (1/1000), AKT(1/1000), p-AKT(1/1000), glutathione peroxidase 4 (GPX4, 1/1000), gasdermin D (GSDMD, 1/1000), RIP (1/1000), p-RIP (1/1000), MLKL (1/1000), p-MLKL (1/1000), RIP3 (1/1000), and p-RIP3 (1/1000)" (page 6).
147) Please replace "dilution ratio: 1/2000–1/20000" with "1/2000–1/20000" (page 6).
148) Please provide city name for the headquarters of "MerckMillipore" mentioned in "The film was treated overnight with ECL luminescent solution (MerckMillipore, USA)" (page 6).
149) From "The target protein bands were observed and scanned after development and fixation" (page 6) is not clear how the "target protein bands" were scanned?
150) Please change "Statistics and bar graphs are" to "Statistics and bar graphs were" (page 6).
151) Please replace "The chemical structure is shown in Fig.1A" with "The chemical structure of scabertopin is shown in Fig. 1A" (page 6).
152) Please change "structure of the molecule with drug activity, α-methylene-γ-lactone and butenolide" to something like "α-methylene-γ-lactone and butenolide moieties that confer the drug activity" (page 6).
153) Please replace "The ability of migratory and invasive" to "Migratory and invasive capacity" and "Try397" with "Tyr397" in Figure 6.
154) Please change "Mechanism ofsabertopine" to "Mechanism of scabertopin" (page 13).
155) Please replace "Verma et al[57]" with "Verma et al. [57]" (page 13).
Author Response
Dear Reviewers and Editors
Thank you very much for your valuable and insightful comments and suggestions which have greatly improved the quality of our manuscript. On behalf of all the authors, I would like to thank you for your efforts. We have carefully revised the manuscript according to your comments: all revisions are shown in the manuscript with tracked changes and distinguishable by different colors (green for revision based on reviewer 1's comments and yellow for reviewer 2).
We now address both reviewers’ comments point counterpoint below:
You can also download the attachment.
To Reviewer 2
Major points:
- Figures 1A–D; 3B,C,F,G; 4A–C appear too pixelated. Please enlarge and/or increase pixel density.
Answer: We are sorry for the poor quality of the images presented to you. Our high-quality figures may have been pixelated in the process of uploading and generation of the pdf file. We have incorporated these figures below for your perusal.
Figures are shown in attachment.
- The resolution of Figures 2B–D; 3A,D,E; 4A; 5A,B is poor. Please enlarge and/or increase pixel density so that individual cellular features can be clearly distinguished.
Answer: Likewise, we apologize for the quality of the images presented to you. Please see our response to your Question 1.
- The resolution of Western blots presented in Figures 3F,G; 4B,C; 5C,D is poor. Please enlarge and/or increase pixel density so that individual bands can be clearly seen.
Answer: Please kindly refer to our response to your Question 1.
Minor points
- Please format "2" in "2 School of Materials Science and Engineering, Hainan University, Haikou, Hainan 570228, P.R.China" using superscript (page 1).
Answer: We have made adjustments to this formatting issue.
1 Central Laboratory, Affiliated Haikou Hospital of Xiangya Medical College, Central South University, Haikou, Hainan 570208, P. R. China
2 School of Materials Science and Engineering, Hainan University, Haikou, Hainan 570228, P. R. China
# These authors have contributed equally to this work.
* Correspondence: Shufang Zhang, zsf66189665@126.com, Xiaolong Yu, yuxiaolong@hainanu.edu.cn
- Please align "School of Materials Science and Engineering, Hainan University, Haikou, Hainan 570228, P.R.China" horizontally to precisely match the position of other headings in the Affiliations section.
Answer: We have made adjustments to this formatting issue as above.
- Please change "P.R.China" to "P. R. China" (page 1).
Answer: We have made adjustments to this formatting issue as above.
- Please replace "promoting the production of mitochondrial ROS" with "promoting the production of mitochondrial reactive oxygen species (ROS)" (page 1).
Answer: We have completed revision based on your suggestion: Our results suggest that scabertopin can induce RIP1/RIP3-dependent necroptosis in bladder cancer cells by promoting the production of mitochondrial reactive oxygen species (ROS),
- Please replace "inhibited the migration and invasion" with "inhibit the migration and invasion" (page 1).
Answer: We have completed revision based on your suggestion: and ultimately inhibit the migration and invasion ability of bladder cancer cells.
- Please change "necroptosis; elephantopus scaber L" to "necroptosis; Elephantopus scaber L" (page 1).
Answer: We have completed revision based on your and reviewer 1 suggestion: Elephantopus scaber L.
- Please replace "2015[3]" with "2015 [3]" (page 1).
Answer: We have completed revision based on your suggestion.
- Please change "radiotherapy[4,5]" to "radiotherapy [4,5]" (page 1).
Answer: We have completed revision based on your suggestion.
- Please replace "down-regulate" with "down-regulate" (pages 2, 10).
Answer: We are not sure about the adjustment required here. We have revised the format of the dash on pages 2 and 10. If we misunderstand your meaning, please let us know so that we can revise it again.
- Please change "[14].In addition" to "[14]. In addition" (page 2).
Answer: We have completed revision based on your suggestion.
- Please replace "toxicity[18,19]" with "toxicity [18,19]" (page 2).
Answer: We have completed revision based on your suggestion.
- Please change "reported at all" to "reported" (page 2).
Answer: We have completed revision based on your suggestion.
- Please replace "consequently trigger inflammation" with "consequent inflammatory response" (page 2).
Answer: We have completed revision based on your suggestion.
- Please change "the abovementioned process[22,23]" to "necroptosis [22,23]" (page 2).
Answer: We have completed revision based on your suggestion.
- Please replace "necroptosis[27]" with "necroptosis [27]" (page 2).
Answer: We have completed revision based on your suggestion.
- Please change "key residues of cysteine" to "key cysteine residues" (page 2).
Answer: We have completed revision based on your suggestion.
- Please replace "adhesion patch kinase" with "focal adhesion kinase" (page 3).
Answer: We have completed revision based on your suggestion.
- The catalog numbers "#8121526" and "#1798320" mentioned in "Scabertopin and deoxyelephantopin was purchased from Chengdu Pusi Biotechnology Co., Ltd. DMEM basal medium (#8121526) and trypsin solution (#1798320) were purchased from Gibco" (page 3) do not seem to be correct as they seem to represent lot numbers. Please fix.
Answer: We have checked the catalog number and made the following changes:
Scabertopin (PS1863-0010) and deoxyelephantopin (PU0338-0010) were purchased from Chengdu Push Biotechnology Co., Ltd (Chengdu, China). DMEM basal medium (10569010) and trypsinEDTA (25200056) were purchased from Gibco (Grand Island, NY, USA).
- Please change "Scabertopin and deoxyelephantopin was" to "Scabertopin and deoxyelephantopin were" (page 3).
Answer: We have completed revision based on your suggestion above.
- Please double-check that the correct company name is "Chengdu Pusi Biotechnology" and not "Chengdu Push Biotechnology" mentioned in "Scabertopin and deoxyelephantopin was purchased from Chengdu Pusi Biotechnology Co., Ltd. DMEM basal medium (#8121526) and trypsin solution (#1798320) were purchased from Gibco" (page 3) and "Scabertopin was purchased from Chengdu Pusi Biotechnology Co.,Ltd. Scabertopin was dissolved in 1ml DMSO and configured into a master batch with a concentration of 10mg/ml (40 μM/L)" (page 3).
Answer: We have corrected the company name as "Chengdu Push Biotechnology".
- Please provide city and state name for the headquarters of "Chengdu Pusi Biotechnology Co., Ltd." (page 3), "Gibco" (page 3), Procell (page 3), "BI Company" (page 3), "Dojindo Corporation" (page 3), "Invitrogen" (page 3), "KeyGen BioTECH" (page 3), "Beyotime Biotechnology Company" (page 3), "Abcam" (page 3), "Cell Signaling" (page 3), "Beijing Labtech Instrument Co., Ltd." (page 3), "BD" (page 6).
Answer: We have provided the cities and countries where the above companies are located.
Chengdu Push Biotechnology Co., Ltd (Chengdu, China);
Gibco (Grand Island, NY, USA);
Procell Life Science&Technology Co.,Ltd(Wuhan, China);
Biological Industries (BI) (Kibbutz Beit Haemek, Israel);
Dojindo Corporation (Kumamoto, Japan);
Invitrogen (Carlsbad, CA, USA);
KeyGen BioTECH (Shanghai, China);
Beyotime Biotechnology (Shanghai, China);
Abcam (Cambridge, MA, USA);
Cell Signaling Technology (Danvers, MA, USA);
Beijing LabTech Instrument Co., Ltd.) (Beijing, China);
BD (Becton, Dickinson and Company, Franklin Lakes, NJ, USA).
- Please replace "#8121526" with "8121526" (page 3).
Answer: We removed “#” and revised the catalog as 105690010.
- Please change "#1798320" to "1798320" (page 3).
Answer: We removed “#” and revised the catalog as 25200056.
- Please replace "#PM150110" with "PM150110" (page 3).
Answer: We have completed revision based on your suggestion.
25) Please change "#1907301" to "1907301" (page 3).
Answer: We have completed revision based on your suggestion.
26) Please define abbreviation for "BI" (page 3), "FTIR" (page 3), "DHE" (page 5), "GSH" (page 5), "GSSG" (page 5), "PI" (page 6), "TPEB" (page 6), "BCA" (page 6).
Answer: We apologize for neglecting to provide the full names of these abbreviations. We have provided these full names in the manuscript.
BI (Biological Industries);
FTIR (fourier transform infrared);
DHE (Dihydroethidium);
BCA (bicinchoninic acid);
GSH (reduced glutathione);
GSSG (oxidized glutathione disulfide);
PI (Propidium Iodide);
TPEB (Total Protein Extraction Kit);
BCA (bicinchoninic acid).
- Please replace "BI Company" with "BI" (page 3).
Answer: We have completed revision based on your suggestion.
- The catalog number "#SA616" mentioned in "Cell Counting Kit-8 (#SA616) was purchased from Dojindo Corporation" does not seem to be correct (page 5). Please fix.
Answer: We have revised this mistake as “Cell Counting Kit-8 (CK04) was purchased from Dojindo Laboratories (Kumamoto, Japan).”
- Please change "#SA616" to "SA616" (page 3).
Answer: We have completed revision as above based on your suggestion.
- Please replace "Dojindo Corporation" with "Dojindo Laboratories" or "Dojindo Molecular Technologies" (page 3).
Answer: We have completed revision as above based on your suggestion.
- The catalog number "#1915835" mentioned in "LIVE/DEAD cell viability assay kit (#1915835) was purchased from Invitrogen" does not seem to be correct (page 5). Please fix.
Answer: We removed “#” and revised the catalog as L3224.
- Please change "#1915835" to "1915835" (page 3).
Answer: We have completed revision as above based on your suggestion.
- The catalog numbers "#20201111" and "#20210114" mentioned in "The cell cycle assay kit (#20201111) and the apoptosis assay kit (#20210114) were purchased from KeyGen BioTECH" do not seem to be correct (page 5). Please fix.
Answer: We removed “#” and revised the catalog as “The cell cycle assay kit (KGA512) and the apoptosis assay kit (KGA1018) were purchased from KeyGen BioTECH (Shanghai, China).”
- Please replace "#20201111" with "20201111" (page 3).
- Please change "#20210114" to "20210114" (page 3).
Answer: We have completed revision as above based on your suggestion.
- Please replace "#S0033S" with "S0033S" (page 3).
Answer: We have completed revision based on your suggestion.
- Please change "Beyotime Biotechnology Company" to "Beyotime Biotechnology" (page 3).
Answer: We have completed revision based on your suggestion.
38) Please replace "BCL-2-associated" with "Bcl-2-associated" (page 3).
Answer: We have completed revision based on your suggestion.
- Please change "(AKT1-3, ab179463)" to "(AKT1-3, ab179463)," (page 3).
Answer: Since the change of superscript/subscript cannot be displayed on the review comments page, we speculate that you suggest that we cancel the original 1-3 subscript format as “AKT1-3”, and we currently change it to “(AKT1-3, ab179463),”. If we misunderstand your meaning, please let us know.
- Please replace "#71433" with "71433" (page 3).
Answer: We have completed revision based on your suggestion.
- Please replace "Phospho-FAK(Y397, #8556)" with "phospho-FAK (Y397, 8556)" (page 3).
Answer: We have completed revision based on your suggestion.
- Please change "#13667" to "13667" (page 3).
Answer: We have completed revision based on your suggestion.
43) Please replace "#4257S" with "4257S" (page 3).
Answer: We have completed revision based on your suggestion.
- Please change "Caspase-3 (#14220)" to "caspase-3 (14220)" (page 3).
Answer: We have completed revision based on your suggestion.
- Please replace "Caspase-8 (#9746)" with "caspase-8 (9746)" (page 3).
Answer: We have completed revision based on your suggestion.
- Please change "Caspase-9 (#9508)" to "caspase-9 (9508)" (page 3).
Answer: We have completed revision based on your suggestion.
- Please replace "#14993" with "14993" (page 3).
Answer: We have completed revision based on your suggestion.
- Please change "(S358,#91689)" to "(S358, 91689)" (page 3).
Answer: We have completed revision based on your suggestion.
- Please replace "#3493" with "3493" (page 3).
Answer: We have completed revision based on your suggestion.
50) Please change "#65746" to "65746" (page 3).
Answer: We have completed revision based on your suggestion.
51) Please change "RIP3 (#13526) and Phospho-RIP3 (Ser227, #93654)" to "RIP3 (13526), and phospho-RIP3 (Ser227, 93654)" (page 3).
Answer: We have completed revision based on your suggestion.
52) Please replace "Cell Signaling company" with "Cell Signaling Technology" (page 3).
Answer: We have completed revision based on your suggestion.
- Please change "Scabertopin was purchased from Chengdu Pusi Biotechnology Co.,Ltd. Scabertopin" to "Scabertopin" (page 3).
Answer: We have completed revision based on your suggestion.
- Please replace "1ml DMSO" with "1 ml DMSO" (page 3).
Answer: We have completed revision based on your suggestion.
- Please change "10mg/ml (40 μM/L)" to "10 mg/ml (40 μM/l)" (page 3).
Answer: We have updated the unit to μM .
56) Please provide the manufacturer and its city and state headquarters for "potassium bromide pressed tablets" mentioned in "Scabertopin and deoxyelephantopin solution was taken 10 μl, applied evenly on potassium bromide pressed tablets, evaporated and dehydrated by infra-red lamp, and analyzed by FTIR spectroscopy using KBr pressed plates" (page 3).
Answer: We have added the following information: KBr (P116274-100g) was purchased from Aladdin Biochemical Technology Co., Ltd (Shanghai, China).
57) Please replace "Scabertopin and deoxyelephantopin solution was taken 10 μl, applied evenly on potassium bromide pressed tablets" with something like "10 μl of scabertopin and deoxyelephantopin solution was applied evenly on KBr tablets" (page 3).
Answer: We have completed revision based on your suggestion.
58) Please change "KBr pressed plates" to "KBr-pressed plates" (page 3).
Answer: We have completed revision based on your suggestion.
59) Please provide city name for the headquarters of "Nexus" (page 3).
Answer: We have provided the city name for the headquarters of "Nexus" as “Fourier transform infrared tester (Nexus-670 produced by Nexus, Madison, Wisconsin, USA)”.
60) Please replace "infrared spectrum" with "IR spectrum" (page 3).
Answer: We have completed revision based on your suggestion.
61) Please change "400-4000cm-1" to "400–4000 cm-1" (page 3).
Answer: We have completed revision based on your suggestion.
- Please format "-1" in "400–4000cm-1" using superscript (page 3).
Answer: We have modified it in superscript.
63) Please replace "32times" with "32" (page 3).
Answer: We have completed revision based on your suggestion.
64) Please change "4cm-1" to "4 cm-1" (page 3).
Answer: We have completed revision based on your suggestion.
- Please format "-1" in "4cm-1" using superscript (page 3).
Answer: We have modified it in superscript.
66) Please replace "The operation is as follows: grind potassium bromide" with "The operation was as follows: grind KBr" (page 3).
Answer: We have completed revision based on your suggestion.
67) Please change "10mg potassium bromide and 1mg" to "10 mg potassium bromide and 1 mg" (page 3).
Answer: We have completed revision based on your suggestion.
68) Please replace "agate mortar and evenly" with "agate mortar evenly" (page 3).
Answer: We have completed revision based on your suggestion.
69) Please change "Infrared spectroscopy" to "IR spectroscopy" (page 3).
Answer: We have completed revision based on your suggestion.
70) Please replace "(UV9100B from Beijing Labtech Instrument Co., Ltd.), with" with "(UV9100B from Beijing LabTech Instruments Co., Ltd.) with" (page 3).
Answer: We have completed revision based on your suggestion.
71) Please change "200-360nm and a resolution of 2nm" to "200–360 nm and a resolution of 2 nm" (page 3).
Answer: We have completed revision based on your suggestion as“200–360 nm and a resolution of 2 nm.”
72) Please replace "ultraviolet maximum" with "UV maximum" (page 3 2x).
Answer: We have completed revision based on your suggestion in both 2 places.
- Please change "ultraviolet spectrum" to "UV spectrum" (page 3 2x).
Answer: We have completed revision based on your suggestion in both 2 places.
- Please replace "214nm" with "214 nm" (page 3).
Answer: We have completed revision based on your suggestion.
- Please change "α,β-unsaturated carbonyl structure" to "α,β-unsaturated carbonyl moiety" (page 3).
Answer: We have completed revision based on your suggestion.
- Please replace "There is no change in 0h, 24h and 48h" with "There was no change in 0, 24, and 48 h" (page 3).
Answer: We have completed revision based on your suggestion.
- Please change "Rambeau Beer's law: When" to "Beer-Lambert law that when" (page 3).
Answer: We have completed revision based on your suggestion.
78) Please format "2" in "cm2·mol" using superscript (page 4).
Answer: We have completed revision based on your suggestion.
79) Please replace "mol/L" with "mol/l" (page 4).
Answer: We have completed revision based on your suggestion.
80) Please provide catalog number for "FBS" mentioned in "All cell culture media and fetal bovine serum (FBS) were obtained from Gibco" (page 4).
Answer: We have provided the catalog number of “FBS” as “ All cell culture media and fetal bovine serum (FBS) (30044333) were obtained from Gibco.”
81) Please change "Dojindo, #CK04" to "Dojindo Laboratories, CK04" or "Dojindo Molecular Technologies, CK04" (page 4).
Answer: We have completed revision based on your suggestion.
82) Please change "J82 was" to "J82 cells were" (page 4).
Answer: We have completed revision based on your suggestion.
83) Please replace "RT4 was" with "RT4 cells were" (page 4).
Answer: We have completed revision based on your suggestion.
84) Please format "2" in "CO2" using subscript (page 4).
Answer: We have completed revision based on your suggestion.
85) Please change "BIO-RAD" to "Bio-Rad" (page 4).
Answer: We have completed revision based on your suggestion.
- Please provide city name for the headquarters of "BIO-RAD" (page 4).
Answer: We have provided the information of Bio-Rad as “Finally, absorbance was detected at 450 nm by using an enzyme marker (Bio-Rad, Hercules, CA, USA).”
87) Please replace "mean ±standard" with "mean ± standard" (page 4).
Answer: We have completed revision based on your suggestion.
88) Please change "generated by using" to "generated using" (page 4).
Answer: We have completed revision based on your suggestion.
89) Please provide city name for the headquarters of "GraphPad Software" (page 4).
Answer: We have provided the information of GraphPad Software as “Dose–response curves were generated using GraphPad Prism 7 (GraphPad Software, San Diego, CA, USA) software.”
90) Please replace "1 × 104" with "1×104" (page 4).
Answer: We have completed revision based on your suggestion.
91) Please provide the manufacturer and its city and state headquarters for "LIVE/DEAD Cell Vitality Assay Kit" mentioned in "Subsequently, the cells were washed with PBS in accordance with the instructions of the LIVE/DEAD Cell Vitality Assay Kit, and each group was incubated for 30 min at room temperature with an appropriate amount of calcein (AM) and ethidium dimers (EthD-1)" (page 4).
Answer: We have provided the information of LIVE/DEAD Cell Vitality Assay Kit as “the LIVE/DEAD cell vitality assay kit (Invitrogen)”. Since the city and country where Invitrogen's headquarters are located have been introduced previously, it will not be repeated here.
92) It is not clear what the authors mean by "appropriate amount of calcein (AM) and ethidium dimers (EthD-1)" in "Subsequently, the cells were washed with PBS in accordance with the instructions of the LIVE/DEAD Cell Vitality Assay Kit, and each group was incubated for 30 min at room temperature with an appro-priate amount of calcein (AM) and ethidium dimers (EthD-1)" (page 4)?
Answer: We have provided the reagents and the dosage used as follows: “and each group was incubated for 30 min at room temperature with 5 μl of the supplied 4-mM calcein AM stock solution to the 10 ml ethidium dimers (EthD-1). ”
93) Please replace "Cell Vitality Assay Kit" with "cell vitality assay kit" (page 4).
Answer: We have completed revision based on your suggestion.
94) Please change "24-well plate then treated with the drug at the concentration of 10μM" to "24-well plate, then treated with the drug at the concentration of 10 μM" (page 4).
Answer: We have completed revision based on your suggestion.
95) Please replace "After being cultured in the incubator for 24h" with "After being cultured in the incubator for 24 h" (page 4).
Answer: We have completed revision based on your suggestion.
96) Please provide manufacturer and its city headquarters for "SU8100" mentioned in "Then, their morphology was observed by using a scanning electron microscope (SU8100, Japan)" (page 4).
Answer: We corrected the model of the instrument incorrectly quoted here, and provided its company name and the city and country where the headquarters are located. “Then, their morphology was observed by using a scanning electron microscope (Regulus8100, HITACHI, Tokyo, Japan).”
97) Please change "J82 cells were inoculated into six-well" to "J82 cells were inoculated into 6-well" (page 4).
Answer: We have completed revision based on your suggestion.
98) Please replace "2.9. Measurement of ROS" with "2.9. Measurement of reactive oxygen species" (page 5).
Answer: We have completed revision based on your suggestion.
99) Please change "gradient concentrations of scabertopin for 24-48h" to "gradient concentrations of scabertopin for 24–48h" (page 5).
Answer: We have completed revision based on your suggestion.
100) Please replace "cells were incubated with Scabertopin for 2 hours" with "cells were incubated with scabertopin for 2 h" (page 5).
Answer: We have completed revision based on your suggestion.
101) Please provide the molar concentration for NAC mentioned in "In rescue experiments, cells were incubated with Scabertopin for 2 hours with 1:200 ROS scavenger N-acetylcysteine (NAC) (MCE#HY-B0215)" (page 5).
Answer: We have provided the molar concentration for NAC as “In rescue experiments, cells were incubated with Scabertopin for 2 h with 5 mM ROS scavenger N-acetylcysteine (NAC) (HY-B0215, MedChemExpress, Monmouth Junction, NJ, United States).”
102) Please provide city and state name for the headquarters for "MCE" mentioned in "In rescue experiments, cells were incubated with Scabertopin for 2 hours with 1:200 ROS scavenger N-acetylcysteine (NAC) (MCE#HY-B0215)" (page 5).
Answer: We have provided the information as above.
- Please change "MCE#HY-B0215" to "MCE, HY-B0215" (page 5).
Answer: We have completed revision based on your suggestion.
104) "The cells were washed three times with PBS and treated with 10 μM dichlorofluorescein diacetate in situ loading probe for 20–30 min" (page 5) does not seem to be grammatically correct with respect to "in situ loading probe for 20–30 min". Please rephrase.
Answer: We have updated the sentence as follows: “The cells were washed three times with PBS and treated with 10 µM dichlorofluorescein diacetate probe for 20–30 min”.
- Please provide city name for the headquarters of "BiOTek" mentioned in "Subsequently, the cells were washed three times with serum-free medium, and their fluorescence intensity was measured with a fluorescence microplate reader (BiOTek, USA)" (page 5).
Answer: We have provided the information as “Subsequently, the cells were washed three times with serum-free medium, and their fluorescence intensity was measured with a fluorescence microplate reader (BioTek, Vermont, USA)”.
106) Please change "BiOTek" to "BioTek" (page 5).
Answer: We have completed revision based on your suggestion.
- Please replace "at a concentration of 25μM" with "at a concentration of 25 μM" (page 5).
Answer: We have completed revision based on your suggestion.
108) Please change "fresh culture medium was used to washed, and the fluorescence intensity of the cells" to "cells were washed with fresh culture medium, the fluorescence intensity" (page 5).
Answer: We have completed revision based on your suggestion.
109) Please replace "the fluorescence intensity of the cells was measured by a fluorescence microplate reader (BiOTek, USA), and" with "the fluorescence intensity of the cells was measured by a fluorescence microplate reader (BioTek, USA). Cells were" (page 5).
Answer: We have completed revision based on your suggestion.
110) Please change "KeyGEN. #KGAF019" to "KeyGen BioTECH, KGAF019" (page 5).
Answer: We have completed revision based on your suggestion.
111) Please replace "Mitochondrial membrane potential assay kit JC-1 kit" with "Mitochondrial membrane potential assay kit with JC-1" (page 5).
Answer: We have completed revision based on your suggestion.
112) Please replace "Beyotime Biotechnology #C2006" with "Beyotime Biotechnology, C2006" (page 5).
Answer: We have completed revision based on your suggestion.
113) It is not exactly clear what the authors mean by "200× JC-1" in "The cells were treat with gradient concentrations of scabertopin, and then washed with PBS, added to cell culture medium and mixed with 200× JC-1, put into 37°C incubator for 20min" (page 5)?
Answer: “200x” refers to the concentration of JC-1 provided in the kit. We have clarified the sentence as “added to cell culture medium and mixed with the JC-1 solution, incubated at 37°C for 20 min according to the manufacturer’s instructions”.
114) Please change "and then washed with PBS" to "then washed with PBS" or "washed with PBS" (page 5).
Answer: We have completed revision based on your suggestion.
115) Please replace "put into 37°C incubator for 20min" with "incubated at 37 °C for 20 min" (page 5).
Answer: We have completed revision based on your suggestion as above.
116) Please change "The fluorescence intensity of the cells was measured by the fluorescence microplate reader (BIOTek, USA)," to "The fluorescence intensity of the cells was measured by the fluorescence microplate reader (BioTek)" (page 5).
Answer: We have completed revision based on your suggestion.
117) Please change "GSH and GSSG Assay Kit (Beyotime Biotechnology #S0053) is a simple and easy-to-use assay that can detect the content of GSH (reduced glutathione) and GSSG (oxidized glutathione disulfide)" to "GSH (reduced glutathione) and GSSG (oxidized glutathione disulfide) assay kit (Beyotime Biotechnology, S0053) is a simple and easy-to-use assay that can detect the content of GSH and GSSG" (page 5).
Answer: We have completed revision based on your suggestion.
118) Please replace "Total Glutathione (GSH+GSSG)- 2×GSSG" with "total glutathione (GSH + GSSG) - 2×GSSG" (page 5).
Answer: We have completed revision based on your suggestion.
119) Please format the formula "GSH = Total Glutathione (GSH+GSSG)- 2×GSSG" (page 5) using the same style as for the "Rambeau Beer's law" (page 4).
Answer: We have completed revision based on your suggestion. It has been shown as follow:
(formula 2)
- It is not exactly clear what the authors mean by "insert area" in "J82 cells were inoculated at the density of 8 × 105 cells/mL into the insert area in the middle of a dish" (page 5)?
Answer: We have improved our description as “J82 cells were inoculated at the density of 8 × 105 cells/ml in 70 μl, into each insert of the Culture-Insert 2 Well (80206, ibidi, Grafelfing, Germany) ”.
121) Please change "8 × 105 cells/mL" to "8×105 cells/ml" (page 5).
Answer: We have completed revision based on your suggestion.
122) It is not clear what the authors mean by "cells were walled" in "After the cells were walled, the insert was removed with forceps, and the old medium was aspirated off" (page 5)?
Answer: We improved our description as “After the cells were attached, the insert was removed with forceps and the old medium was aspirated off.”
123) Please replace "removed with forceps, and" with "removed with forceps and" (page 5).
Answer: We have completed revision based on your suggestion.
124) Please change "with PBS 1–2 times" to "1–2 times with PBS" (page 5).
Answer: We have completed revision based on your suggestion.
125) Please replace "Olympus, IX71" with "IX71, Olympus" (page 5).
Answer: We have completed revision based on your suggestion.
126) Please provide city name for the headquarters of "Olympus" mentioned in "The cells were washed gently with PBS 1–2 times, treated with different concentrations of the drugs, placed in the incubator for further incubation, and removed at 0 and 24 h for fluorescent inverted microscopy (Olympus, IX71, Japan) to observe whether the peripheral cells had migrated to the central scratch area" (page 5).
Answer: We have provided the information as “fluorescent inverted microscopy (IX71, Olympus, Tokyo, Japan)”
127) Please format the formula "(initial area – area at a certain time point/initial area)" (page 5) using the same style as for the "Rambeau Beer's law" (page 4).
Answer: We have completed revision based on your suggestion. It has been shown as follow:
(formula 3)
128) Please change "that had been treated" to "treated" (page 5).
Answer: We have completed revision based on your suggestion.
129) Please replace "drugs for 24h were collected, and" with "drugs for 24 h were collected and" (page 5).
Answer: We have completed revision based on your suggestion.
- Please change "5×104cells/well" to "5×104cells/well" (page 5).
Answer: We have completed revision based on your suggestion.
- Please replace "Transwell plate, and" with "transwell plate and" (page 5).
Answer: We have completed revision based on your suggestion.
132) Please change "lower chamber was added" to "lower chamber was supplemented" (page 5).
Answer: We have completed revision based on your suggestion.
133) Please replace "0.1%Giemsa staining solution" with "0.1% Giemsa staining solution" (page 5).
Answer: We have completed revision based on your suggestion.
134) Please change "different concentrations of drugs for 24h" to "different concentrations of drugs for 24 h" (page 6).
Answer: We have completed revision based on your suggestion.
135) Please replace "1-2 times with PBS" with "1–2 times with PBS" (page 6).
Answer: We have completed revision based on your suggestion.
136) Please change "ethanol, then" to "ethanol and then" (page 6).
Answer: We have completed revision based on your suggestion.
137) Please replace "cells were added with" with "cells were administered with" (page 6).
Answer: We have completed revision based on your suggestion.
138) Please change "(BD FACSCanto), and" to "(BD FACSCanto) and" (page 6).
Answer: We have completed revision based on your suggestion.
139) It is not clear what the authors mean by "Bing Buffer" in "The precipitates were resuspended again with a small amount of Bing Buffer, mixed with Annexin-V-FITC mixed working solution, incubated for 5 min at room temperature under protection from light, and mixed with PI reagent and PBS" (page 6)?
Answer: We are very sorry about this error. The “bing buffer” here should be “binding buffer”. We have corrected it in the manuscript.
140) Please replace "Bing Buffer" with "bing buffer" (page 6).
Answer: We have corrected it as “binding buffer”.
141) Please change "cytometry, and the" to "cytometry and the" (page 6).
Answer: We have completed revision based on your suggestion.
142) Please specify the composition of the "TPEB buffer" and the "protease inhibitor" and "phosphatase inhibitor" mixtures mentioned in "The cells were mixed with TPEB lysis buffer (1% protease inhibitor and 1% phospha-tase inhibitor mixture) on ice after 24 h of scabertopin treatment" (page 6).
Answer: We have described this in more detail in the manuscript as “Add 10 μl ProteinSafe™ Phosphatase Inhibitor Cocktail (DI201, TransGen Biotech, Beijing, China) and ProteinSafe™ Protease Inhibitor Cocktail (DI101, TransGen Biotech) into every 1ml ProteinExt® Mammalian Total Protein Extraction Kit (TPEB, DE101, TransGen Biotech). ”
143) Please provide the manufacturer and its city and state headquarters for "BCA kit" mentioned in "The lysates were collected and centrifuged at 14,000×g for 10 min at 4 °C. Protein concentrations were analyzed by using a BCA kit" (page 6).
Answer: We have provided the information as “Protein concentrations were analyzed by using a bicinchoninic acid (BCA) kit (P0010, Beyotime Biotechnology).”
144) Please provide the manufacturer and its city and state headquarters for "10% precast gel" mentioned in "Equal amounts of protein samples were separated through electrophoresis on a 10% precast gel" (page 6).
Answer: We have provided the information as “10% precast gel (M00664, GenScript Biotech, Nanjing, China).”
145) Please replace "different primary antibodies, including" with "different primary antibodies including" (page 6).
Answer: We have completed revision based on your suggestion.
146) Please change "(dilution ratio: 1/10000), Bcl-2 (dilution ratio: 1/10000), Caspase-9 (dilution ratio: 1/1000), Caspase-3 (dilution ratio: 1/1000), Caspase-8 (dilution ratio: 1/1000), GAPDH ( dilution ratio: 1/10000), MMP-9 (dilution ratio: 1/500), PI3K (dilution ratio: 1/1000), p-PI3K (dilution ratio: 1/1000), AKT(dilution ratio: 1/1000), p-AKT(dilution ratio: 1/1000), glutathione peroxidase 4 (GPX4, dilution ratio: 1/1000), Gasdermin-D (GSDMD, dilution ratio: 1/1000), RIP (dilution ratio: 1/1000), p-RIP (dilution ratio: 1/1000), MLKL (dilution ratio: 1/1000), p-MLKL (dilution ratio: 1/1000), RIP3 (dilution ratio: 1/1000), and p-RIP3 (dilution ratio: 1/1000)" to "(1/10000), Bcl-2 (1/10000), caspase-9 (1/1000), caspase-3 (1/1000), caspase-8 (1/1000), GAPDH (1/10000), MMP-9 (1/500), PI3K (1/1000), p-PI3K (1/1000), AKT(1/1000), p-AKT(1/1000), glutathione peroxidase 4 (GPX4, 1/1000), gasdermin D (GSDMD, 1/1000), RIP (1/1000), p-RIP (1/1000), MLKL (1/1000), p-MLKL (1/1000), RIP3 (1/1000), and p-RIP3 (1/1000)" (page 6).
Answer: Thank you very much for picking this up. We have made the changes according to your suggestions.
147) Please replace "dilution ratio: 1/2000–1/20000" with "1/2000–1/20000" (page 6).
Answer: We have completed revision based on your suggestion.
148) Please provide city name for the headquarters of "MerckMillipore" mentioned in "The film was treated overnight with ECL luminescent solution (MerckMillipore, USA)" (page 6).
Answer: We have provided the information as “ECL luminescent solution (WBKLS0500, Merck Millipore, Darmstadt, Germany).”
- From "The target protein bands were observed and scanned after development and fixation" (page 6) is not clear how the "target protein bands" were scanned?
Answer: We have clarified the statement as: “The target protein lane were imaged by iBright 1500 (Invitrogen) using enhanced chemiluminescent substrates (WBKLS0500, Merck Millipore, Darmstadt, Germany).”
150) Please change "Statistics and bar graphs are" to "Statistics and bar graphs were" (page 6).
Answer: We have completed revision based on your suggestion.
151) Please replace "The chemical structure is shown in Fig.1A" with "The chemical structure of scabertopin is shown in Fig. 1A" (page 6).
Answer: We have completed revision based on your suggestion.
- Please change "structure of the molecule with drug activity, α-methylene-γ-lactone and butenolide" to something like "α-methylene-γ-lactone and butenolide moieties that confer the drug activity" (page 6).
Answer: We have completed revision based on your suggestion.
153) Please replace "The ability of migratory and invasive" to "Migratory and invasive capacity" and "Try397" with "Tyr397" in Figure 6.
Answer: We have completed revision based on your suggestion.
Figure 6. is shown in attachment
154) Please change "Mechanism ofsabertopine" to "Mechanism of scabertopin" (page 13).
Answer: We have completed revision based on your suggestion.
155) Please replace "Verma et al[57]" with "Verma et al. [57]" (page 13).
Answer: We have completed revision based on your suggestion.
Once again, we appreciate your positive comments and valuable inputs. We hope the revised manuscript could meet your standard and be considered for final acceptance for publication in your journal.
Sincerely
Zhenyu Nie,

Round 2
Reviewer 2 Report
Major points:
I) general
1) Scabertopin treatment seems not to have been strictly controlled in Figures 1–3, 5 as "0uM" scabertopin does not account for the effect of the solvent, in which scabertopin was dissolved. Please repeat these experiments using the same volume of control DMSO as used for the parallel dose of scabertopin.
2) Despite the authors claim in Figure 2A,D that scabertopin increased mitochondrial ROS in J82 cells, this conclusion is rather indirect as the probe they used was not specific to assay for mitochondrial-specific ROS. Please either deploy a probe specific for mitochondrial superoxide or demonstrate the effect of a mitochondria-targeted antioxidant. In addition, please note that DHE detects both superoxide and hydrogen peroxide.
3) The authors claim in Figure 2C that NAC treatment can restore scabertopin-induced mitochondrial membrane potential decrease in J82 cells, however it is not clear why the reduction of the green signal is not accompanied by reciprocal increase in the red signal following scabertopin treatment in Figure 2C?
4) The authors show in Figures 2D,E that scabertopin augments ROS production and depletes GSH levels in J82 cells, however the eventuality that this effect is due to the interference of scabertopin with the indicating probe has not been experimentally excluded. Please provide data showing that scabertopin interacts purely with the biological system and not the analytical componentry deployed.
5) On one hand, the authors claim in Figure 3A that the cell spreading area increased following scabertopin treatment, on the other hand, cell spreading area was not increased but rather decreased in Figure 4A.
6) The claim that scabertopin induces necrosis in J82 cells in Figure 3C needs a more thorough support since Annexin V and PI double-staining is not a specific indicator of necrosis but, in fact, labels the pool of late apoptotic, necrotic, and dead J82 cells.
7) Whereas the authors claim that scabertopin-treated J82 cells did not die by apoptosis since no change in pro- and anti-apoptotic proteins have been observed in Figure 3F, this experiment needs to show the comparison between cleaved and uncleaved caspases revealing the bands specific for both entities on the WB.
8) The authors claim in Figure 4C that NAC reverses scabertopin-induced phosphorylation of RIP1, RIP3, and MLKL, however expression of total RIP1, RIP3, and MLKL proteins is missing from that figure thereby precluding to draw such conclusion.
9) Similarly, expression of total FAK, PI3K, Akt is missing from Figure 5D, thereby preventing the conclusion that scabertopin impinges on the FAK/PI3K/Akt signal transduction pathway.
II) presentation quality
There are still graphs and images seen in the manuscript that suffer from low size/resolution. The authors are encouraged to replot the specific figures mentioned below so that the readers can fully appreciate the content of the presented data.
1) Figures 1A; 3B,C,F,G; 4A–C still appear too pixelated for a publication quality standard. Please enlarge and/or increase pixel density.
2) The resolution of Figures 2B–D; 3A,D,E; 4A; 5A,B is poor. Please enlarge and/or increase pixel density so that individual cellular features can be clearly distinguished.
3) The resolution of Western blots presented in Figures 3F,G; 4B,C; 5C,D is poor. Please enlarge and/or increase pixel density so that individual bands can be clearly seen.
Minor points:
1) Reference to Figure 6 seems to be missing in the main text. Please fix.
2) Please incorporate a subchapter dedicated to the chemical reagents used in the Materials and Methods section.
3) Please change "Shufang Zhang 1*" to "Shufang Zhang1*" (page 1).
4) Please replace "Scabertopin is one of the major sesquiterpene lactones from" to "Scabertopin is one of the major sesquiterpene lactones found in" (page 1).
5) Please change "and also inhibit" to "inhibit" (page 1).
6) Please replace "inhibiting the FAK/PI3K-Akt signal" with "inhibiting the FAK/PI3K/Akt signaling" (page 1).
7) Please change "J82, T24, RT4 and 5637" to "J82, T24, RT4, and 5637" (page 1).
8) It is not exactly clear what the authors mean by "druggability in terms of molecular structure" in "In addition, scabertopin also has good druggability in terms of molecular structure" (page 1)?
9) Do the authors actually mean "half-inhibitory concentration (IC50)" instead of "half-inhibition rate (IC50)" mentioned in "At the same time, we also demonstrated that the half-inhibition rate (IC50) of scabertopin on various bladder cancer cell lines (J82, T24, RT4 and 5637) is much lower than that on human ureteral epithelial immortalized cells (SV-HUC-1)" (page 1)? Please revise.
10) Please replace "The above indicated that scabertopin" with "The above observations indicated that scabertopin" (page 1).
11) Please change "cancer that acts by inhibiting metastasis and inducing necroptosis" to "cancer that acts by inducing necroptosis and inhibiting metastasis" (page 1).
12) "Scabertopin is a sesquiterpene lactone compound that is mainly derived from the chemical constituents of Elephantopus scaber L, a herbaceous plant belongs to phylum Angiosperm, class Dicotyledonous, order Campanulaceae, family Asteraceae, genus Elephantopus [8] that is widely distributed worldwide, particularly in East Asia, Southeast Asia, Africa, Australia, India, and South America, and has various pharmacological properties, such as anti-bacterial, anti-diabetic, anti-inflammatory and anti-cancer efficacy" (page 2) is too long. Please split into at least two sentences.
13) Please replace "scabertopin; bladder cancer; reactive oxygen species; necroptosis; Elephantopus scaber L" with "scabertopin; bladder cancer; reactive oxygen species; necroptosis; Elephantopus scaber L." (page 1).
14) Please change "derived from the chemical constituents of Elephantopus scaber L" to "derived from the chemical constituents of Elephantopus scaber L." (page 2).
15) Please replace "can significantly increase ROS levels" with "can significantly increase reactive oxygen species (ROS) levels" (page 2) and "Reactive oxygen species (ROS) have been recently" with "ROS have been recently" (page 2).
16) Please replace "belongs to phylum Angiosperm" with "belonging to phylum Angiosperm" (page 2).
17) Please change "Elephantopus [8] that is widely distributed worldwide" to "Elephantopus [8], which is widely distributed worldwide" (page 2).
18) Please replace "pharmacological properties, such as anti-bacterial" with "pharmacological properties such as anti-bacterial" (page 2).
19) Please change "anti-inflammatory and anti-cancer efficacy" to "anti-inflammatory, and anti-cancer efficacy" (page 2).
20) Please replace "the antitumor activity of Elephantopus scaber L" with "the antitumor activity of Elephantopus scaber L." (page 2).
21) Please change "compound derived from Elephan-topus scaber L" to "compound derived from Elephan-topus scaber L." (page 2).
22) "invasion" appears twice in "In the mouse breast cancer cell line TS/A, DET can inhibit TNF-α-induced NF-κB activity and down-regulate the NF-κB regulated gene products of matrix metalloprotease (MMP)-2 and MMP-9 involved in invasion, thereby inhibiting the migration and invasion in vitro and in vivo" (page 2). Please fix.
23) Please replace "(MMP)-2 and MMP-9 involved in invasion" with "(MMP)-2 and MMP-9 involved in cellular invasion" (page 2).
24) Please change "as well as selectively target tumor" to "as well as they selectively target tumor" (page 2).
25) Please replace "mechanism of scabertopin on bladder cancer" with "mechanism of scabertopin in bladder" (page 2).
26) Please change "unique form of programmed cell death that" to "unique form of programmed cell death, which" (page 2).
27) Please replace "membranes that result in the spillage" with "membranes resulting in the spillage" (page 2).
28) Please change "contents into cells and consequently" to "contents into the surrounding cells and consequently triggering" (page 2).
29) "Receptor-interacting protein (RIP)1, RIP3, and mixed lineage kinase-like (MLKL) are three key factors in the regulation of necroptosis; RIP3 has been well established to recruit MLKL and induce its phosphorylation, and phosphorylated-MLKL (p-MLKL) then undergoes oligomerization and translocates to the plasma membrane to execute cellular necrosis" (page 2) is too long. Please split into two sentences.
30) Please replace "phosphorylation, and phosphorylated-MLKL" with "phosphorylation. Phosphorylated-MLKL" (page 2).
31) Please change "activate necroptosis [27] and are one of the drivers of necroptosis" to "activate and drive necroptosis [27]" (page 2).
32) Please change "and subsequently inducing its" to "and subsequently induces its" (page 2).
33) Please define abbreviation for "RCC" (page 2), "PBS" (page 4), "s.e.m." (page 9), "Q4" (page 10), "Q2" (page 10), "TEM" (page 12).
34) Please replace "that promote increments" with "that promote increases" (page 2).
35) Please change "determined our research object as" to "determined our research objective to study" (page 3).
36) Please replace "cancer cell lines, and explore" with "cancer cell lines and explore" (page 3).
37) Please change "trypsinEDTA" to "trypsin-EDTA" (page 3).
38) Please replace "Science&Technology" with "Science & Technology" (page 3).
39) Please change "Fetal bovine serum (1907301)" to "Fetal bovine serum (FBS) (1907301)" (page 3) and "fetal bovine serum (FBS) (30044333)" to "FBS (30044333)" (page 4).
40) Please provide city name for the headquarters of "Biological Industries" mentioned in "Fetal bovine serum (1907301) was purchased from Biological Industries (BI) (Kibbutz Beit Haemek, Israel)" (page 3).
41) Please change "Haemek" to "HaEmek" (page 3).
42) Please replace "2.2. Material Preparation" with "2.2. Material preparation" (page 3).
43) Please change "(The maximum concentration of DMSO in the cell culture medium is 0.143% )" to "such that the maximum concentration of DMSO in the cell culture medium was 0.143%" (page 3).
44) It is not exactly clear what the authors refer to by "KBr-pressed plates" in "10 μl of scabertopin and deoxyelephantopin solution was applied evenly on KBr tablets, evaporated and dehydrated by infrared lamp, and analyzed by fourier transform infrared (FTIR) spectroscopy using KBr-pressed plates" (page 3)? If these are the same as KBr tablets than please stick with only one terminology.
45) Please replace "10 mg potassium bromide and and 1 mg scabertopin powder are" with "10 mg of KBr and and 1 mg of scabertopin powder were" (page 3).
46) From "IR spectroscopy was performed on the obtained background pieces and samples" (page 3) is not clear what is the difference between "background pieces" and "samples"?
47) Please change "(UV9100B from Beijing LabTech Instrument Co., Ltd.) (Beijing, China)" to "(UV9100B from Beijing LabTech Instrument Co., Ltd, Beijing, China)" (page 3).
48) Please replace "was no change in 0, 24 and 48 h, and it can be known" with "was no change at 0, 24, and 48 h and it can be inferred" (page 4).
49) Please change "light source and the type of the" to "light source, and the type of the" (page 4).
50) Please provide manufacturer and its city and state headquarters for "McCoys’ 5A" and "F-12K" mentioned in "J82 cells were cultured in DMEM (10% FBS) medium, T24 and 5637 cells were cultured in RPMI 1640 (10% FBS), RT4 cells were cultured in McCoys’ 5A (10% FBS) medium, and SV-HUC-1 cells were cultured in F-12K (10% FBS) medium" (page 4).
51) Please replace "McCoys’" with "McCoy’s" (page 4).
52) Please change "of the supplied 4-mM calcein" to "of the supplied 4 mM calcein" (page 4).
53) Please replace "observed by using a scanning" with "observed using a scanning" (page 4).
54) Please provide manufacturer and its city and state headquarters for "paraformaldehyde" mentioned in "The scabertopin-treated cells were fixed with 4% paraformaldehyde at room temperature and dehydrated with alcohol" (page 4).
55) Please change "using a scanning electron microscope" to "using a scanning electron microscope (SEM)" (page 4).
56) Please provide manufacturer and its city and state headquarters for "electron microscope fixative" mentioned in "The cells were treated with an electron microscope fixative for 2–4 h at 4°C, collected, and then centrifuged at low speed until green bean-sized clumps of cells could be seen at the bottom of the tubes" (page 5).
57) Please change "transmission electron microscope, and" to "transmission electron microscope and" (page 5).
58) Please get rid of the blank space on page 5.
59) Please provide manufacturer and its city and state headquarters for "intracellular ROS assay kit" mentioned in "Intracellular ROS levels were measured by using an intracellular ROS assay kit" (page 6).
60) Please change "gradient concentrations of scabertopin for 24-48 h" to "gradient concentrations of scabertopin for 24–48 h" (page 6).
61) Please replace "cells were incubated with Scabertopin" with "cells were incubated with scabertopin" (page 6).
62) Please format "N" in "N-acetylcysteine" using italics (pages 6, 9, 11–13, 15).
63) Please change "HY-B0215, MedChemExpress, Monmouth Junction, NJ, United States" to "MedChemExpress, Monmouth Junction, NJ, USA, HY-B0215" (page 6).
64) Please replace "of scabertopin, and then loaded" with "of scabertopin and then loaded" (page 6).
65) Please provide city and state name for the headquarters of "KeyGEN BioTECH" mentioned in "In order to determine the type of ROS, J82 cells were treated with gradient concentrations of scabertopin, and then loaded with dihydroethidium (DHE) probe (KeyGEN BioTECH, KGAF019) at a concentration of 25 μM and incubated at 37 °C for 60 min in the darkness" (page 6).
66) Please change "fresh culture medium, the" to "fresh culture medium and the" (page 6).
67) Please replace "The cells were treat with gradient concentrations" with "Cells were treated with gradient concentrations" (page 6).
68) Please change "medium and mixed with the JC-1 solution, incubated at 37°C" to "medium, mixed with the JC-1 solution, and incubated at 37 °C" (page 6).
69) Please provide model for the "fluorescence microplate reader" mentioned in "The fluorescence intensity of the cells was measured by the fluorescence microplate reader (BioTek), and the cells were imaged under the inverted fluorescence microscope" (page 6).
70) Please provide model and manufacturer including its city and state headquarters for "inverted fluorescence microscope" mentioned in "The fluorescence intensity of the cells was measured by the fluorescence microplate reader (BioTek), and the cells were imaged under the inverted fluorescence microscope" (page 6).
71) Please replace "(BioTek), and the cells were imaged under the" with "(BioTek) and the cells were imaged under an" (page 6).
72) Please change "Green fluorescence is an indicator of depolarized mitochondria, but" to "Whereas green fluorescence is an indicator of depolarized mitochondria," (page 6).
73) Please replace "according to the following formula 2" with "according to the following formula" (page 6).
74) Please format "GSH=total glutathione (GSH+GSSG) − 2×GSSG (formula 2)" (page 6) using the same font size as for formula 1 (page 4).
75) Please replace "J82 cells were inoculated at the density of 8 × 105 cells/ml in 70 μl, into" with "70 μl of J82 cells were inoculated at the density of 8 × 105 cells/ml into" or "70 μl of J82 cell suspension was inoculated at the density of 8 × 105 cells/ml into" (page 6).
76) Please change "80206, ibidi, Grafelfing, Germany" to "Ibidi, Grafelfing, Germany, 80206" (page 6).
77) Please replace "1-2 times with PBS, treated with different concentrations of the drugs" with "1–2 times with PBS, treated with different concentrations of drugs" (page 6).
78) Please change "area, following formula 3" to "area according to the following formula" (page 6).
79) Please format "initial area−area at a certain time point initial area (formula 3)" (page 6) using the same font size as for formula 1 (page 4).
80) Please leave a gap between "area that had healed at a certain time to the initial area, following formula 3" mentioned in "The percentage of wound healing was analyzed by using ImageJ software and calculated as the ratio of the initial scratch area minus the partially healed area that had healed at a certain time to the initial area, following formula 3" (page 6) and "initial area−area at a certain time point initial area (formula 3)" (page 6).
81) Please replace "transwell plate, and the lower chamber" with "transwell plate and the lower chamber" (page 6).
82) Please specify the instrument used for photographing the images in "The number of migrated cells was recorded by photography under multiple high-magnification fields and by counting the number of migrated cells" (page 7).
83) Please change "RNase/PI (propidium Iodide)" to "RNase/propidium iodide (PI)" (page 7).
84) Please replace "(BD FACSCanto) (Becton, Dickinson and Company, Franklin Lakes, NJ, USA)" with "(FACSCanto, Becton, Dickinson and Company, Franklin Lakes, NJ, USA)" (page 7).
85) Please change "with a small amount of binding Buffer" to "with a small amount of binding buffer" (page 7).
86) Please replace "mixed with Annexin-V-FITC mixed" with "mixed with Annexin-V-FITC" (page 7).
87) Please change "through flow cytometry and the results were analyzed by using" to "by flow cytometry and the results were analyzed using" or "using flow cytometry and the results were analyzed by" (page 7).
88) Please remove italics formatting from "ProteinSafe™", "ProteinSafe", and "ProteinExt" in "Add 10 μl ProteinSafe™ Phosphatase Inhibitor Cocktail (DI201, TransGen Biotech, Beijing, China) and ProteinSafe™ Protease Inhibitor Cocktail (DI101, TransGen Biotech) into every 1ml ProteinExt® Mammalian Total Protein Extraction Kit (TPEB, DE101, TransGen Biotech)" (page 7).
89) Please format "μ" in "Add 10 μl" (page 7) using formatting consistent with the rest of the text.
90) Please replace "Phosphatase Inhibitor Cocktail" with "Phosphatase inhibitor cocktail" (page 7).
91) Please change "Protease Inhibitor Cocktail" to "Protease inhibitor cocktail" (page 7).
92) Please replace "1ml ProteinExt® Mammalian Total Protein Extraction Kit (TPEB, DE101, TransGen Biotech)" with "1 ml of ProteinExt® Mammalian total protein extraction kit (TPEB) (DE101, TransGen Biotech)" (page 7).
93) It is not unequivocally clear what the authors mean by "TPEB reagents" in "The cells were mixed with TPEB reagents on ice after 24 h of scabertopin treatment" (page 7)? Does "TPEB" relate to the mixture of "ProteinSafe™ Phosphatase Inhibitor Cocktail", "ProteinSafe™ Protease Inhibitor Cocktail", and "ProteinExt® Mammalian Total Protein Extraction Kit" or just "ProteinExt® Mammalian Total Protein Extraction Kit"?
94) Please change "by using a bicinchoninic acid" to "by using a Bicinchoninic acid" (page 7).
95) Please replace "1/10000" with "1/10,000" (page 7 3x).
96) Please change "1/1000" to "1/1000" (page 7 18x).
97) Please replace "1/2000–1/20000" with "1/2,000–1/20,000" (page 7).
98) Please change "WBKLS0500, Merck Millipore, Darmstadt, Germany" to "Merck Millipore, Darmstadt, Germany, WBKLS0500" (page 7).
99) Please replace "and the ggplot2 R package.The" with "and the ggplot2 R package. The" (page 7).
100) Please change "The chemical structure is shown in Fig. 1A, the" to "The chemical structure of scabertopin is shown in Fig. 1A. The" (page 8).
101) Please replace "The FTIR spectrum of scabertopin are shown in Fig. 1B. And" with "The FTIR spectrum of scabertopin is shown in Fig. 1B and" (page 8).
102) Please remove the tiny mark that appears above the first "u" in "α,β-unsaturated ester moiety" in Figure 1B.
103) Please change "0h, 24h and 48h are shown in Fig. 1C.According" to "0, 24, and 48 h is shown in Fig. 1C. According" (page 8).
104) Please replace "λmax=214 Abs =0.55" with either "λmax=214 nm, Abs=0.55" or "λmax = 214 nm, Abs = 0.55" in Figure 1C.
105) Please provide error bars for all graphs in Figure 1D.
106) Please replot all graphs in Figure 1D using uniform x- and y-axis title font size.
107) Please replot all graphs in Figure 1D using uniform font size for the caption describing IC50 values.
108) The caption "Cell viability (%) seems to be cut through for 5637 cells both at the 24 and 48 h time points in Figure 1D. Please fix.
109) Please replace "the light source and the type of" with "the light source, and the type of" (page 8).
110) Please change "0 h, 24 h and 48 h were" to "0, 24, and 48 h were" (page 8).
111) Please replace "it meant that the concentration of scabertopin" with "the concentration of scabertopin" (page 8 2x).
112) Whereas SV-HUC-1 cells are referred to as "human ureteral epithelial immortalized cells" in "The viability of scabertopin treated on bladder cancer cells (T24, J82, T24 and 5637) and human ureteral epithelial immortalized cells (SV-HUC-1) was determined by CCK-8 assay" (page 8), these are designated as "human bladder cancer cells" in "The results showed that scabertopin significantly inhibited the viability of the human bladder cancer cells (J82,T24, RT4 and SV-HUC-1) in a dose-dependent manner" (page 8). Please resolve this dichotomy.
113) Please change "viability of scabertopin treated on" to "viability of scabertopin-treated" (page 8).
114) Please replace "T24, J82, T24 and 5637" with "J82, T24, and 5637" (page 8).
115) Please change "was determined by CCK-8 assay" to "was determined by the CCK-8 assay" (page 8).
116) From "The results showed that scabertopin significantly inhibited the viability of the human bladder cancer cells (J82,T24, RT4 and SV-HUC-1) in a dose-dependent manner (Fig. 1D)" is not evident what was the effect of scabertopin in 5637 cells?
117) Please replace "J82,T24, RT4 and SV-HUC-1" with "J82, T24, RT4 and SV-HUC-1" (page 8).
118) From "The 24-hour IC50 of scabertopin for bladder cancer cell lines is approximately 20 μM, and the 48-hour IC50 is even lower" (page 8) is not explicitly clear what was the 48 hour IC50 of scabertopin for bladder cancer cell lines?
119) Please replace "24-hour IC50" with "24 h IC50" (page 8).
120) Please change "and the 48-hour IC50 is even lower" to "while the 48 h IC50 is even lower" (page 8).
121) Please replace "55.84μM" with "55.84 μM, respectively" (page 8).
122) Please change "selected for the follow-up study" to "selected for further study" (page 8).
123) Please replace "Chemical structure, stability and" with "Chemical structure, stability, and" (page 8).
124) Please change "Chemical structure scabertopin, the" to "Chemical structure of scabertopin. The" (page 8).
125) Please format "2" in "vibration of =CH2" using subscript (page 8).
126) Please replace "3080cm-1, and" with "3080 cm-1 and" (page 8).
127) Please format "-1" in "3080cm-1" using superscript (page 8).
128) Please change "1760cm-1" to "1760 cm-1" (page 8).
129) Please format "-1" in "1760cm-1" using superscript (page 8).
130) Please replace "1710cm-1" with "1710 cm-1" (page 8).
131) Please format "-1" in "1710cm-1" using superscript (page 8).
132) Please change "α,β-unsaturated ester, and" to "α,β-unsaturated ester and" (page 8).
133) Please format "-1" in "1650 cm-1" using superscript (page 8).
134) Please replace "0h, 24h and 48h" with "0, 24, and 48 h" (page 8).
135) Please change "0 h, 24 h and 48 h" to "0, 24, and 48 h" (page 8).
136) Please remove "Since the intensity of the UV maximum absorption peaks of scabertopin at 0 h, 24 h and 48 h were all around 0.55, it meant that the concentration of scabertopin did not change" from the legend of Figure 1 due to its duplicity within the main text (page 8).
137) From "24 and 48hours IC50 of bladder cancer cell lines(J82,T24, RT4,5637)and human ureteral epithelial immortalized cells (SV-HUC-1) (n=4)" (page 8) is not clear whether what metric was used to represent the data and whether the experiments were independent?
138) Please replace "24 and 48hours IC50" with "24 and 48 h IC50" (page 8).
139) Please change "lines(J82,T24, RT4,5637)and" to "lines (J82,T24, 5637, and RT4) and" (page 8).
140) Please replace "n=4" with "n = 4" (page 8).
141) Please change "J82 cells treated with Scabertopin." to "J82 cells treated with scabertopin" (page 8).
142) Please replace "treated with scabertopin for 24 and 48 hours" with "treated with scabertopin for 24 and 48 h" (page 9).
143) Please change "probe, and found that scabertopin" to "probe and found that scabertopin" (page 9).
144) Please replace "Fig.2A" with "Fig. 2A" (page 9).
145) It is not clear what is the difference between Figure 2A and 2D? If these are identical experiments, please merge both figures into one. If these are different experiments, please relabel Figure 2D as 2A, Figure 2A as 2B, and Figures 2B,C as 2C,D.
146) Despite the authors claim that "To determine the mechanism of ROS production caused by scabertopin, we used JC-1 assay to observe the changes of ΔΨ in bladder cancer cells treated with scabertopin" (page 9), this experiment seems to have been performed only in J82 cells according to the legend of Figure 2B.
147) Although the authors claim that "The results showed that ΔΨ decreased (red fluorescence increased) after scabertopin treatment (Fig.2B)" (page 9), red fluorescent signal seems to disappear following 10 and 15 uM scabertopin treatment. Please correct.
148) Please change "Fig.2B" to "Fig. 2B" (page 9).
149) Please provide at least one scale bar for Figures 2B,C; 3D,E; 5A,B.
150) Please change "0uM to "0", "10uM" to "10", and "15uM" to "15" in Figures 2B, 3C.
151) Please replace "0uM+NAC" to "0+NAC", "10uM+NAC" to "10+NAC", and "15uM+NAC" to "15+NAC" in Figures 2C, 3C.
152) Please explain the use of "monomer/polymer" in the y-axis title of Figure 2C in the respective figure legend and/or in the Materials and Methods section.
153) Please replace "Rate (Monomer/polymer)" with "Rate (monomer/polymer)" in the y-axis title of Figure 2C.
154) Please change "Concentration of GSH(uM)" to "Concentration of GSH (uM)" or "GSH level (uM)" in the y-axis title of Figure 2E.
155) Please indicate the abbreviation for "ns" in the legend of Figures 2–5.
156) Please indicate the abbreviation for "SA" in the legend of Figures 2–3, 5.
157) From the legend of Figures 2B–E; 3G; 4A–C is not clear how long were J82 cells treated with scabertopin?
158) From the legend of Figures 2C,E; 3C,E; 4C is not clear what concentration of NAC was used to treat J82 cells?
159) From the legend to Figure 2C is not clear what probe was used for imaging?
160) From the legend of Figure 2E; 3A; 4C is not clear what concentration of scabertopin was used to treat J82 cells?
161) The mention of "fluorescence microplate reader" in "Detected by fluorescence microplate reader, the ΔΨ of scabertopin treatment group decreased significantly compared with the control group" (page 9) is rather puzzling since it is not clear whether the same device was used only as part of this measurement or also as part of the ΔΨ measurement mentioned in "The results showed that ΔΨ decreased (red fluorescence increased) after scabertopin treatment (Fig.2B)" (page 9)?
162) "The ROS produced in mitochondria are hydrogen peroxide (H2O2) and superoxide anions (·O2-) in generally, but H2O2 cannot be detected by escaping into the cytoplasm through the inner mitochondrial membrane, superoxide anion produced in mitochondria and transported to the cytoplasm has become a key signaling factor for mitochondrial ROS" is disputable for the two following reasons:
a) Although it is true that H2O2 is produced in mitochondria, this seems to be solely due to its conversion from superoxide, and therefore the claim that mitochondria produce H2O2 is rather misleading. Please either provide reference demonstrating that H2O2 is directly generated withing the mitochondria or rephrase the sentence to take into account this context.
b) Given the short half-life of superoxide, the claim that superoxide is transported to the cytoplasm for signaling requires a specific reference.
c) The claim that "H2O2 cannot be detected by escaping into the cytoplasm through the inner mitochondrial membrane" is debatable since H2O2 seems to be lipid soluble and therefore would be expected to freely permeate phospholipid membranes. In contrast to superoxide, the extramitochondrial signaling function of H2O2 would be more expectedly be promoted by its neutrality and increased stability as a signaling molecule capable of diffusing across longer distances.
163) "The ROS produced in mitochondria are hydrogen peroxide (H2O2) and superoxide anions (·O2-) in generally, but H2O2 cannot be detected by escaping into the cytoplasm through the inner mitochondrial membrane, superoxide anion produced in mitochondria and transported to the cytoplasm has become a key signaling factor for mitochondrial ROS" (page 9) is too long. Please split into two sentences.
164) Please replace "The ROS produced in mitochondria are hydrogen peroxide (H2O2) and superoxide anions (·O2-) in generally" with "In general, ROS produced in mitochondria are hydrogen peroxide (H2O2) and superoxide anion (O2·-)" (page 9).
165) "H2O2 cannot be detected by escaping into the cytoplasm" does not seem to make sense in "The ROS produced in mitochondria are hydrogen peroxide (H2O2) and superoxide anions (·O2-) in generally, but H2O2 cannot be detected by escaping into the cytoplasm through the inner mitochondrial membrane, superoxide anion produced in mitochondria and transported to the cytoplasm has become a key signaling factor for mitochondrial ROS" (page 9). Please rephrase.
166) Please replace "group(Fig.2C)" with "group (Fig. 2D)" (page 9).
167) Please change "reduced glutathione (GSH) is an important antioxidant in cells, and" to "GSH is an important antioxidant in cells and" (page 9).
168) Please replace "Fig.2D" with "Fig. 2E" (page 9).
169) Please change "intracellular ROS and the efficacy" to "intracellular ROS, and the efficacy" (page 9).
170) Please replace "Reduced mitochondrial membrane potential and increased mitochondrial ROS" with "Increased mitochondrial ROS and reduced mitochondrial membrane potential" (page 9).
171) Please change "Scabertopin induced ROS production" to "A. Scabertopin induced ROS production" (page 9).
172) Please replace "cells in a dose-dependent manne" with "cells in a dose-dependent manner" (page 9).
173) Please change "NAC treatment can restore" to "NAC treatment could restore" (page 9).
174) Please change "correlated with scabertopin dose" to "correlated with the scabertopin dose" (page 9).
175) Please replace "could also deplete" with something like "was efficient in depleting" (page 9).
176) Please change "cells and it can be reduced after" to "cells, which could be reversed with" (page 9).
177) Please replace "Data in Fig.2" with "Data" (page 9).
178) Please change "**< 0.01" to "**P < 0.01" (pages 9, 13).
179) Please replace "N-acetylcysteine;" with "N-acetylcysteine" (page 9).
180) Please change "was not apoptosis and ferroptosis." to "was not apoptosis or ferroptosis" (page 9).
181) Please replace "showed that the cells without" with "showed that cells without" (page 9).
182) Please change "Fig3A" to "Fig. 3A" (page 10).
183) Please replace "And then, we analyzed the effect of scabertopin treatment on the potential disruption of cell cycle phases by using flow cytometry to elucidate preliminarily the potential mechanism of scabertopin cytotoxicity in bladder cancer cells" with "To preliminarily elucidate the potential mechanism of scabertopin cytotoxicity in bladder cancer cells, we analyzed the effect of scabertopin treatment on the potential disruption of cell cycle phases by using flow cytometry" (page 10).
184) Please change "(Fig.3B) that scabertopin treatment induced a significant increase in the percentage of cells both in the S and G2/M phase in a concentration-dependent manner" to "that scabertopin treatment induced a significant increase in the percentage of cells both in the S and G2/M phase in a concentration-dependent manner (Fig. 3B)" (page 10).
185) Please remove the caption "bar=50 um" from Figure 3A.
186) The scale bar presented in Figures 3A; 4A is rather confusing as it seems to be dotted.
187) Please either make the ineligible white captions at the bottom of the micrographs presented in Figures 3A; 4A visible in sufficient resolution so that they can be clearly read or remove them completely.
188) Please rescale all y-axis ranges to the same value in the flow cytometry data of Figure 3B so that the data can be directly compared.
189) Please provide y-axis title for the 10 uM scabertopin concentration in the flow cytometry data of Figure 3B.
190) Please replace "FCS Percentage" to "FCS (%)" in the y-axis title in the quantitation graph of Figure 3B.
191) Please provide description for the flow cytometry data in Figure 3B. It is namely not clear what does black trace, the gray-shaded peak, the plain red peaks, and the black arrowheads indicate? Moreover, the black arrowhead seems to be missing from the small plain red peak in the no scabertopin condition. More importantly, provide a clue as to how the cell cycle phases were deduced from the flow cytometry data in the respective figure legend.
192) Please indicate the abbreviation for "PE-A" and "FCS" in the legend of Figure 3B.
193) The claim that "The transfer of the cell membrane phospholipid phosphatidylserine from the inner layer of the plasma membrane to the outer layer of the plasma membrane is one of the early features of apoptosis, and a single positive cell with high affinity for AnnexinV-FITC/PI for PS was considered as a typical cell in early apoptosis, i.e. Q4, whereas AnnexinV-FITC and PI double-positive cells are considered as necrotic cells, i.e. Q2" (page 10) is not correct since cells that stain positive for both FITC Annexin V and PI are, in fact, expected to undergo the end stage of apoptosis, necrosis, or are already dead.
194) It is not exactly clear what the authors mean by "high affinity for AnnexinV-FITC/PI for PS" in "The transfer of the cell membrane phospholipid phosphatidylserine from the inner layer of the plasma membrane to the outer layer of the plasma membrane is one of the early features of apoptosis, and a single positive cell with high affinity for AnnexinV-FITC/PI for PS was considered as a typical cell in early apoptosis, i.e. Q4, whereas AnnexinV-FITC and PI double-positive cells are considered as necrotic cells, i.e. Q2" (page 10)?
195) "The transfer of the cell membrane phospholipid phosphatidylserine from the inner layer of the plasma membrane to the outer layer of the plasma membrane is one of the early features of apoptosis, and a single positive cell with high affinity for AnnexinV-FITC/PI for PS was considered as a typical cell in early apoptosis, i.e. Q4, whereas AnnexinV-FITC and PI double-positive cells are considered as necrotic cells, i.e. Q2" (page 10) is too long. Please split into at least two sentences.
196) Please change "layer of the plasma membrane to the outer layer" to "to the outer layer" (page 10).
197) Please replace "of apoptosis, and a single" with "of apoptosis and a single" (page 10).
198) Please change "double-positive cells are considered" to "double-positive cells were considered" (page 10).
199) Please replace "Fig.3C" with "Fig. 3C" (page 10).
200) It is not exactly clear what the authors mean by "which were similar to the percentages of necrotic cells" in "Under scabertopin treatment, the number of necrotic cells significantly increased in a dose-de-pendent manner, which were similar to the percentages of necrotic cells" (page 10)? Do the authors indeed mean to say that the numbers of necrotic cells were similar to the obtained percentage values?
201) Please change "cells significantly increased in a dose-de-pendent manner, which were similar to the percentages of necrotic cells" to "cells, which were similar to the percentages of necrotic cells, significantly increased in a dose-de-pendent manner" (page 10).
202) The different values plotted for "Q2" in the upper and lower quantitation graphs of Figure 3C is puzzling as one would expect identical cell distribution. Please compile both graphs into one including the effect of NAC.
203) The y-axis "PI PerCP-Cy5-5-A" title seems to be cut through for the 10 uM no NAC datapoint in flow cytometry data of Figure 3C. Please fix.
204) Please replace "Cell Distribution(%)" with "Cell distribution (%)" in the y-axis title in the upper quantitation graph of Figure 3C.
205) Please change "Cell Distribution (%)" to "Cell distribution (%)" in the y-axis title in the lower quantitation graph of Figure 3C.
206) Please indicate the abbreviation for "PerCP-Cy5-5-A" in the legend of Figure 3C.
207) Please replace "Cell viability was measured by using water-soluble tetrazolium salts and green fluorescent calcein-AM" to something like "In addition, water-soluble tetrazolium salts and green fluorescent calcein-AM were deployed" (page 10).
208) From "The results showed that in contrast to the control treatment, scabertopin increased cytotoxicity in a concentration- and time-dependent ways, thus resulting in cell death while significantly reducing cell survival (Fig.3D)" is not exactly clear why the authors are redundantly mentioning that scabertopin reduced cell survival along its capacity to induce cell death?
209) Please change "results showed that in" to "results showed that, in" (pages 10, 11).
210) Please replace "concentration- and time-dependent ways" with "concentration- and time-dependent manner" or "concentration- and time-dependent fashion" (page 10).
211) Please change "Fig.3D" to "Fig. 3D" (page 10).
212) Please replace "24" to "24h" with "48" to "48h" in the y-axis title of the micrographs of Figures 3D,E.
213) Please change "24h" to "24 h" with "48h" to "48 h" in the x-axis title in the quantitation graphs of Figures 3D,E.
214) Please change "Viability(%)" to "Viability (%)" in the quantitation graphs of Figures 3D,E.
215) Please compile the quantitation graphs of Figures 3D,E into one including the demonstration of any effect that NAC might have on cell viability.
216) From the legend of Figures 3D,G is not clear what cells were used for the experiment?
217) Please change "apoptosis-related caspase proteins, Bcl-2" to "apoptosis-related caspase proteins Bcl-2" (page 10).
218) Please replace "GSDMD on the other hand by" with "GSDMD, on the other hand, by" (page 10).
219) Please change "Scabertopin-treated" to "scabertopin-treated" (page 10).
220) Please provide vertical marks for the kDa sizes depicted in the WB of Figures 3F,G; 4B,C; 5C,D.
221) Please align vertically the lane descriptors "0", "10", and "15" in the WB of Figures 3F,G.
222) Please center vertically the y-axis title "Grayscale value" in the quantitation graph of Figures 3F; 4B.
223) Please replace "GasderminD" to "GSDMD" in the x-axis of the quantitation graph of Figure 3G.
224) Please format the y-axis numbers to one decimal point in the quantitation graph of Figures 3G; 4B.
225) Please increase the font size of the y-axis title of Figure 3G to match that of Figure 3F.
226) From the legend of Figures 3F,G; 4B,C; 5C,D is not clear whether grayscale values were normalized to GAPDH?
227) From the legend of Figure 3G is not clear which bands represent GSDMD and what is the lower molecular weight that gradually disappears with increasing concentration of scabertopin? Please label any nonspecific bands using an asterisk.
228) Please replace "die in the manner of apoptosis, ferroptosis" with "die by apoptosis, ferroptosis," (page 10).
229) Please change "area increased, and the cell" to "area increased and the cell" (page 11).
230) Please replace "thinned out; Scabertopin" with "thinned out. Scabertopin" (page 11).
231) Please change "Scabertopin induced cell cycle arrest (B) and necrosis (C) in J82 cells for 24h" to "24 h scabertopin treatment induced cell cycle arrest (B) and necrosis (C) in J82 cells" (page 11).
232) "D. NAC treatment reduces scabertopin-induced death" does not seem to be correct since NAC was not used in Figure 3D.
233) Despite the authors claim that "Live/death cells assay shows scabertopin inhibited the viability of J82 cells in a dose- and time-dependent manner after 24 and 48 h of treatment, and it can be reduced after NAC treatment of cells" (page 11), Figure 3D shows that there was, in fact, no time-dependent effect of scabertopin on cell viability. Please resolve this dichotomy.
234) Please replace "NAC treatment reduces scabertopin-induced death" with "NAC treatment reduced scabertopin-induced cell death" (page 11).
235) Please change "Live/death cells assay shows" to "Live/death cell assay shows that" (page 11).
236) Please replace "and it can be reduced after NAC treatment of cells" with "and this effect could be reduced by NAC treatment" (page 11).
237) Please change "Caspase-9, Caspase-8, Caspase-3" to "caspase-9, caspase-8, caspase-3" (page 11).
238) Please replace "GPX4 and GSDMD" with "GPX4 and GSDMD, respectively" (page 11).
239) Please change "Data in Fig.3" to "Data" (page 11).
240) Please replace "of three independent experiments, *P < 0.05; **P < 0.01; ***P < 0.001" with "of three independent experiments. *P < 0.05, **P < 0.01, ***P < 0.001" (page 11).
241) Please change "Bcl-2, B cell lymphoma-2; Bax, BCL-2-associated X; GSDMD, Gasdermin-D; GPX4, glutathione peroxidase 4; NAC, N-acetylcysteine" to " Bax, Bcl-2-associated X; Bcl-2, B cell lymphoma-2; GPX4, glutathione peroxidase 4; GSDMD, Gasdermin-D; NAC, N-acetylcysteine." (page 11).
242) Please replace "We observed whether cell organelles were altered after scabertopin treatment by per-forming transmission electron microscopy to further determine the manner in which scabertopin induces death in bladder cancer cells" with something like "To further investigate the extent to which cell organelles were altered by scabertopin treatment and to further determine the manner in which scabertopin induces death in bladder cancer cells we performed transmission electron microscopy" (page 11).
243) Please change "the scabertopin-treated cells were swollen" to "scabertopin-treated cells were swollen" (page 12).
244) Please replace "and disappeared organelles and perforation of" with "or absent organelles and perforated" (page 12).
245) Please change "Fig.4A" to "Fig. 4A" (page 12).
246) Please comment on in the text as to why the control J82 cell contains rounded-like blebs at the periphery while the scabertopin-treated cell plasma membrane is rather smooth in Figure 4A.
247) In addition, please comment on in the text as to why the scabertopin-treated J82 cell contains way more organelles/secretory granules that the control cell in the zoomed area of Figure 4A.
248) Lastly, please explain in the text why the control J82 cell nucleus appears fragmented at its bottom left corner?
249) From the zoomed image in Figure 4A is difficult to appreciate that the plasma membrane is without perforations in the control J82 cell as the boundary between the cell and the extracellular space is obscured by rounded-like blebs. Please fix.
250) From the legend of Figures 4A,C; 5C is not clear how many experiments were performed, what metric was used to represent the data, and whether the experiments were independent?
251) Please indicate the abbreviation for "CON" in the legend of Figure 4.
252) Please remove the captions "Bar= 5.0 um" and "Bar= 2.0 um" from Figure 4A.
253) Please replace "western blotting, and the expressions" with "western blotting and the expression" (page 12).
254) Please change "phosphorylated-RIP1, RIP3" to "phosphorylated RIP1, RIP3," (page 12).
255) Please replace "Fig.4B" with "Fig. 4B" (page 12).
256) Please swap the order of the phosphorylated and total proteins presented in the WB of Figures 4B; 5C such that the bands of the phosphorylated version of each protein are right above those of the total protein.
257) Please swap the order of the phosphorylated and total proteins presented in the quantitation graph of Figures 4B; 5C such that the bars corresponding to the phosphorylated version of each protein appear before those of the total protein in the left to right order.
258) From Figures 4B,C is not clear at which amino acid was phosphorylation of RIP1, RIP3, and MLKL detected? Please fix.
259) Please change "using of NAC can significantly inhibit the expression of the p-RIP1, p-RIP3, p-MLKL" to "NAC significantly inhibited the expression of phospho-RIP1, phospho-RIP3, and phospho-MLKL" or to "NAC significantly inhibited the expression of the phosphorylated forms of RIP1, RIP3, and MLKL" (page 12).
260) Please replace "Fig.4C" with "Fig. 4C" (page 12).
261) Please replace "─" with "-" to match the symbol width of "+" in the WB of Figures 4C; 5D.
262) Please increase the font size of the y-axis title of Figure 4C to match that of Figure 4B.
263) Please change "J82 cells undergo necroptosis and" to "J82 cells undergo necroptosis, which" (page 12).
264) Please replace "TEM images of" with "TEM images are shown for" (page 12).
265) Please change "10μM scabertopin (SA) treated J82 cells, the" to "J82 cells treated with 10 μM scabertopin (SA). The" (page 12).
266) Please replace "the left, and the red arrow points out" with "the left and the red arrows point to" (page 12).
267) From "Scabertopin can significantly increase the phosphorylation of the necroptosis-related proteins RIP1, RIP3, and MLKL (n =3)" (page 12) is not clear what metric was used to represent the data and whether the experiments were independent?
268) Please change "(n =3) ;" to "(n = 3);" (page 12).
269) Please replace "C. And it can be reduced after NAC treatment" with "C. Scabertopin-induced phosphorylation of RIP1, RIP3, and MLKL could be reversed by NAC" (page 12).
270) Please change "*P < 0.05; **P < 0.01;" to "*P < 0.05, **P < 0.01," (page 12).
271) Please replace "RIP1, receptor-interacting protein-1; RIP3:receptor-interacting protein-3; p-, phosphorylated. MLKL, mixed lineage kinase like; NAC, N-acetylcysteine;" with "MLKL, mixed lineage kinase like; NAC, N-acetylcysteine; p-, phosphorylated; RIP1, receptor-interacting protein" (page 12).
272) Despite the authors claim that they have performed "Transwell assays" in "We performed wound healing and Transwell assays to characterize how scabertopin affected the migratory and invasive abilities of bladder cancer cells" (page 12) as part of the "3.5. Scabertopin inhibits the migration/invasion of bladder cancer cells" chapter, there is no mention of this assay in this section. Please fix.
273) From "3.5. Scabertopin inhibits the migration/invasion of bladder cancer cells" (page 12) is not unambiguously clear whether the authors mean to say that "Scabertopin inhibits the migration and invasion" or "Scabertopin inhibits the migration or invasion" of bladder cancer cells?
274) Please change "healing and Transwell assays" to "healing and transwell assays" (page 12).
275) Please replace "As shown in the figure, scabertopin inhibited the wound-healing ability of J82 cells in a dose-dependent manner after 24h of treatment (Fig.5A)" with "To this end, 24h treatment with scabertopin inhibited the wound healing ability of J82 cells in a dose-dependent manner (Fig. 5A)" (page 12).
276) Please increase the font size of figure panel descriptors "A", "B", "C" to match that of panel "D" in Figure 5.
277) Please replace "Wound healing percentage" to "Wound healing (%)" in the y-axis title of the quantitation graph of Figure 5A.
278) Please indicate the abbreviation for "CON" in the legend of Figure 5.
279) From the legend of Figure 5A is not clear what do the red lines indicate? In addition, it is not clear what is the difference between the upper and lower panels?
280) Please change "(Fig.5B) that after scabertopin treatment, the invasive ability of cells was significantly reduced" to "that after scabertopin treatment, the invasive ability of cells was significantly reduced (Fig. 5B)" (page 12).
281) It is not exactly clear what the authors mean by "Relative Invaded cell numbers" in the y-axis title of the quantitation graph of Figure 5B. Please rephrase or provide description in the respective figure legend.
282) Please replace "Relative Invaded cell numbers" with "Relative invaded cell numbers" in the y-axis title of the quantitation graph of Figure 5B.
283) Please replace "(FAK, p-FAK, PI3K, p-PI3K, AKT,p-AKT and MMP-9)" with "p-FAK, FAK, p-PI3K, PI3K, p-AKT, AKT, and MMP-9" (page 12).
284) Please change "Western blot analysis (WB)" to "western blot analysis" (page 12).
285) Please replace "p-FAK (Try397), p-PI3K(Y607), p-AKT(S472, S473, S474)" with "phospho-FAK (Tyr397), phospho-PI3K (Tyr607), and phospho-AKT (Ser472, Ser473, Ser474)," (page 12).
286) Please change "in a dose-dependent manner after scabertopin treatment (Fig.5C)" to "after scabertopin treatment in a dose-dependent manner (Fig. 5C)" (page 12).
287) It is not clear what the authors refer to by "AKT1-3" in Figure 5C? Please explain in the respective figure legend.
288) Please replace "p-FAK (Y397)" with "p-FAK (Y397)", "PI3K (Y607)" with "PI3K (Y607)", "AKT1-3" with "Akt1–3", "p-AKT (S472, S473,S474)" with "p-Akt (S472, S473, S474)", "kDA" with "kDa" 8x in Figure 5C.
289) Please horizontally align "FAK", "p-FAK (Y397)", "PI3K p85"," p-PI3K (Y607)", "AKT1-3", and "MMP-9" captions closer to the WB panels in Figure 5C.
290) Please change "p-FAK (Y397)", "p-AKT (S472, S473,S474)" to "p-Akt (S472, S473, S474)", "kDA" to "kDa" 5x in Figure 5D.
291) Please horizontally align "SA" and "NAC" closer to the WB panels in Figure 5D.
292) Please replace "scabertopin can down-regulate" with "scabertopin can downregulate" (page 12).
293) Please change "activation of FAK/PI3K-AKT signaling pathway, and" to "activation of the FAK/PI3K/Akt signaling axis and" (page 12).
294) Please replace "p-FAK, p-PI3K, p-Akt" with "phospho-FAK, phospho-PI3K, phospho-Akt," (page 12).
295) Please change "inhibited after treatment with NAC (Fig.5D)" to "inhibited by NAC (Fig. 5D)" (page 12).
296) Please increase the font size of all captions of Figure 5D to match that of Figure 5C.
297) Please replace "Scabertopin promoting the invasion" with "scabertopin-mediated invasion" (page 12).
298) Please change "migratory and invasive abilities" to "migratory and invasive ability" (page 13).
299) Please replace "FAK /PI3K-Akt /MMP-9" with "FAK/PI3K/Akt/MMP-9" (page 13).
300) From "Scabertopin significantly inhibited the migration ability (n =3) (A) and invasion ability (n =5) (B) of J82 cells treated with gradient concentrations of scabertopin for 24 h" (page 13) is not clear what metric was used to represent the data and whether the experiments were independent?
301) Please change "ability (n =3)" to "(n = 3)" (page 13 2x).
302) Please replace "ability (n =5) (B)" with "(n = 5) (B) ability" (page 13).
303) From "C. p-FAK, p-PI3K, p-AKT and MMP-9 expression levels significantly decreased after 24 h of scabertopin treatment" (page 13) is not explicitly clear which cells were used for the experiment?
304) Please change "p-FAK, p-PI3K, p-AKT" to "phospho-FAK, phospho-PI3K, phospho-Akt," (page 13).
305) From "And it can be reduced after NAC treatment. Data are presented as the mean ± s.e.m. of three independent experiments (n =3)" (page 13) is not clear what metric was used to represent the data and whether the experiments were independent?
306) Please change "D. And it can be reduced after NAC treatment" to "D. Scabertopin-induced phosphorylation of p-FAK, p-PI3K, p-Akt and MMP-9 expression could be reversed by NAC" (page 13).
307) Please replace "AKT, Akt-protein kinase B; p-AKT, phosphorylated Akt-protein kinase B; FAK, focal adhesion kinase; p-FAK, phosphorylated FAK; MMP-9, matrix metalloprotease-9; PI3K, phosphoinositide 3-kinase; p-PI3K, phosphorylated phosphoinositide 3-kinase; NAC, N-acetylcysteine" with "AKT, Akt-protein kinase B; FAK, focal adhesion kinase; MMP-9, matrix metalloprotease-9; NAC, N-acetylcysteine; p-, phosphorylated; PI3K, phosphoinositide 3-kinase" (page 13).
308) Please change "tumors.Natural" to "tumors. Natural" (page 13).
309) Please replace "treated with scabertopin had" with "treated with scabertopin displayed" (page 13).
310) Please change "superoxide anion, a type of ROS" to "superoxide anion production" (page 13).
311) Please replace "However, the further" with "However, further" (page 13).
312) Please change "cell death[33,34]" to "cell death [33,34]" (page 13).
313) Please replace "For example: Isoalantolactone" with "For example, isoalantolactone" (page 13).
314) Please change "arrests cells in the S phase, thereby inhibiting cell proliferation and inducing apoptosis[35]" to "arrests these cells in the S phase, thereby inhibits cell proliferation and induces apoptosis [35]" (page 13).
315) Please replace "migration of breast cancer cells[37]" with "migration capacity of breast cancer cells [37]" (page 14).
316) Please change "Try397 is the major" to "Tyr397 is the major" (page 14).
317) Please replace "site of FAK, and p-FAK" with "site of FAK and phosphorylation of FAK" (page 14).
318) Please change "migration and invasion[38]" to "migration and invasion [38]" (page 14).
319) Please replace "FAK/PI3K-AKT signaling pathway, and" with "FAK/PI3K/Akt signaling pathway and" (page 14).
320) Please change "invasive and metastatic[39]" to "invasive and metastatic [39]" (page 14).
321) Please replace "pathway of FAK/PI3K-Akt/MMP-9" with "FAK/PI3K/Akt/MMP-9 axis" (page 14).
322) Please change "anion-dominated mitochondrial ROS (Fig2)" to "anion-dominated mitochondrial ROS (Fig. 2)" (page 14).
323) From "However, in this study, although we found an increase in ROS, ferroptosis-related signaling molecules did not respond (Fig3)" (page 14) is not explicitly clear to what stimuli "ferroptosis-related signaling molecules" failed to respond?
324) "In fact, ROS is a class of molecules including peroxides, superoxides, singlet oxygen, and free radicals" (page 14) seems to share the same meaning with "They include peroxides, superoxides, singlet oxygen, and free radicals. ROS levels are higher in different types of cancer cells than in normal cells." (page 13). Please remove this redundancy.
325) Please replace "did not respond (Fig3)" with "did not respond (Fig. 3)" (page 14).
326) Please change "RNA and lipid molecules[41]During" to "RNA, and lipid molecules [41]. During" (page 14).
327) Please replace "ferroptosis[42]" with "ferroptosis [42]" (page 14).
328) Please change "arachidonoyl(AA)- phosphatidylethanolamine-(PE) and adrenoyl(AdA)-PE(PE)" to "arachidonoyl (AA)-phosphatidylethanolamine (PE) and adrenoyl (AdA)-PE" (page 14).
329) Please replace "[43].But this" with "[43]. But this" (page 14).
330) Please change "lipid peroxides[44]" to "lipid peroxides [44]" (page 14).
331) Please replace "rupture and perforation" with "rupture, and perforation" (page 14).
332) Please change "cause various cell death" to "induce different forms of cell death" (page 14).
333) It is not exactly clear what the authors mean by "damage of mitochondrial" and "reduction of MMPs" in "Although the mechanism of ROS in necroptosis is not fully understood, there is a lot of evidence that ROS plays a crucial role in many drugs-induced necroptosis [50,51], and is accompanied by damage of mitochondrial and reduction of MMPs[52]" (page 14)?
334) Please replace "in many drugs-induced necroptosis [50,51], and" with "in drug-induced necroptosis [50,51], which" (page 14).
335) Please change "MMPs[52]" to "MMPs [52]" (page 14).
336) Please replace "in turn, forming" with "via" (page 14).
337) It is not exactly clear what the authors mean by "anti-apoptotic cells" in "Because the occurrence of necroptosis not only bypasses apoptosis [55,56], but also causes death in anti-apoptotic cells" (page 15)?
338) Please change "we observed, such as" to "we observed such as" (page 15).
339) The authors claim that "Flow cytometry showed that the treatment of scabertopin could increase the number of late apoptotic cells in the Q2 region" (page 15), however this region is specific for the mixture of late apoptotic, necrotic, and dead J82 cells. Please revise.
340) Please replace "apoptosis-related Caspase protein, Bcl-2 and Bax had" with "apoptosis-related caspase proteins, Bcl-2, and Bax displayed" (page 15).
341) Please change "was the same as ferroptosis" to "was the same as for ferroptosis" (page 15).
342) Please replace "p-RIP1, p-RIP3, and p-MLKL were significantly up-regulated" with "phospho-RIP1, phospho-RIP3, and phospho-MLKL were significantly upregulated" (page 15).
343) Please change "RIP1/RIP3/MLKL, and finally causing" to "RIP1/RIP3/MLKL and finally triggering" (page 15).
344) Please replace "activating RIP /RIP3 /MLKL phosphorylation, mediating J82 cell necroptosis, and inhibiting FAK /PI3K-Akt /MMP-9" with "activate RIP1/RIP3/MLKL phosphorylation, mediate J82 cell necroptosis, and inhibit the FAK/PI3K/Akt/MMP-9" (page 15).
345) Please change "This in turn inhibited the migration and invasion" to "This in turn inhibits the migration and invasion potential" (page 15).
346) Please replace "cells.AKT, Akt-protein kinase B; p-AKT, phosphorylated Akt-protein kinase B; FAK, focal adhesion kinase; p-FAK, phosphorylated FAK; MMP-9, matrix metalloprotease-9; PI3K, phosphoinositide 3-kinase; p-PI3K, phosphorylated phosphoinositide 3-kinase; RIP, receptor-interacting protein ; p-, phosphorylated. MLKL, mixed lineage kinase like; NAC, N-acetylcysteine;" with "cells. Akt, Akt-protein kinase B; FAK, focal adhesion kinase; MLKL, mixed lineage kinase like; MMP-9, matrix metalloprotease 9; NAC, N-acetylcysteine; PI3K, phosphoinositide 3-kinase; p-, phosphorylated; RIP, receptor-interacting protein" (page 15).
347) Please change "In fact, some previous" to "In fact, previous" (page 15).
348) Please replace "[57]. found that isodeoxyelephantopin and deoxyelephantopin can inhibit the activation of NF-κB by inducing the production of ROS and inhibit the growth of breast cancer" with "found that isodeoxyelephantopin and deoxyelephantopin can inhibit the activation of NF-κB by inducing the production of ROS and inhibit the growth of breast cancer [57]" (page 15).
349) Please replace "mitochondrial damage and apoptosis" with "mitochondrial damage, and apoptosis" (page 15).
350) Please change "generation may come from" to "generation may stem from" (page 15).
351) Please replace "Fig.1A" with "Fig. 1A" (page 15).
352) Please change "voltage-dependent anion channels (VDAC) on mitochondria, resulting" to "mitochondrial voltage-dependent anion channel resulting" (page 15).
353) Please replace "through a certain role and" with "and" (page 16).
354) Please change "not only has the possible conditions to trigger this mechanism in structure, but" to something like "not only has the capacity to trigger this mechanism owing to its unique structure but" (page 16).
355) Please replace "Lipinski et al. [64] proposed that most of the compounds that can be druggable should satisfy the molecular weight ≤500, LogP≤5, hydrogen bond donor count =0, hydrogen bond acceptor count =6, rotatable bond count =3 (PubChem CID: 93959111), and there is no evidence that SA is carcinogenic or genotoxic at present, which further supports that scabertopin can be a good drug candidate" with "To this end, Lipinski et al. proposed that most of druggable compounds should satisfy the condition of molecular weight ≤ 500, LogP ≤ 5, hydrogen bond donor count = 0, hydrogen bond acceptor count = 6, rotatable bond count = 3 (PubChem CID: 93959111), and there is no evidence that SA is carcinogenic or genotoxic at present, which further supports that scabertopin can be a promising drug candidate [64]" (page 16).
356) It is not clear what the authors mean by "LogP", " hydrogen bond donor count", "hydrogen bond acceptor count", and "rotatable bond count" in "Lipinski et al. [64] proposed that most of the compounds that can be druggable should satisfy the molecular weight ≤500, LogP≤5, hydrogen bond donor count =0, hydrogen bond acceptor count =6, rotatable bond count =3 (PubChem CID: 93959111), and there is no evidence that SA is carcinogenic or genotoxic at present, which further supports that scabertopin can be a good drug candidate" (page 16)?
357) From "Lipinski et al. [64] proposed that most of the compounds that can be druggable should satisfy the molecular weight ≤500, LogP≤5, hydrogen bond donor count =0, hydrogen bond acceptor count =6, rotatable bond count =3 (PubChem CID: 93959111), and there is no evidence that SA is carcinogenic or genotoxic at present, which further supports that scabertopin can be a good drug candidate" (page 16) is also not clear what is the unit for the molecular weight limit and why the authors are mentioning "PubChem CID: 93959111"?
358) Lastly, from "Lipinski et al. [64] proposed that most of the compounds that can be druggable should satisfy the molecular weight ≤500, LogP≤5, hydrogen bond donor count =0, hydrogen bond acceptor count =6, rotatable bond count =3 (PubChem CID: 93959111), and there is no evidence that SA is carcinogenic or genotoxic at present, which further supports that scabertopin can be a good drug candidate" (page 16) is also not clear against which disease scabertopin represents a "good drug candidate"?
359) "Lipinski et al. [64] proposed that most of the compounds that can be druggable should satisfy the molecular weight ≤500, LogP≤5, hydrogen bond donor count =0, hydrogen bond acceptor count =6, rotatable bond count =3 (PubChem CID: 93959111), and there is no evidence that SA is carcinogenic or genotoxic at present, which further supports that scabertopin can be a good drug candidate" (page 16) is too long. Please split into two sentences.
360) Please change "likely to be associated to the decreased of" to "likely to be associated with decreased" (page 16).
361) Please replace "[67]NAC-treated" with "[67]. NAC-treated" (page 16).
362) Please change "function and reduce ROS" to "function, and reduce ROS" (page 16).
363) Please replace "FAK/ PI3K-AK/ MMP-9" with "the FAK/PI3K/Akt/MMP-9 axis" (page 16).
364) Please change "It can mediate the" to something like "Furthermore, scabertopin can mediate" (page 16).
365) Please replace "RIP1/ RIP3/ MLKL" with "RIP1/RIP3/MLKL" (page 16).
366) Please change "phosphorylation, and inhibit" to "phosphorylation and inhibit" (page 16).
367) Please replace "ROS scavenger, NAC, can" with "the ROS scavenger NAC can" (page 16).
368) Please change "scabertopin, but also" to "scabertopin but also" (page 16).
369) Please replace "inhibited the abilities of invasion and migration" with "limit their invasive and migratory ability" (page 16).
370) Please change "the above two" to "the two abovementioned" (page 16).
371) Please replace "Our study provided" with "In conclusion, our study provides" (page 16).
372) Please change "and scabertopin may be" to "suggesting that scabertopin may be" (page 16).
373) Please replace "YX and DH" with "YX, and DH" (page 16).
374) Please change "ZN, all" to "ZN. All" (page 16).
375) Please replace "(grant numbers 820QN423, 821QN424 and ZDYF2021SHFZ096).the" with "(grant numbers 820QN423, 821QN424, and ZDYF2021SHFZ096), the" (page 16).
376) Please change "(grant No. 82160531).Health" to "(grant number 82160531), and the Health" (page 16).
377) Please replace "grant No.21A200412" with "grant numbers 21A200412" (page 16).
Author Response
Responding
Dear review 2:
Hello!
Thank you for reviewing our manuscript in great detail again. The questions and suggestions you put forward are very valuable. All the authors read your comments together and made revisions. Based on your suggestions, we have made a lot of optimization and revisions to the text in the manuscript. Similarly, we have adjusted and optimized the quality and content of all Figures. These adjustments make the meaning expressed in the text and figures more direct and rigorous. All the authors thank you very much for your work and efforts in this manuscript.
However, your comments are also full of adjustments on punctuation, spaces and other formats. These errors may come from our mistakes in writing the manuscript, for which we are very sorry. Some of these errors may come from the adjustment of the specific format of the journal. However, we believe that these amendments can be made after the manuscript is accepted by the Cancers, and the proofreader can contact us and cooperate with us to make revise. Such a large amount of repetitive work should not occupy your time, which will make us feel very guilty and regret.
In general, we have modified and answered all your comments one by one. We have carefully revised the manuscript according to your comments: all revisions are shown in the manuscript with tracked changes and highlight in blue color. Of course, we chose to keep the last modification trace. The green highlight is the suggestion of reviewer 1, and the yellow highlight is the suggestion of reviewer 2 in the first round.
Similarly, the original images of our newly added blots have been uploaded to the MDPI system.
At the same time, the responding includes necessary figures (pictures cannot be inserted in this answer box). Therefore, we strongly recommend that you download the Responding file for your review.
Now, we address all of your comments point counterpoint below:
Major points:
- I) general
- Scabertopin treatment seems not to have been strictly controlled in Figures 1–3, 5 as "0uM" scabertopin does not account for the effect of the solvent, in which scabertopin was dissolved. Please repeat these experiments using the same volume of control DMSO as used for the parallel dose of scabertopin.
All the culture medium used in the control group contain 0.143% DMSO to eliminate its interference. We have updated this illustration in the section of Materials and Methods as “Scabertopin (10 mg) was dissolved in 1 ml dimethyl sulfoxide DMSO (Beyotime Biotechnology, ST2335-100ml) and configured into a master batch with a concentration of 40 μM such that the maximum concentration of DMSO in the cell culture medium was 0.143%. In the subsequent experiment, in order to exclude the effect of DMSO on cells, 0.143% DMSO was contained in all negative control group media as control. ”
- Despite the authors claim in Figure 2A,D that scabertopin increased mitochondrial ROS in J82 cells, this conclusion is rather indirect as the probe they used was not specific to assay for mitochondrial-specific ROS. Please either deploy a probe specific for mitochondrial superoxide or demonstrate the effect of a mitochondria-targeted antioxidant. In addition, please note that DHE detects both superoxide and hydrogen peroxide.
In Figure 2A, we showed that ROS increased in J82 cells treated with scarbertopin was detected using ROS probe, and Figure 2B showed that mitochondrial membrane potential (ΔΨ) decreased. In Figure 2D, we specifically detected the increased superoxide anion in J82 cells treated with drugs using DHE probe. Therefore, we believe the conclusion that scarbertopin can increase ROS in J82 cells, especially superoxide anion is direct. In addition, according to the protocol of DHE, it is a specific assay of superoxide anion. For example, in these references (PMID: 29206074, PMID: 28455949, PMID: 17045638, PMID: 27989546, PMID: 29957360, PMID: 29227865, PMID: 18055522, PMID: 35771455), it is clearly pointed out that DHE probe can only specifically bind intracellular superoxide anion. Especially in PMID: 35771455, the author invented a plate reader based measurement to measure cyclosolid superoxide anion based on DHE. Therefore, it is in accordance with the existing theory that we use DHE to specifically detect intracellular superoxide anion.
- The authors claim in Figure 2C that NAC treatment can restore scabertopin-induced mitochondrial membrane potential decrease in J82 cells, however it is not clear why the reduction of the green signal is not accompanied by reciprocal increase in the red signal following scabertopin treatment in Figure 2C?
NAC rescue experiment can be recovered to a certain extent ΔΨ, However, it is not possible to save all mitochondria back to the original level, which also shows that scarbertopin has good abilities to reduction ΔΨ and induce the increase of ROS. So the Fig. 2C should be compared with Fig. 2B together. In general, although NAC can save the reduced ΔΨ, but it cannot be saved to level of 0 μM's scabertopin group. Based on the scientific ethics, we show all the original pictures. Therefore, such a query is inappropriate. Please note that we have re-ordered the images in Figure 2.
4) The authors show in Figures 2D,E that scabertopin augments ROS production and depletes GSH levels in J82 cells, however the eventuality that this effect is due to the interference of scabertopin with the indicating probe has not been experimentally excluded. Please provide data showing that scabertopin interacts purely with the biological system and not the analytical componentry deployed.
We show the stability of scabertopin in the medium for 0, 24 and 48 hours in Fig. 1C. The UV absorption peak shows that scabertopin is stable in the medium and does not react with it. In the GSH probe detection experiment, the experimental results are based on the readings of the fluorescent probe in the microplate reader (Materials& Methods). Therefore, our blank control group is the medium containing scabertopin and GSH probe (without cells). All the readings of the treatment groups are subtracted from the blank group. This is based on the reagent protocol and the basic principles of the experiment, so we did not make too redundant illustrations.
5) On one hand, the authors claim in Figure 3A that the cell spreading area increased following scabertopin treatment, on the other hand, cell spreading area was not increased but rather decreased in Figure 4A.
The images of Fig. 3A are based on scanning electron microscope (SEM), while Fig. 4A is transmission electron microscope. Cell samples should be made into cell slides, and then dehydrated, fixed and dried before being used for SEM observation. SEM images are mainly used to observe the morphological changes at the cellular level. The cell samples used for TEM observation do not need these processes, but only need to be centrifuged and embedded after forming cell clumps, which can maintain the original appearance of cells, so as to observe the changes of subcellular structure in cells. It may be that the differences in sample preparation and observation objectives between them lead to the misunderstanding.
6) The claim that scabertopin induces necrosis in J82 cells in Figure 3C needs a more thorough support since Annexin V and PI double-staining is not a specific indicator of necrosis but, in fact, labels the pool of late apoptotic, necrotic, and dead J82 cells.
We have corrected this wording expression in the manuscript, this issue also occurs many times in minor points, and we have corrected it together.
7) Whereas the authors claim that scabertopin-treated J82 cells did not die by apoptosis since no change in pro- and anti-apoptotic proteins have been observed in Figure 3F, this experiment needs to show the comparison between cleaved and uncleaved caspases revealing the bands specific for both entities on the WB.
The reagents we used “caspase-3 (14220), caspase-8 (9746), caspase-9 (9508)” can detect cleaved and uncleaved caspases simultaneously. We also completed the assays at the beginning, but we thought it was unnecessary to show them because there was no difference between them. We have shown these results in Figure 3F at present, and have re-created the corresponding quantitative graph. Gels images of three independent parallel experiment will also be uploaded to MDPI system.
8) The authors claim in Figure 4C that NAC reverses scabertopin-induced phosphorylation of RIP1, RIP3, and MLKL, however expression of total RIP1, RIP3, and MLKL proteins is missing from that figure thereby precluding to draw such conclusion.
Also, we completed these experiments in the early stage, but did not show these results because of the lack of significant differences. Now we show them and re-create the corresponding quantitative graph. Gels images of three independent parallel experiment will also be uploaded to MDPI system.
9) Similarly, expression of total FAK, PI3K, Akt is missing from Figure 5D, thereby preventing the conclusion that scabertopin impinges on the FAK/PI3K/Akt signal transduction pathway.
Similarly, we completed these experiments in the early stage, but did not show these results at Beginning. Now we show them and re-create the corresponding quantitative graph. Gels images of three independent parallel experiment will also be uploaded to MDPI system.
- II) presentation quality
There are still graphs and images seen in the manuscript that suffer from low size/resolution. The authors are encouraged to replot the specific figures mentioned below so that the readers can fully appreciate the content of the presented data.
1) Figures 1A; 3B,C,F,G; 4A–C still appear too pixelated for a publication quality standard. Please enlarge and/or increase pixel density.
We have re-made all the figure in high quality for the tiff format, We uploaded these pictures to the MDPI system when we uploaded the manuscript, we have made every effort to solve this problem. However, we speculate that the quality of images embedded in reviewed-manuscript may be slightly impaired. Fortunately, since Cancers is a high-quality open access journal, readers will be able to easily download these images from the website. And we believe that the editors of Cancers will also successfully solve this problem.
Figure 1. Chemical structure, stability, and efficacy of scabertopin against bladder cancer cells. A. Chemical structure scabertopin. The two boxes show the structure of the molecule with drug activity, α-methylene-γ-lactone and butenolide; B. Infrared spectrum of scabertopin. The stretching vibration of =CH2 exists at 3080 cm-1, and the carbonyl stretching vibration of α-methylene-γ-lactone structure exists at 1760 cm-1. The peak at 1710 cm-1 is the carbonyl stretching vibration of another non-lactone α,β-unsaturated ester and the C=C stretching vibration exists at 1650 cm-1; C. The UV absorption peak of scabertopin in DMEM medium at 0, 24 and 48h. D. 24 and 48 h IC50 of bladder cancer cell lines(J82, T24, 5637, and RT4)and human ureteral epithelial immortalized cells (SV-HUC-1) (n = 4).
Figure 2. Increased mitochondrial ROS and reduced mitochondrial membrane potential in J82 cells treated with scabertopin. A. Scabertopin induced ROS production in a dose- and time-dependent manner; B. DHE probe assay showed that superoxide anion was positively correlated with the scabertopin dose; C. JC-1 assay showed that after 24 h treatment, scabertopin decreased mitochondrial membrane potential in J82 cells in a dose-dependent manner; D. JC-1 assay showed that NAC treatment could restore the mitochondrial membrane potential reduced by 24 h treatment of scabertopin to some extent; D. DHE probe assay showed that superoxide anion was positively correlated with the scabertopin dose; E. After 24 h treatment with 10μM scabertopin was efficient in depleting GSH through a dose-dependent mechanism in J82 cells, which could be reversed with NAC treatment. 5 μM of NAC was used in the experiments. Data represent the mean ± standard error of mean (s.e.m.) of three independent experiments. *P < 0.05, **P< 0.01, ***P < 0.001, ns: no significance vs. 0 μM scabertopin-treated group. NAC, N-acetylcysteine; SA: scabertopin.
Please note that we re-order the images in figure 2.
Figure 3. Cell death induced by scabertopin treatment can be rescued by NAC. A. SEM images show that in J82 cells treated for 24 h with 10μM scabertopin, the cell spreading area increased and the cell pseudopods shortened and thinned out. 24 h scabertopin treatment induced cell cycle arrest (B) and death (C) in J82 cells in a concentration-dependent manner (the black arrowheads in Fig. 3B indicate the values of the peaks). Live/death cell assay shows scabertopin inhibited the viability of J82 cells after 24 and 48 h of treatment (D), and this effect could be reduced by NAC treatment (E). F. The expression levels of the apoptosis-related proteins caspase-9, caspase-8, caspase-3, Bax, and Bcl-2 in J82 cells did not significantly change after 24 h of scabertopin treatment; G. Scabertopin did not significantly alter the expression of the ferroptosis and pyroptosis-related proteins GPX4 and GSDMD, respectively in J82 cells after 24 h treatment. 5 μM of NAC was used in the experiments, * means nonspecific bands. Data represent the mean ± s.e.m. of three independent experiments. *P < 0.05; **P < 0.01; ***P < 0.001, ns: no significance vs. 0 μM scabertopin-treated group. The grayscale values were normalized to GAPDH. Bax, Bcl-2-associated X; Bcl-2, B cell lymphoma-2; GPX4, glutathione peroxidase 4; GSDMD, Gasdermin-D; NAC, N-acetylcysteine; PE-A, phycoerythrin area; PerCP-Cy5-5-A, Peridinin-Chlorophyll-Protein Complex-Cyanine5.5 area; SA, scabertopin.
Figure 4. Scabertopin-treated J82 cells undergo necroptosis, which can be rescued by NAC. A. TEM images are shown for the control group (CON) and J82 cells treated with 10 μM scabertopin (SA) 24 h. The image on the right is the enlarged image on the left and the red arrow points out the perforation on the cell membrane during necroptosis; at least three independent samples were observed in each group. B. Scabertopin can significantly increase the phosphorylation of the necroptosis-related proteins RIP1, RIP3, and MLKL (n = 3) after 24 h treatment; C. Scabertopin-induced phosphorylation of RIP1, RIP3, and MLKL could be reversed by NAC in the group of 10 μM scabertopin 24 h treatment. 5 μM of NAC was used in the experiments. The grayscale values were normalized to GAPDH. Data represent the mean ± s.e.m. of three independent experiments. *P < 0.05, **P < 0.01, ***P < 0.001, ns: no significance vs. 0 μM scabertopin-treated group. MLKL, mixed lineage kinase like; NAC, N-acetylcysteine; p-, phosphorylated; RIP1, receptor-interacting protein.
Figure 5. Scabertopin treatment inhibited the migratory and invasive ability of J82 cells by inhibiting the FAK/PI3K/Akt/MMP-9 signaling pathway, which was rescued by NAC. Scabertopin significantly inhibited the migration ability (difference in area reduction between red lines) (n = 3) (A) and invasion with (n = 5) (B) of J82 cells treated with gradient concentrations of scabertopin for 24 h; C. phospho-FAK, phospho-PI3K, phospho-Akt and MMP-9 expression levels significantly decreased in J82 cells after 24 h of scabertopin treatment; D. Scabertopin-induced phosphorylation of p-FAK, p-PI3K, p-Akt and MMP-9 expression could be reversed by NAC. 5 μM of NAC was used in the experiments. The grayscale values were normalized to GAPDH. Data are presented as the mean ± s.e.m. of three independent experiments (n = 3). *P < 0.05, **< 0.01, ***P < 0.001, ns: no significance vs. 0 μM scabertopin-treated group. AKT, Akt-protein kinase B; CON, control group; FAK, focal adhesion kinase; MMP-9, matrix metalloprotease-9; NAC, N-acetylcysteine; p-, phosphorylated; PI3K, phosphoinositide 3-kinase; SA, scabertopin.
2) The resolution of Figures 2B–D; 3A,D,E; 4A; 5A,B is poor. Please enlarge and/or increase pixel density so that individual cellular features can be clearly distinguished.
This issue has been solved as above.
3) The resolution of Western blots presented in Figures 3F,G; 4B,C; 5C,D is poor. Please enlarge and/or increase pixel density so that individual bands can be clearly seen.
This issue has been solved as above.
Minor points:
1) Reference to Figure 6 seems to be missing in the main text. Please fix.
It has been added in the page 15.
2)Please incorporate a subchapter dedicated to the chemical reagents used in the Materials and Methods section.
The first subchapter in the section if Materials and Methods is the main reagents of this study. Readers can find the main chemical reagents involved in this study from this subsection. In addition, special chemical reagents in some experiments are described in other subchapter of this section to facilitate readers' access to reagents needed for this experiment.
3) Please change "Shufang Zhang 1*" to "Shufang Zhang1*" (page 1).
Changed.
4) Please replace "Scabertopin is one of the major sesquiterpene lactones from" to "Scabertopin is one of the major sesquiterpene lactones found in" (page 1).
Changed.
5) Please change "and also inhibit" to "inhibit" (page 1).
Changed.
6) Please replace "inhibiting the FAK/PI3K-Akt signal" with "inhibiting the FAK/PI3K/Akt signaling" (page 1).
Replaced.
7) Please change "J82, T24, RT4 and 5637" to "J82, T24, RT4, and 5637" (page 1).
Changed.
8) It is not exactly clear what the authors mean by "druggability in terms of molecular structure" in "In addition, scabertopin also has good druggability in terms of molecular structure" (page 1)?
This sentence is based on the discussion on the druggability of scarbertopin in the discussion section and the stability of the active ingredient of the drug and the culture medium in Figure 1 A-C. But in order to avoid unnecessary misunderstanding, we have deleted this sentence.
9) Do the authors actually mean "half-inhibitory concentration (IC50)" instead of "half-inhibition rate (IC50)" mentioned in "At the same time, we also demonstrated that the half-inhibition rate (IC50) of scabertopin on various bladder cancer cell lines (J82, T24, RT4 and 5637) is much lower than that on human ureteral epithelial immortalized cells (SV-HUC-1)" (page 1)? Please revise.
Revised.
10) Please replace "The above indicated that scabertopin" with "The above observations indicated that scabertopin" (page 1).
Replaced.
11) Please change "cancer that acts by inhibiting metastasis and inducing necroptosis" to "cancer that acts by inducing necroptosis and inhibiting metastasis" (page 1).
Changed.
12) "Scabertopin is a sesquiterpene lactone compound that is mainly derived from the chemical constituents of Elephantopus scaber L, a herbaceous plant belongs to phylum Angiosperm, class Dicotyledonous, order Campanulaceae, family Asteraceae, genus Elephantopus [8] that is widely distributed worldwide, particularly in East Asia, Southeast Asia, Africa, Australia, India, and South America, and has various pharmacological properties, such as anti-bacterial, anti-diabetic, anti-inflammatory and anti-cancer efficacy" (page 2) is too long. Please split into at least two sentences.
It has been split into “Scabertopin is a sesquiterpene lactone compound that is mainly derived from the chemical constituents of Elephantopus scaber L. It is a kind of herbaceous plant belonging to phylum Angiosperm, class Dicotyledonous, order Campanulaceae, family Asteraceae, genus Elephantopus [8], which is widely distributed worldwide, particularly in East Asia, Southeast Asia, Africa, Australia, India, and South America, and has been reported various pharmacological properties such as anti-bacterial, anti-diabetic, anti-inflammatory, and anti-cancer efficacy [9,10].”
13) Please replace "scabertopin; bladder cancer; reactive oxygen species; necroptosis; Elephantopus scaber L" with "scabertopin; bladder cancer; reactive oxygen species; necroptosis; Elephantopus scaber L." (page 1).
Changed.
14) Please change "derived from the chemical constituents of Elephantopus scaber L" to "derived from the chemical constituents of Elephantopus scaber L." (page 2).
Changed.
15) Please replace "can significantly increase ROS levels" with "can significantly increase reactive oxygen species (ROS) levels" (page 2) and "Reactive oxygen species (ROS) have been recently" with "ROS have been recently" (page 2).
Replaced.
16) Please replace "belongs to phylum Angiosperm" with "belonging to phylum Angiosperm" (page 2).
Replaced.
17) Please change "Elephantopus [8] that is widely distributed worldwide" to "Elephantopus [8], which is widely distributed worldwide" (page 2).
Changed.
18) Please replace "pharmacological properties, such as anti-bacterial" with "pharmacological properties such as anti-bacterial" (page 2).
Replaced.
19) Please change "anti-inflammatory and anti-cancer efficacy" to "anti-inflammatory, and anti-cancer efficacy" (page 2).
Changed.
20) Please replace "the antitumor activity of Elephantopus scaber L" with "the antitumor activity of Elephantopus scaber L." (page 2).
Replaced.
21) Please change "compound derived from Elephan-topus scaber L" to "compound derived from Elephan-topus scaber L." (page 2).
Changed.
22) "invasion" appears twice in "In the mouse breast cancer cell line TS/A, DET can inhibit TNF-α-induced NF-κB activity and down-regulate the NF-κB regulated gene products of matrix metalloprotease (MMP)-2 and MMP-9 involved in invasion, thereby inhibiting the migration and invasion in vitro and in vivo" (page 2). Please fix.
It has been fix into “In the mouse breast cancer cell line TS/A, DET can inhibit TNF-α-induced NF-κB activity and down-regulate the NF-κB regulated gene products of matrix metalloprotease (MMP)-2 and MMP-9, thereby inhibiting the migration and invasion in vitro and in vivo [14].”
23) Please replace "(MMP)-2 and MMP-9 involved in invasion" with "(MMP)-2 and MMP-9 involved in cellular invasion" (page 2).
According to the Q22, “involved in invasion” has been deleted.
24) Please change "as well as selectively target tumor" to "as well as they selectively target tumor" (page 2).
Changed.
25) Please replace "mechanism of scabertopin on bladder cancer" with "mechanism of scabertopin in bladder" (page 2).
Replaced
26) Please change "unique form of programmed cell death that" to "unique form of programmed cell death, which" (page 2).
Changed.
27) Please replace "membranes that result in the spillage" with "membranes resulting in the spillage" (page 2).
Replaced
- Please change "contents into cells and consequently" to "contents into the surrounding cells and consequently triggering" (page 2).
Changed
29) "Receptor-interacting protein (RIP)1, RIP3, and mixed lineage kinase-like (MLKL) are three key factors in the regulation of necroptosis; RIP3 has been well established to recruit MLKL and induce its phosphorylation, and phosphorylated-MLKL (p-MLKL) then undergoes oligomerization and translocates to the plasma membrane to execute cellular necrosis" (page 2) is too long. Please split into two sentences.
According to the Q30, it has been split into “ Receptor-interacting protein (RIP)1, RIP3, and mixed lineage kinase-like (MLKL) are three key factors in the regulation of necroptosis; RIP3 has been well established to recruit MLKL and induce its phosphorylation. Phosphorylated-MLKL (p-MLKL) then undergoes oligomerization and translocates to the plasma membrane to execute cellular necrosis [24-26]. ”
30) Please replace "phosphorylation, and phosphorylated-MLKL" with "phosphorylation. Phosphorylated-MLKL" (page 2).
Replaced.
31) Please change "activate necroptosis [27] and are one of the drivers of necroptosis" to "activate and drive necroptosis [27]" (page 2).
Changed.
32) Please change "and subsequently inducing its" to "and subsequently induces its" (page 2).
Changed.
33) Please define abbreviation for "RCC" (page 2), "PBS" (page 4), "s.e.m." (page 9), "Q4" (page 10), "Q2" (page 10), "TEM" (page 12).
We have define the abbreviation as “renal cell carcinoma (RCC)” (page 2), “phosphate buffered saline (PBS)” (page 4), “standard error of mean (s.e.m.) ” (page 9), "the fourth quadrant (Q4)" (page 10), "the second quadrant (Q2)" (page 10), “transmission electron microscopy (TEM) ” (page 5).
34) Please replace "that promote increments" with "that promote increases" (page 2).
35) Please change "determined our research object as" to "determined our research objective to study" (page 3).
Changed.
36) Please replace "cancer cell lines, and explore" with "cancer cell lines and explore" (page 3).
Replaced.
37) Please change "trypsinEDTA" to "trypsin-EDTA" (page 3).
Changed.
38) Please replace "Science&Technology" with "Science & Technology" (page 3).
Replaced.
39) Please change "Fetal bovine serum (1907301)" to "Fetal bovine serum (FBS) (1907301)" (page 3) and "fetal bovine serum (FBS) (30044333)" to "FBS (30044333)" (page 4).
Changed.
40) Please provide city name for the headquarters of "Biological Industries" mentioned in "Fetal bovine serum (1907301) was purchased from Biological Industries (BI) (Kibbutz Beit Haemek, Israel)" (page 3).
“Kibbutz Beit Haemek” is the city name for the headquarters of “Biologicla industries”. (The link is an introduction on the BI official website: https://www.bioind.com/worldwide/about/organization/ )
41) Please change "Haemek" to "HaEmek" (page 3).
According to BI’s official website, it was “Haemek” instead of “HaEmek” (The screenshot is taken from BI's official website, as shown by the red arrow).
42) Please replace "2.2. Material Preparation" with "2.2. Material preparation" (page 3).
Replaced.
43) Please change "(The maximum concentration of DMSO in the cell culture medium is 0.143% )" to "such that the maximum concentration of DMSO in the cell culture medium was 0.143%" (page 3).
Changed.
44) It is not exactly clear what the authors refer to by "KBr-pressed plates" in "10 μl of scabertopin and deoxyelephantopin solution was applied evenly on KBr tablets, evaporated and dehydrated by infrared lamp, and analyzed by fourier transform infrared (FTIR) spectroscopy using KBr-pressed plates" (page 3)? If these are the same as KBr tablets than please stick with only one terminology.
We have unified the term into “KBr pressed plates”.
45) Please replace "10 mg potassium bromide and and 1 mg scabertopin powder are" with "10 mg of KBr and and 1 mg of scabertopin powder were" (page 3).
Replaced.
46) From "IR spectroscopy was performed on the obtained background pieces and samples" (page 3) is not clear what is the difference between "background pieces" and "samples"?
The background pieces only contain 10mg KBr. The samples contain not only 10mg KBr, but also 1mg scabertopin powder. We have added notes in the manuscript as “IR spectroscopy was performed on the obtained background pieces (only KBr) and samples (KBr and scabertopin powder). ”
47) Please change "(UV9100B from Beijing LabTech Instrument Co., Ltd.) (Beijing, China)" to "(UV9100B from Beijing LabTech Instrument Co., Ltd, Beijing, China)" (page 3).
Changed.
48) Please replace "was no change in 0, 24 and 48 h, and it can be known" with "was no change at 0, 24, and 48 h and it can be inferred" (page 4).
Replaced.
49) Please change "light source and the type of the" to "light source, and the type of the" (page 4).
Changed.
50) Please provide manufacturer and its city and state headquarters for "McCoys’ 5A" and "F-12K" mentioned in "J82 cells were cultured in DMEM (10% FBS) medium, T24 and 5637 cells were cultured in RPMI 1640 (10% FBS), RT4 cells were cultured in McCoys’ 5A (10% FBS) medium, and SV-HUC-1 cells were cultured in F-12K (10% FBS) medium" (page 4).
We have updated the manufacture and catalogs of McCoys’ 5A and F-12K as “RT4 cells were cultured in McCoy’s 5A (Gibco, 16600082) (10% FBS) medium, and SV-HUC-1 cells were cultured in F-12K (Gibco, 21127022) (10% FBS) medium.” The city and state of Gibco’s headquarters have been provided previously (page 3).
51) Please replace "McCoys’" with "McCoy’s" (page 4).
Replaced.
52) Please change "of the supplied 4-mM calcein" to "of the supplied 4 mM calcein" (page 4).
Changed.
53) Please replace "observed by using a scanning" with "observed using a scanning" (page 4).
Replaced.
54) Please provide manufacturer and its city and state headquarters for "paraformaldehyde" mentioned in "The scabertopin-treated cells were fixed with 4% paraformaldehyde at room temperature and dehydrated with alcohol" (page 4).
We have updated the manufacturer and catalog of “paraformaldehyde (Beyotime Biotechnology, P0099-500ml)”. The city and state of Beyotime Biotechnology has been provided previously (page 3).
55) Please change "using a scanning electron microscope" to "using a scanning electron microscope (SEM)" (page 4).
Changed.
56) Please provide manufacturer and its city and state headquarters for "electron microscope fixative" mentioned in "The cells were treated with an electron microscope fixative for 2–4 h at 4°C, collected, and then centrifuged at low speed until green bean-sized clumps of cells could be seen at the bottom of the tubes" (page 5).
We have updated the manufacturer, catalog, city and state headquarters of “electron microscope fixative (Servicebio Technology, Wuhan, China, G1102-100ML)”.
57) Please change "transmission electron microscope, and" to "transmission electron microscope and" (page 5).
Changed.
58) Please get rid of the blank space on page 5.
Solved.
59) Please provide manufacturer and its city and state headquarters for "intracellular ROS assay kit" mentioned in "Intracellular ROS levels were measured by using an intracellular ROS assay kit" (page 6).
We have updated the manufacturer, catalog, city and state headquarters of “intracellular ROS assay kit (Solarbio Science & Technology, Beijing, China, CA1420)”.
60) Please change "gradient concentrations of scabertopin for 24-48 h" to "gradient concentrations of scabertopin for 24–48 h" (page 6).
Changed.
61) Please replace "cells were incubated with Scabertopin" with "cells were incubated with scabertopin" (page 6).
Replaced.
62) Please format "N" in "N-acetylcysteine" using italics (pages 6, 9, 11–13, 15).
Revised.
63) Please change "HY-B0215, MedChemExpress, Monmouth Junction, NJ, United States" to "MedChemExpress, Monmouth Junction, NJ, USA, HY-B0215" (page 6).
Changed.
64) Please replace "of scabertopin, and then loaded" with "of scabertopin and then loaded" (page 6).
Replaced.
65) Please provide city and state name for the headquarters of "KeyGEN BioTECH" mentioned in "In order to determine the type of ROS, J82 cells were treated with gradient concentrations of scabertopin, and then loaded with dihydroethidium (DHE) probe (KeyGEN BioTECH, KGAF019) at a concentration of 25 μM and incubated at 37 °C for 60 min in the darkness" (page 6).
It had been provided previously in Page 3 (Shanghai, China).
66) Please change "fresh culture medium, the" to "fresh culture medium and the" (page 6).
Changed.
67) Please replace "The cells were treat with gradient concentrations" with "Cells were treated with gradient concentrations" (page 6).
Replaced.
68) Please change "medium and mixed with the JC-1 solution, incubated at 37°C" to "medium, mixed with the JC-1 solution, and incubated at 37 °C" (page 6).
Changed.
69) Please provide model for the "fluorescence microplate reader" mentioned in "The fluorescence intensity of the cells was measured by the fluorescence microplate reader (BioTek), and the cells were imaged under the inverted fluorescence microscope" (page 6).
Provided as “fluorescence microplate reader (BioTek, Synergy H1)”.
70) Please provide model and manufacturer including its city and state headquarters for "inverted fluorescence microscope" mentioned in "The fluorescence intensity of the cells was measured by the fluorescence microplate reader (BioTek), and the cells were imaged under the inverted fluorescence microscope" (page 6).
It had been provided previously in page 5 (BioTek, Vermont, USA).
71) Please replace "(BioTek), and the cells were imaged under the" with "(BioTek) and the cells were imaged under an" (page 6).
Replaced.
72) Please change "Green fluorescence is an indicator of depolarized mitochondria, but" to "Whereas green fluorescence is an indicator of depolarized mitochondria," (page 6).
Changed.
73) Please replace "according to the following formula 2" with "according to the following formula" (page 6).
Replaced.
- Please format "GSH=total glutathione (GSH+GSSG) − 2×GSSG (formula 2)" (page 6) using the same font size as for formula 1 (page 4).
Revised.
75) Please replace "J82 cells were inoculated at the density of 8 × 105 cells/ml in 70 μl, into" with "70 μl of J82 cells were inoculated at the density of 8 × 105 cells/ml into" or "70 μl of J82 cell suspension was inoculated at the density of 8 × 105 cells/ml into" (page 6).
Replaced.
76) Please change "80206, ibidi, Grafelfing, Germany" to "Ibidi, Grafelfing, Germany, 80206" (page 6).
Changed.
77) Please replace "1-2 times with PBS, treated with different concentrations of the drugs" with "1–2 times with PBS, treated with different concentrations of drugs" (page 6).
Replaced.
78) Please change "area, following formula 3" to "area according to the following formula" (page 6).
Changed.
79) Please format "initial area−area at a certain time point initial area (formula 3)" (page 6) using the same font size as for formula 1 (page 4).
Revised.
80) Please leave a gap between "area that had healed at a certain time to the initial area, following formula 3" mentioned in "The percentage of wound healing was analyzed by using ImageJ software and calculated as the ratio of the initial scratch area minus the partially healed area that had healed at a certain time to the initial area, following formula 3" (page 6) and "initial area−area at a certain time point initial area (formula 3)" (page 6).
Updated.
81) Please replace "transwell plate, and the lower chamber" with "transwell plate and the lower chamber" (page 6).
Replaced.
82) Please specify the instrument used for photographing the images in "The number of migrated cells was recorded by photography under multiple high-magnification fields and by counting the number of migrated cells" (page 7).
We have updated the informations of manufacturer, city and state headquarters, and model as “The number of migrated cells was recorded by photography under multiple high-magnification fields by microscope (Etaluma, Inc., San Diego, CA, USA, LS720)”
83) Please change "RNase/PI (propidium Iodide)" to "RNase/propidium iodide (PI)" (page 7).
Changed.
84) Please replace "(BD FACSCanto) (Becton, Dickinson and Company, Franklin Lakes, NJ, USA)" with "(FACSCanto, Becton, Dickinson and Company, Franklin Lakes, NJ, USA)" (page 7).
Replaced.
85) Please change "with a small amount of binding Buffer" to "with a small amount of binding buffer" (page 7).
Changed.
86) Please replace "mixed with Annexin-V-FITC mixed" with "mixed with Annexin-V-FITC" (page 7).
Replaced.
87) Please change "through flow cytometry and the results were analyzed by using" to "by flow cytometry and the results were analyzed using" or "using flow cytometry and the results were analyzed by" (page 7).
Changed.
88) Please remove italics formatting from "ProteinSafe™", "ProteinSafe", and "ProteinExt" in "Add 10 μl ProteinSafe™ Phosphatase Inhibitor Cocktail (DI201, TransGen Biotech, Beijing, China) and ProteinSafe™ Protease Inhibitor Cocktail (DI101, TransGen Biotech) into every 1ml ProteinExt® Mammalian Total Protein Extraction Kit (TPEB, DE101, TransGen Biotech)" (page 7).
Removed.
89) Please format "μ" in "Add 10 μl" (page 7) using formatting consistent with the rest of the text.
Revised.
90) Please replace "Phosphatase Inhibitor Cocktail" with "Phosphatase inhibitor cocktail" (page 7).
Replaced.
91) Please change "Protease Inhibitor Cocktail" to "Protease inhibitor cocktail" (page 7).
Changed.
92) Please replace "1ml ProteinExt® Mammalian Total Protein Extraction Kit (TPEB, DE101, TransGen Biotech)" with "1 ml of ProteinExt® Mammalian total protein extraction kit (TPEB) (DE101, TransGen Biotech)" (page 7).
Replaced.
93) It is not unequivocally clear what the authors mean by "TPEB reagents" in "The cells were mixed with TPEB reagents on ice after 24 h of scabertopin treatment" (page 7)? Does "TPEB" relate to the mixture of "ProteinSafe™ Phosphatase Inhibitor Cocktail", "ProteinSafe™ Protease Inhibitor Cocktail", and "ProteinExt® Mammalian Total Protein Extraction Kit" or just "ProteinExt® Mammalian Total Protein Extraction Kit"?
"TPEB reagents" refers to the the mixture of "ProteinSafe™ Phosphatase Inhibitor Cocktail", "ProteinSafe™ Protease Inhibitor Cocktail", and "ProteinExt® Mammalian Total Protein Extraction Kit". We added notes in the manuscript as “TPEB mixed reagents are made by mixing the above three reagents according to the protocols. The cells were mixed with TPEB mixed reagents ”
94) Please change "by using a bicinchoninic acid" to "by using a Bicinchoninic acid" (page 7).
Changed.
95) Please replace "1/10000" with "1/10,000" (page 7 3x).
Replaced.
96) Please change "1/1000" to "1/1000" (page 7 18x).
We can not distinguish the difference between them, but we speculate that you want us to change "1/1000" to "1/1,000". We completed this change, although there were only 15 instead of 18.
97) Please replace "1/2000–1/20000" with "1/2,000–1/20,000" (page 7).
Replaced.
98) Please change "WBKLS0500, Merck Millipore, Darmstadt, Germany" to "Merck Millipore, Darmstadt, Germany, WBKLS0500" (page 7).
Changed.
99) Please replace "and the ggplot2 R package.The" with "and the ggplot2 R package. The" (page 7).
Replaced.
100) Please change "The chemical structure is shown in Fig. 1A, the" to "The chemical structure of scabertopin is shown in Fig. 1A. The" (page 8).
Changed.
101) Please replace "The FTIR spectrum of scabertopin are shown in Fig. 1B. And" with "The FTIR spectrum of scabertopin is shown in Fig. 1B and" (page 8).
Changed.
102) Please remove the tiny mark that appears above the first "u" in "α,β-unsaturated ester moiety" in Figure 1B.
The tiny mark is a dashed line indicating the peak value. We adjusted the position of the text in the picture.
103) Please change "0h, 24h and 48h are shown in Fig. 1C.According" to "0, 24, and 48 h is shown in Fig. 1C. According" (page 8).
Changed.
104) Please replace "λmax=214 Abs =0.55" with either "λmax=214 nm, Abs=0.55" or "λmax = 214 nm, Abs = 0.55" in Figure 1C.
Replaced.
105) Please provide error bars for all graphs in Figure 1D.
We re-made all graphs in Figure 1D to revised the Q105-Q108 as follow:
106) Please replot all graphs in Figure 1D using uniform x- and y-axis title font size.
Reploted.
107) Please replot all graphs in Figure 1D using uniform font size for the caption describing IC50 values.
Reploted.
108) The caption "Cell viability (%) seems to be cut through for 5637 cells both at the 24 and 48 h time points in Figure 1D. Please fix.
Fixed.
109) Please replace "the light source and the type of" with "the light source, and the type of" (page 8).
Replaced.
110) Please change "0 h, 24 h and 48 h were" to "0, 24, and 48 h were" (page 8).
Changed.
111) Please replace "it meant that the concentration of scabertopin" with "the concentration of scabertopin" (page 8 2x).
Replaced.
112) Whereas SV-HUC-1 cells are referred to as "human ureteral epithelial immortalized cells" in "The viability of scabertopin treated on bladder cancer cells (T24, J82, T24 and 5637) and human ureteral epithelial immortalized cells (SV-HUC-1) was determined by CCK-8 assay" (page 8), these are designated as "human bladder cancer cells" in "The results showed that scabertopin significantly inhibited the viability of the human bladder cancer cells (J82,T24, RT4 and SV-HUC-1) in a dose-dependent manner" (page 8). Please resolve this dichotomy.
This is a clerical error. We have revised as “The viability of-scabertopin treated on bladder cancer cells (T24, J82, RT4, and 5637) and human ureteral epithelial immortalized cells (SV-HUC-1) was determined by the CCK-8 assay. The results showed that scabertopin significantly inhibited the viability of the human bladder cancer cells (J82,T24, RT4, and 5637) in a dose-dependent manner (Fig. 1D).”
113) Please change "viability of scabertopin treated on" to "viability of scabertopin-treated" (page 8).
Changed.
114) Please replace "T24, J82, T24 and 5637" with "J82, T24, and 5637" (page 8).
Replaced.
115) Please change "was determined by CCK-8 assay" to "was determined by the CCK-8 assay" (page 8).
Changed.
116) From "The results showed that scabertopin significantly inhibited the viability of the human bladder cancer cells (J82,T24, RT4 and SV-HUC-1) in a dose-dependent manner (Fig. 1D)" is not evident what was the effect of scabertopin in 5637 cells?
This is the same clerical error as Q112, we have revised it as above.
117) Please replace "J82,T24, RT4 and SV-HUC-1" with "J82, T24, RT4 and SV-HUC-1" (page 8).
Replaced as Q112.
118) From "The 24-hour IC50 of scabertopin for bladder cancer cell lines is approximately 20 μM, and the 48-hour IC50 is even lower" (page 8) is not explicitly clear what was the 48 hour IC50 of scabertopin for bladder cancer cell lines?
It can be known from Fig. 1D that among various bladder cancer cells, IC50 of scarbertopin is approximately 18 μM. We have made a clearer explanation in the manuscript as “ The 24 h IC50 of scabertopin for bladder cancer cell lines is approximately 20 μM, and the 48 h IC50 is even lower (approximately 18 μM).”
119) Please replace "24-hour IC50" with "24 h IC50" (page 8).
Replaced.
120) Please change "and the 48-hour IC50 is even lower" to "while the 48 h IC50 is even lower" (page 8).
Changed.
121) Please replace "55.84μM" with "55.84 μM, respectively" (page 8).
Replaced.
122) Please change "selected for the follow-up study" to "selected for further study" (page 8).
Changed.
123) Please replace "Chemical structure, stability and" with "Chemical structure, stability, and" (page 8).
Replaced.
124) Please change "Chemical structure scabertopin, the" to "Chemical structure of scabertopin. The" (page 8).
Changed.
125) Please format "2" in "vibration of =CH2" using subscript (page 8).
Revised.
126) Please replace "3080cm-1, and" with "3080 cm-1 and" (page 8).
Replaced
127) Please format "-1" in "3080cm-1" using superscript (page 8).
Revised.
128) Please change "1760cm-1" to "1760 cm-1" (page 8).
Changed.
129) Please format "-1" in "1760cm-1" using superscript (page 8).
Revised.
130) Please replace "1710cm-1" with "1710 cm-1" (page 8).
Replaced.
131) Please format "-1" in "1710cm-1" using superscript (page 8).
Revised.
132) Please change "α,β-unsaturated ester, and" to "α,β-unsaturated ester and" (page 8).
Changed.
133) Please format "-1" in "1650 cm-1" using superscript (page 8).
Revised.
134) Please replace "0h, 24h and 48h" with "0, 24, and 48 h" (page 8).
Replaced.
135) Please change "0 h, 24 h and 48 h" to "0, 24, and 48 h" (page 8).
Replaced.
136) Please remove "Since the intensity of the UV maximum absorption peaks of scabertopin at 0 h, 24 h and 48 h were all around 0.55, it meant that the concentration of scabertopin did not change" from the legend of Figure 1 due to its duplicity within the main text (page 8).
Removed.
137) From "24 and 48hours IC50 of bladder cancer cell lines(J82,T24, RT4,5637)and human ureteral epithelial immortalized cells (SV-HUC-1) (n=4)" (page 8) is not clear whether what metric was used to represent the data and whether the experiments were independent?
We used CCK-8 kit to count the viable cell number of scarbertopin-treated bladder cancer cell lines, and measured the absorbance at 450 nm to evaluate the IC50 at 24 and 48 h. Each test was performed 4 times independently. We added error bar to the updated image to display the standard deviation of each group of data (as Q105 requested).
138) Please replace "24 and 48hours IC50" with "24 and 48 h IC50" (page 8).
Replaced.
139) Please change "lines(J82,T24, RT4,5637)and" to "lines (J82,T24, 5637, and RT4) and" (page 8).
Changed.
140) Please replace "n=4" with "n = 4" (page 8).
Changed.
141) Please change "J82 cells treated with Scabertopin." to "J82 cells treated with scabertopin" (page 8).
Changed.
142) Please replace "treated with scabertopin for 24 and 48 hours" with "treated with scabertopin for 24 and 48 h" (page 9).
Replaced.
143) Please change "probe, and found that scabertopin" to "probe and found that scabertopin" (page 9).
Changed.
144) Please replace "Fig.2A" with "Fig. 2A" (page 9).
Replaced.
- It is not clear what is the difference between Figure 2A and 2D? If these are identical experiments, please merge both figures into one. If these are different experiments, please relabel Figure 2D as 2A, Figure 2A as 2B, and Figures 2B,C as 2C,D.
Fig. 2A and 2D are not an identical experiment. Fig. 2A is shown the total ROS of the J82 cells after scabertopin-treated by using the ROS detection kit (Beyotime Biotechnology, S0033S). Figure 2D shows the superoxide anion in J82 cells after scabertopin-treated by using the DHE probe (KeyGEN BioTECH, KGAF019). However, we adjusted the order of the images in Fig. 2 to make it more logical as follow:
Figure 2. Increased mitochondrial ROS and reduced mitochondrial membrane potential in J82 cells treated with scabertopin. A. Scabertopin induced ROS production in a dose- and time-dependent manner; B. DHE probe assay showed that superoxide anion was positively correlated with the scabertopin dose; C. JC-1 assay showed that after 24 h treatment, scabertopin decreased mitochondrial membrane potential in J82 cells in a dose-dependent manner; D. JC-1 assay showed that NAC treatment could restore the mitochondrial membrane potential reduced by 24 h treatment of scabertopin to some extent; D. DHE probe assay showed that superoxide anion was positively correlated with the scabertopin dose; E. After 24 h treatment with 10μM scabertopin was efficient in depleting GSH through a dose-dependent mechanism in J82 cells, which could be reversed with NAC treatment. 5 μM of NAC was used in the experiments. Data represent the mean ± standard error of mean (s.e.m.) of three independent experiments. *P < 0.05, **P< 0.01, ***P < 0.001, ns: no significance vs. 0 μM scabertopin-treated group. NAC, N-acetylcysteine; SA: scabertopin.
146) Despite the authors claim that "To determine the mechanism of ROS production caused by scabertopin, we used JC-1 assay to observe the changes of ΔΨ in bladder cancer cells treated with scabertopin" (page 9), this experiment seems to have been performed only in J82 cells according to the legend of Figure 2B.
We have adjusted the wording here to make it more precise as “To determine the mechanism of ROS production caused by scabertopin, we used JC-1 assay to observe the changes of ΔΨ in J82 cells treated with scabertopin.”
147) Although the authors claim that "The results showed that ΔΨ decreased (red fluorescence increased) after scabertopin treatment (Fig.2B)" (page 9), red fluorescent signal seems to disappear following 10 and 15 uM scabertopin treatment. Please correct.
This is a clerical error. The decrease of ΔΨ is accompanied by the decrease of red fluorescence and the increase of green fluorescence. We have corrected this mistake as “The results showed that ΔΨ decreased (red fluorescence decreased while green fluorescence increased) after scabertopin treatment (Fig.2B)”
148) Please change "Fig.2B" to "Fig. 2B" (page 9).
- Please provide at least one scale bar for Figures 2B,C; 3D,E; 5A,B.
Provided.
150) Please change "0uM to "0", "10uM" to "10", and "15uM" to "15" in Figures 2B, 3C.
Changed.
- Please replace "0uM+NAC" to "0+NAC", "10uM+NAC" to "10+NAC", and "15uM+NAC" to "15+NAC" in Figures 2C, 3C.
Changed.
152) Please explain the use of "monomer/polymer" in the y-axis title of Figure 2C in the respective figure legend and/or in the Materials and Methods section.
We have added the explain of “monomer/polymer” in the Materials and Methods section as “ When ΔΨ at higher levels, JC-1 aggregates in the matrix of mitochondria to form polymers and produce red fluorescence. When ΔΨ is lower, JC-1 cannot aggregate in the mitochondrial matrix. At this time, JC-1 is monomer and can produce green fluorescence. The ratio of monomer/polymer represents the ratio of green fluorescence/red fluorescence, which can be used to measure the proportion of mitochondrial depolarization.”
153) Please replace "Rate (Monomer/polymer)" with "Rate (monomer/polymer)" in the y-axis title of Figure 2C.
Replaced.
154) Please change "Concentration of GSH(uM)" to "Concentration of GSH (uM)" or "GSH level (uM)" in the y-axis title of Figure 2E.
Changed.
155) Please indicate the abbreviation for "ns" in the legend of Figures 2–5.
We updated “ns: no significance” in the legend of Figures 2–5.
156) Please indicate the abbreviation for "SA" in the legend of Figures 2–3, 5.
We updated “SA: scabertopin” in the legend of Figures 2–5
157) From the legend of Figures 2B–E; 3G; 4A–C is not clear how long were J82 cells treated with scabertopin?
In this manuscript, if the treatment time of scabertopin is not explicitly mentioned, it defaults to 24 hours. We updated this information in page 7 as “In the following experiment, if the treatment time of scabertopin is not explicitly mentioned, it defaults to 24 hours.” Also, we updated the legends one by one.
158) From the legend of Figures 2C,E; 3C,E; 4C is not clear what concentration of NAC was used to treat J82 cells?
We had provided the concentration of NAC (5 mM) in Materials and Methods section, subchapter 2.9 (page 5). The same concentration was used in subsequent experiments. Also, we have updated the lengends with “5 μM of NAC was used in the experiments”
159) From the legend to Figure 2C is not clear what probe was used for imaging?
The probe was used in Fig. 2C is the same one as the Fig.2B, we have upadated this information as “JC-1 assay showed that NAC treatment can restore the mitochondrial membrane potential reduced by scabertopin treatment to some extent”. However, please note that we updated the order of images of Figure 2.
160) From the legend of Figure 2E; 3A; 4C is not clear what concentration of scabertopin was used to treat J82 cells?
10 μM scabertopin was used in these experiments. We have provided this information in the each legends as “2E. 10μM scabertopin was efficient in depleting GSH through a dose-dependent mechanism in J82 cells, which could be reversed with NAC treatment”, “3A. SEM images show that in J82 cells treated for 24 h with 10μM scabertopin”, and “4C. And it can be reduced after NAC treatment in the group of 10 μM scabertopin treatment”.
161) The mention of "fluorescence microplate reader" in "Detected by fluorescence microplate reader, the ΔΨ of scabertopin treatment group decreased significantly compared with the control group" (page 9) is rather puzzling since it is not clear whether the same device was used only as part of this measurement or also as part of the ΔΨ measurement mentioned in "The results showed that ΔΨ decreased (red fluorescence increased) after scabertopin treatment (Fig.2B)" (page 9)?
This is a mistake in language expression. Both refer to the same measurement. We changed the wording as “The results showed that ΔΨ decreased (red fluorescence decreased while green fluorescence increased) after scabertopin treatment (Fig. 2B). In conclusion, the ΔΨ of scabertopin treatment group decreased significantly compared with the control group”
162) "The ROS produced in mitochondria are hydrogen peroxide (H2O2) and superoxide anions (·O2-) in generally, but H2O2 cannot be detected by escaping into the cytoplasm through the inner mitochondrial membrane, superoxide anion produced in mitochondria and transported to the cytoplasm has become a key signaling factor for mitochondrial ROS" is disputable for the two following reasons:
- a) Although it is true that H2O2 is produced in mitochondria, this seems to be solely due to its conversion from superoxide, and therefore the claim that mitochondria produce H2O2 is rather misleading. Please either provide reference demonstrating that H2O2 is directly generated withing the mitochondria or rephrase the sentence to take into account this context.
This is a mistake in language expression. The hydrogen peroxide in mitochondria obviously comes from superoxide anion produced by superoxide dismutase. We rephrased the sentence as “The ROS in mitochondria are usually in the form of hydrogen peroxide (H2O2) and superoxide anions (·O2-)”.
- b) Given the short half-life of superoxide, the claim that superoxide is transported to the cytoplasm for signaling requires a specific reference.
In fact, the release of the superoxide anion from mitochondria to cytosol is controled by voltage-dependent anion channels (PMID: 12482755). It can also participate in a variety of signal regulation, such as cell surface receptor Fas (PMID: 8617197), or activate JNK in hepatic stellate cells (PMID: 19386027). However, these studies are not highly relevant to the studies concerned in this manuscript, and we do not think that these reference need to be cited in this manuscript.
- c) The claim that "H2O2 cannot be detected by escaping into the cytoplasm through the inner mitochondrial membrane" is debatable since H2O2 seems to be lipid soluble and therefore would be expected to freely permeate phospholipid membranes. In contrast to superoxide, the extramitochondrial signaling function of H2O2 would be more expectedly be promoted by its neutrality and increased stability as a signaling molecule capable of diffusing across longer distances.
This conclusion is based on reference 31 (PMID: 32130885). In fact, although hydrogen peroxide is lipid soluble, it still needs to pass through aquaporin to leave mitochondria instead of freely permeate phospholipid membranes (PMID: 25342630). Thioredoxin system is responce for is responsible for restricting intracytosolic diffusion of H2O2 (PMID: 30864831), and it also been found that thioredoxin system is responsible for entrapping H2O2 within the mitochondrial matrix. H2O2 is able to leak from the mitochondria to the cytosol only in case of a decrease in thioredoxin system functionality or upon a sustained generation of high concentrations of H2O2 in the matrix, when the thioredoxin system is most likely exhausted (PMID: 30138563). In addition, there are a large number of enzymes that can degrade H2O2 rapidly in the cytoplasm matrix (PMID: 32130885). Therefore, the lipid solubility and stability of hydrogen peroxide can not be the condition that it can freely permeate through the mitochondrial membrane and conduct long-distance signal transmission.
We revised the sentence to make it more accurate as “However, H2O2 is hardly to escape through the mitochondrial membrane and be detected in the cytoplasm, superoxide anion produced in mitochondria and transported to the cytoplasm has become a key signaling factor for mitochondrial ROS”.
163) "The ROS produced in mitochondria are hydrogen peroxide (H2O2) and superoxide anions (·O2-) in generally, but H2O2 cannot be detected by escaping into the cytoplasm through the inner mitochondrial membrane, superoxide anion produced in mitochondria and transported to the cytoplasm has become a key signaling factor for mitochondrial ROS" (page 9) is too long. Please split into two sentences.
It is split into “ The ROS in mitochondria are usually in the form of hydrogen peroxide (H2O2) and superoxide anions (O2·-). However, H2O2 is hardly to escape through the mitochondrial membrane and be detected in the cytoplasm, superoxide anion produced in mitochondria and transported to the cytoplasm has become a key signaling factor for mitochondrial ROS.”
164) Please replace "The ROS produced in mitochondria are hydrogen peroxide (H2O2) and superoxide anions (·O2-) in generally" with "In general, ROS produced in mitochondria are hydrogen peroxide (H2O2) and superoxide anion (O2·-)" (page 9).
Replaced.
- "H2O2 cannot be detected by escaping into the cytoplasm" does not seem to make sense in "The ROS produced in mitochondria are hydrogen peroxide (H2O2) and superoxide anions (·O2-) in generally, but H2O2 cannot be detected by escaping into the cytoplasm through the inner mitochondrial membrane, superoxide anion produced in mitochondria and transported to the cytoplasm has become a key signaling factor for mitochondrial ROS" (page 9). Please rephrase.
It has been rephrased as “The ROS in mitochondria are usually in the form of hydrogen peroxide (H2O2) and superoxide anions (O2·-). However, H2O2 is hardly to escape through the mitochondrial membrane and be detected in the cytoplasm, superoxide anion produced in mitochondria and transported to the cytoplasm has become a key signaling factor for mitochondrial ROS.”
166) Please replace "group(Fig.2C)" with "group (Fig. 2D)" (page 9).
It has been replaced. Please note that we have updated the order of images in Figure 2.
167) Please change "reduced glutathione (GSH) is an important antioxidant in cells, and" to "GSH is an important antioxidant in cells and" (page 9).
Changed.
168) Please replace "Fig.2D" with "Fig. 2E" (page 9).
Changed.
169) Please change "intracellular ROS and the efficacy" to "intracellular ROS, and the efficacy" (page 9).
Changed.
170) Please replace "Reduced mitochondrial membrane potential and increased mitochondrial ROS" with "Increased mitochondrial ROS and reduced mitochondrial membrane potential" (page 9).
Changed.
171) Please change "Scabertopin induced ROS production" to "A. Scabertopin induced ROS production" (page 9).
Changed.
172) Please replace "cells in a dose-dependent manne" with "cells in a dose-dependent manner" (page 9).
Replaced.
173) Please change "NAC treatment can restore" to "NAC treatment could restore" (page 9).
Changed.
174) Please change "correlated with scabertopin dose" to "correlated with the scabertopin dose" (page 9).
Changed.
175) Please replace "could also deplete" with something like "was efficient in depleting" (page 9).
Replaced.
176) Please change "cells and it can be reduced after" to "cells, which could be reversed with" (page 9).
Changed.
177) Please replace "Data in Fig.2" with "Data" (page 9).
Replaced.
178) Please change "**< 0.01" to "**P < 0.01" (pages 9, 13).
Changed.
179) Please replace "N-acetylcysteine;" with "N-acetylcysteine" (page 9).
Since there is no difference can be distinguished here, we have changed it to italics according to the previous suggestions. N-acetylcysteine
180) Please change "was not apoptosis and ferroptosis." to "was not apoptosis or ferroptosis" (page 9).
Changed.
181) Please replace "showed that the cells without" with "showed that cells without" (page 9).
Replaced.
182) Please change "Fig3A" to "Fig. 3A" (page 10).
Changed.
183) Please replace "And then, we analyzed the effect of scabertopin treatment on the potential disruption of cell cycle phases by using flow cytometry to elucidate preliminarily the potential mechanism of scabertopin cytotoxicity in bladder cancer cells" with "To preliminarily elucidate the potential mechanism of scabertopin cytotoxicity in bladder cancer cells, we analyzed the effect of scabertopin treatment on the potential disruption of cell cycle phases by using flow cytometry" (page 10).
Replaced.
184) Please change "(Fig.3B) that scabertopin treatment induced a significant increase in the percentage of cells both in the S and G2/M phase in a concentration-dependent manner" to "that scabertopin treatment induced a significant increase in the percentage of cells both in the S and G2/M phase in a concentration-dependent manner (Fig. 3B)" (page 10).
Changed.
185) Please remove the caption "bar=50 um" from Figure 3A.
Removed.
186) The scale bar presented in Figures 3A; 4A is rather confusing as it seems to be dotted.
This is because scale bars are very short. We have redrawn these scale bars.
187) Please either make the ineligible white captions at the bottom of the micrographs presented in Figures 3A; 4A visible in sufficient resolution so that they can be clearly read or remove them completely.
This promblem have been solved.
188) Please rescale all y-axis ranges to the same value in the flow cytometry data of Figure 3B so that the data can be directly compared.
I'm sorry that we can't fix this problem, because these pictures are automatically analyzed by the instrument and can't be modified once they are output as pictures. In order to facilitate readers to quickly compare the differences between the three, we have provided a statistical chart. We hope you can accept this answer, thank you.
189) Please provide y-axis title for the 10 uM scabertopin concentration in the flow cytometry data of Figure 3B.
Revised.
190) Please replace "FCS Percentage" to "FCS (%)" in the y-axis title in the quantitation graph of Figure 3B.
Replaced.
- Please provide description for the flow cytometry data in Figure 3B. It is namely not clear what does black trace, the gray-shaded peak, the plain red peaks, and the black arrowheads indicate? Moreover, the black arrowhead seems to be missing from the small plain red peak in the no scabertopin condition. More importantly, provide a clue as to how the cell cycle phases were deduced from the flow cytometry data in the respective figure legend.
We have updated the Fig. 2B as follow:
The black arrowhead is indicated the value of the peak. The reason why an arrow seems to disappear is that the wave is not enough to form a peak. We added a illustration in the legends as “The black arrowheads in Fig. 3B indicate the values of the peaks”.
At the same time, we indicated the abbreviation for “dip” as “diploid” in the legends of Fig. 3.
192) Please indicate the abbreviation for "PE-A" and "FCS" in the legend of Figure 3B.
We have indicated the abbreviation in the legend of Figure 3 as “PE-A: phycoerythrin area” and “FCS” is a clerical error. It should be FSC, which means forward scanner. We corrected it in the Fig. 3.
193) The claim that "The transfer of the cell membrane phospholipid phosphatidylserine from the inner layer of the plasma membrane to the outer layer of the plasma membrane is one of the early features of apoptosis, and a single positive cell with high affinity for AnnexinV-FITC/PI for PS was considered as a typical cell in early apoptosis, i.e. Q4, whereas AnnexinV-FITC and PI double-positive cells are considered as necrotic cells, i.e. Q2" (page 10) is not correct since cells that stain positive for both FITC Annexin V and PI are, in fact, expected to undergo the end stage of apoptosis, necrosis, or are already dead.
We are appreciated that you have pointed this problem out. We have revised this sentence as “The transfer of the cell membrane phospholipid phosphatidylserine from the inner to the outer layer of the plasma membrane is one of the early features of apoptosis. A single positive for AnnexinV-FITC was considered as a typical cell in early apoptosis, i.e. the fourth quadrant (Q4), whereas AnnexinV-FITC and PI double-positive cells were considered as in the end stage of apoptosis, necrosis, or are already dead, i.e. the second quadrant (Q2).. ”
194) It is not exactly clear what the authors mean by "high affinity for AnnexinV-FITC/PI for PS" in "The transfer of the cell membrane phospholipid phosphatidylserine from the inner layer of the plasma membrane to the outer layer of the plasma membrane is one of the early features of apoptosis, and a single positive cell with high affinity for AnnexinV-FITC/PI for PS was considered as a typical cell in early apoptosis, i.e. Q4, whereas AnnexinV-FITC and PI double-positive cells are considered as necrotic cells, i.e. Q2" (page 10)?
This is a misunderstanding caused by the expression, "The transfer of the cell membrane phospholipid phosphatidylserine from the inner to the outer layer of the plasma membrane is one of the early features of apoptosis. A single positive for AnnexinV-FITC was considered as a typical cell in early apoptosis, i.e. the fourth quadrant (Q4), whereas AnnexinV-FITC and PI double-positive cells were considered as in the end stage of apoptosis, necrosis, or are already dead, i.e. the second quadrant (Q2).”.
195) "The transfer of the cell membrane phospholipid phosphatidylserine from the inner layer of the plasma membrane to the outer layer of the plasma membrane is one of the early features of apoptosis, and a single positive cell with high affinity for AnnexinV-FITC/PI for PS was considered as a typical cell in early apoptosis, i.e. Q4, whereas AnnexinV-FITC and PI double-positive cells are considered as necrotic cells, i.e. Q2" (page 10) is too long. Please split into at least two sentences.
It has been splite as “The transfer of the cell membrane phospholipid phosphatidylserine from the inner to the outer layer of the plasma membrane is one of the early features of apoptosis. A single positive for AnnexinV-FITC was considered as a typical cell in early apoptosis, i.e. the fourth quadrant (Q4), whereas AnnexinV-FITC and PI double-positive cells were considered as in the end stage of apoptosis, necrosis, or are already dead, i.e. the second quadrant (Q2).”.
196) Please change "layer of the plasma membrane to the outer layer" to "to the outer layer" (page 10).
Changed.
197) Please replace "of apoptosis, and a single" with "of apoptosis and a single" (page 10).
According to the suggestion 195, we have split the sentence in here as “The transfer of the cell membrane phospholipid phosphatidylserine from the inner to the outer layer of the plasma membrane is one of the early features of apoptosis. A single positive for AnnexinV-FITC was considered as a typical cell in early apoptosis, i.e. the fourth quadrant (Q4), whereas AnnexinV-FITC and PI double-positive cells were considered as in the end stage of apoptosis, necrosis, or are already dead, i.e. the second quadrant (Q2).”
198) Please change "double-positive cells are considered" to "double-positive cells were considered" (page 10).
Changed.
199) Please replace "Fig.3C" with "Fig. 3C" (page 10).
Replaced.
200) It is not exactly clear what the authors mean by "which were similar to the percentages of necrotic cells" in "Under scabertopin treatment, the number of necrotic cells significantly increased in a dose-de-pendent manner, which were similar to the percentages of necrotic cells" (page 10)? Do the authors indeed mean to say that the numbers of necrotic cells were similar to the obtained percentage values?
This is a mistake of language expression. We have deleted the error and redundant expression as “Under scabertopin treatment, the number of necrosis cells significantly increased in a dose-dependent manner”.
201) Please change "cells significantly increased in a dose-de-pendent manner, which were similar to the percentages of necrotic cells" to "cells, which were similar to the percentages of necrotic cells, significantly increased in a dose-de-pendent manner" (page 10).
We have updated this sentence as above.
202) The different values plotted for "Q2" in the upper and lower quantitation graphs of Figure 3C is puzzling as one would expect identical cell distribution. Please compile both graphs into one including the effect of NAC.
According to this and the following comments, we have remade the quantitation graph of Fig. 3C as follow:
203) The y-axis "PI PerCP-Cy5-5-A" title seems to be cut through for the 10 uM no NAC datapoint in flow cytometry data of Figure 3C. Please fix.
Fixed
204) Please replace "Cell Distribution(%)" with "Cell distribution (%)" in the y-axis title in the upper quantitation graph of Figure 3C.
Replaced as above.
205) Please change "Cell Distribution (%)" to "Cell distribution (%)" in the y-axis title in the lower quantitation graph of Figure 3C.
We compiled both graphs into one as above
- Please indicate the abbreviation for "PerCP-Cy5-5-A" in the legend of Figure 3C.
We have indicated it as “PerCP-Cy5-5-A: Peridinin-Chlorophyll-Protein Complex-Cyanine5.5 area”
207) Please replace "Cell viability was measured by using water-soluble tetrazolium salts and green fluorescent calcein-AM" to something like "In addition, water-soluble tetrazolium salts and green fluorescent calcein-AM were deployed" (page 10).
Replaced.
208) From "The results showed that in contrast to the control treatment, scabertopin increased cytotoxicity in a concentration- and time-dependent ways, thus resulting in cell death while significantly reducing cell survival (Fig.3D)" is not exactly clear why the authors are redundantly mentioning that scabertopin reduced cell survival along its capacity to induce cell death?
This is a mistake in our expression. We have deleted redundant descriptions.
209) Please change "results showed that in" to "results showed that, in" (pages 10, 11).
Changed.
210) Please replace "concentration- and time-dependent ways" with "concentration- and time-dependent manner" or "concentration- and time-dependent fashion" (page 10).
Replaced.
211) Please change "Fig.3D" to "Fig. 3D" (page 10).
Changed.
212) Please replace "24" to "24h" with "48" to "48h" in the y-axis title of the micrographs of Figures 3D,E.
Replaced.
213) Please change "24h" to "24 h" with "48h" to "48 h" in the x-axis title in the quantitation graphs of Figures 3D,E.
Changed.
214) Please change "Viability(%)" to "Viability (%)" in the quantitation graphs of Figures 3D,E.
According to this and the following comments, we have remade the quantitation graph of Fig. 3C as follow:
215) Please compile the quantitation graphs of Figures 3D,E into one including the demonstration of any effect that NAC might have on cell viability.
They have been compiled as above.
216) From the legend of Figures 3D,G is not clear what cells were used for the experiment?
We added “J82 cells” in the legends of Figures 3D, G as “D. NAC treatment reduces scabertopin-induced J82 cells death;” and “Scabertopin did not significantly alter the expression of the ferroptosis and pyroptosis-related proteins GPX4 and GSDMD in J82 cells”.
217) Please change "apoptosis-related caspase proteins, Bcl-2" to "apoptosis-related caspase proteins Bcl-2" (page 10).
Changed.
218) Please replace "GSDMD on the other hand by" with "GSDMD, on the other hand, by" (page 10).
Replaced.
219) Please change "Scabertopin-treated" to "scabertopin-treated" (page 10).
Changed.
220) Please provide vertical marks for the kDa sizes depicted in the WB of Figures 3F,G; 4B,C; 5C,D.
Provided.
221) Please align vertically the lane descriptors "0", "10", and "15" in the WB of Figures 3F,G.
Aligned.
222) Please center vertically the y-axis title "Grayscale value" in the quantitation graph of Figures 3F; 4B.
Centered.
223) Please replace "GasderminD" to "GSDMD" in the x-axis of the quantitation graph of Figure 3G.
Replaced.
224) Please format the y-axis numbers to one decimal point in the quantitation graph of Figures 3G; 4B.
Changed.
225) Please increase the font size of the y-axis title of Figure 3G to match that of Figure 3F.
Revised.
226) From the legend of Figures 3F,G; 4B,C; 5C,D is not clear whether grayscale values were normalized to GAPDH?
Yes, we added this information in the each legends.
227) From the legend of Figure 3G is not clear which bands represent GSDMD and what is the lower molecular weight that gradually disappears with increasing concentration of scabertopin? Please label any nonspecific bands using an asterisk.
Because GSDMD (CST, 97558) can also recognize 30 and 43 kDa non-specific bands, respectively. We have marked these two places with an asterisk (*) and marked the molecular weight. And added an illustration in the legends of Fig. 3G.
228) Please replace "die in the manner of apoptosis, ferroptosis" with "die by apoptosis, ferroptosis," (page 10).
Replaced
229) Please change "area increased, and the cell" to "area increased and the cell" (page 11).
Changed.
230) Please replace "thinned out; Scabertopin" with "thinned out. Scabertopin" (page 11).
Replaced.
231) Please change "Scabertopin induced cell cycle arrest (B) and necrosis (C) in J82 cells for 24h" to "24 h scabertopin treatment induced cell cycle arrest (B) and necrosis (C) in J82 cells" (page 11).
We have changed it as “24 h scabertopin treatment induced cell cycle arrest (B) and death (C) in J82 cells in a concentration-dependent manner”.
232) "D. NAC treatment reduces scabertopin-induced death" does not seem to be correct since NAC was not used in Figure 3D.
We combined the legends of Fig. 3D and Fig. 3E as “Live/death cell assay shows scabertopin inhibited the viability of J82 cells after 24 and 48 h of treatment (D), and this effect could be reduced by NAC treatment (E).”
233) Despite the authors claim that "Live/death cells assay shows scabertopin inhibited the viability of J82 cells in a dose- and time-dependent manner after 24 and 48 h of treatment, and it can be reduced after NAC treatment of cells" (page 11), Figure 3D shows that there was, in fact, no time-dependent effect of scabertopin on cell viability. Please resolve this dichotomy.
We have resolved this problem as the answer of 232.
234) Please replace "NAC treatment reduces scabertopin-induced death" with "NAC treatment reduced scabertopin-induced cell death" (page 11).
The sentence has been fixed as the answer of 232
235) Please change "Live/death cells assay shows" to "Live/death cell assay shows that" (page 11).
Changed.
236) Please replace "and it can be reduced after NAC treatment of cells" with "and this effect could be reduced by NAC treatment" (page 11).
Changed.
237) Please change "Caspase-9, Caspase-8, Caspase-3" to "caspase-9, caspase-8, caspase-3" (page 11).
Changed.
238) Please replace "GPX4 and GSDMD" with "GPX4 and GSDMD, respectively" (page 11).
Replaced.
239) Please change "Data in Fig.3" to "Data" (page 11).
Changed.
240) Please replace "of three independent experiments, *P < 0.05; **P < 0.01; ***P < 0.001" with "of three independent experiments. *P < 0.05, **P < 0.01, ***P < 0.001" (page 11).
Replaced.
241) Please change "Bcl-2, B cell lymphoma-2; Bax, BCL-2-associated X; GSDMD, Gasdermin-D; GPX4, glutathione peroxidase 4; NAC, N-acetylcysteine" to " Bax, Bcl-2-associated X; Bcl-2, B cell lymphoma-2; GPX4, glutathione peroxidase 4; GSDMD, Gasdermin-D; NAC, N-acetylcysteine." (page 11).
Changed.
242) Please replace "We observed whether cell organelles were altered after scabertopin treatment by per-forming transmission electron microscopy to further determine the manner in which scabertopin induces death in bladder cancer cells" with something like "To further investigate the extent to which cell organelles were altered by scabertopin treatment and to further determine the manner in which scabertopin induces death in bladder cancer cells we performed transmission electron microscopy" (page 11).
Replaced.
243) Please change "the scabertopin-treated cells were swollen" to "scabertopin-treated cells were swollen" (page 12).
Changed.
244) Please replace "and disappeared organelles and perforation of" with "or absent organelles and perforated" (page 12).
Replaced.
245) Please change "Fig.4A" to "Fig. 4A" (page 12).
Changed.
246) Please comment on in the text as to why the control J82 cell contains rounded-like blebs at the periphery while the scabertopin-treated cell plasma membrane is rather smooth in Figure 4A.
Cellular blebbing is a unique form of dynamic protrusion emanating from the plasma membrane which can be either apoptotic or nonapoptotic in nature. Blebs have been observed in a wide variety of cell types and in response to multiple mechanical and chemical stimuli. They have been linked to various physiological and pathological processes including tumor motility and invasion, as well as to various immunological disorders (PMID: 26488882). The phenomenon of blebbing has been observed in response to brief exposure to extracellular alkaline pH, which leads to enhanced invasive capacity. Genetic or pharmacological targeting of cellular blebs could serve as a potential therapeutic option to control tumor metastasis (PMID: 26488882). However, the phenomenon of cellular blebbing will not be observed when neoptosis occurs (Reference 32; PMID: 34869390, table. 1).
Since these phenomena are the common features of tumor cells and necroptosis, we do not think it is necessary to make redundant explanations in the text, and our readers can obtain these information from the relevant reaseach article or review. We have provided a brief description in the text as “ The results showed that in contrast to the control group cells, scabertopin-treated cells were swollen with vacuolated cytoplasm, severely swollen mitochondria, lack of membrane blebbing, partially dissolved or absent organelles and perforated of cell membranes (Fig. 4A). These morphological changes are consistent with the characteristics of necroptosis. [32] ” (page 11).
247) In addition, please comment on in the text as to why the scabertopin-treated J82 cell contains way more organelles/secretory granules that the control cell in the zoomed area of Figure 4A.
Similar to the previous question, cell swelling and cell membrane perforation are the common features of neotropis. Therefore, under the observation of TEM, the secretory granules you refer to may be vacuoles in swollen cells. We believe that this issue does not need to be explained too redundantly in the text.
- Lastly, please explain in the text why the control J82 cell nucleus appears fragmented at its bottom left corner?
This phenomenon may be caused by cell changes during the preparation of samples for TEM observation. Figure 4A focuses on showing the characteristic morphological changes of the scarbertop created J82 cell. But in fact, when we enlarged the images, we found that the change might not be nuclear fragments, because the integrity and continuity of the nuclear membrane still remained. We have provided a clear image here. You can enlarge the image embedded in the document to observe.
249) From the zoomed image in Figure 4A is difficult to appreciate that the plasma membrane is without perforations in the control J82 cell as the boundary between the cell and the extracellular space is obscured by rounded-like blebs. Please fix.
It may be that the image quality is too low, which leads to this misunderstanding. In fact, the continuity and integrity of the cell membrane of the control J82 cells can be seen by zooming in the format of TIFF picture. We re-uploaded the images again. Of course, the quality of the images embedded in the text may still be slightly inferior to the original image.
250) From the legend of Figures 4A,C; 5C is not clear how many experiments were performed, what metric was used to represent the data, and whether the experiments were independent?
At least three independent samples have been observed for each group in Figure 4A, and all western blotting have conducted at least three independent experiments. According to the requirements of MDPI, we have uploaded all the gels images of western blotting to the website at the initial stage of submission. Thank you very much for putting forward this question. We have made illustration in the each legends.
251) Please indicate the abbreviation for "CON" in the legend of Figure 4.
It has been indicated as “control group”
252) Please remove the captions "Bar= 5.0 um" and "Bar= 2.0 um" from Figure 4A.
Removed.
253) Please replace "western blotting, and the expressions" with "western blotting and the expression" (page 12).
Replaced.
254) Please change "phosphorylated-RIP1, RIP3" to "phosphorylated RIP1, RIP3," (page 12).
Changed.
255) Please replace "Fig.4B" with "Fig. 4B" (page 12).
Replaced.
256) Please swap the order of the phosphorylated and total proteins presented in the WB of Figures 4B; 5C such that the bands of the phosphorylated version of each protein are right above those of the total protein.
Swaped.
257) Please swap the order of the phosphorylated and total proteins presented in the quantitation graph of Figures 4B; 5C such that the bars corresponding to the phosphorylated version of each protein appear before those of the total protein in the left to right order.
Swaped
258) From Figures 4B,C is not clear at which amino acid was phosphorylation of RIP1, RIP3, and MLKL detected? Please fix.
Fixed.
259) Please change "using of NAC can significantly inhibit the expression of the p-RIP1, p-RIP3, p-MLKL" to "NAC significantly inhibited the expression of phospho-RIP1, phospho-RIP3, and phospho-MLKL" or to "NAC significantly inhibited the expression of the phosphorylated forms of RIP1, RIP3, and MLKL" (page 12).
Changed.
260) Please replace "Fig.4C" with "Fig. 4C" (page 12).
Replaced.
261) Please replace "─" with "-" to match the symbol width of "+" in the WB of Figures 4C; 5D.
Replaced
262) Please increase the font size of the y-axis title of Figure 4C to match that of Figure 4B.
We have updated Fig. 4B and also fixed this problem.
263) Please change "J82 cells undergo necroptosis and" to "J82 cells undergo necroptosis, which" (page 12).
Changed.
264) Please replace "TEM images of" with "TEM images are shown for" (page 12).
Replaced.
265) Please change "10μM scabertopin (SA) treated J82 cells, the" to "J82 cells treated with 10 μM scabertopin (SA). The" (page 12).
Changed.
266) Please replace "the left, and the red arrow points out" with "the left and the red arrows point to" (page 12).
Replaced.
267) From "Scabertopin can significantly increase the phosphorylation of the necroptosis-related proteins RIP1, RIP3, and MLKL (n =3)" (page 12) is not clear what metric was used to represent the data and whether the experiments were independent?
We have added the illustration as “The grayscale values were normalized to GAPDH. Data represent the mean ± s.e.m. of three independent experiments” in the legends.
268) Please change "(n =3) ;" to "(n = 3);" (page 12).
Changed.
269) Please replace "C. And it can be reduced after NAC treatment" with "C. Scabertopin-induced phosphorylation of RIP1, RIP3, and MLKL could be reversed by NAC" (page 12).
Replaced.
270) Please change "*P < 0.05; **P < 0.01;" to "*P < 0.05, **P < 0.01," (page 12).
Changed.
271) Please replace "RIP1, receptor-interacting protein-1; RIP3:receptor-interacting protein-3; p-, phosphorylated. MLKL, mixed lineage kinase like; NAC, N-acetylcysteine;" with "MLKL, mixed lineage kinase like; NAC, N-acetylcysteine; p-, phosphorylated; RIP1, receptor-interacting protein" (page 12).
Replaced.
272) Despite the authors claim that they have performed "Transwell assays" in "We performed wound healing and Transwell assays to characterize how scabertopin affected the migratory and invasive abilities of bladder cancer cells" (page 12) as part of the "3.5. Scabertopin inhibits the migration/invasion of bladder cancer cells" chapter, there is no mention of this assay in this section. Please fix.
Fixed as “Similarly, the results of the transwell assays also showed (Fig.5B) that after scabertopin treatment, the invasive ability of cells was significantly reduced.”
273) From "3.5. Scabertopin inhibits the migration/invasion of bladder cancer cells" (page 12) is not unambiguously clear whether the authors mean to say that "Scabertopin inhibits the migration and invasion" or "Scabertopin inhibits the migration or invasion" of bladder cancer cells?
We have revised the subtitle as “3.5. Scabertopin inhibits the migration and invasion of bladder cancer cells”.
274) Please change "healing and Transwell assays" to "healing and transwell assays" (page 12).
Changed.
275) Please replace "As shown in the figure, scabertopin inhibited the wound-healing ability of J82 cells in a dose-dependent manner after 24h of treatment (Fig.5A)" with "To this end, 24h treatment with scabertopin inhibited the wound healing ability of J82 cells in a dose-dependent manner (Fig. 5A)" (page 12).
Replaced.
276) Please increase the font size of figure panel descriptors "A", "B", "C" to match that of panel "D" in Figure 5.
Increased.
277) Please replace "Wound healing percentage" to "Wound healing (%)" in the y-axis title of the quantitation graph of Figure 5A.
Replaced.
278) Please indicate the abbreviation for "CON" in the legend of Figure 5.
Updated.
279) From the legend of Figure 5A is not clear what do the red lines indicate? In addition, it is not clear what is the difference between the upper and lower panels?
We provided a brief illustation in the legends as “Scabertopin significantly inhibited the migration ability (difference in area reduction between red lines)”. The “0 h” and “24 h” have added in the Fig. 5A.
280) Please change "(Fig.5B) that after scabertopin treatment, the invasive ability of cells was significantly reduced" to "that after scabertopin treatment, the invasive ability of cells was significantly reduced (Fig. 5B)" (page 12).
Changed.
281) It is not exactly clear what the authors mean by "Relative Invaded cell numbers" in the y-axis title of the quantitation graph of Figure 5B. Please rephrase or provide description in the respective figure legend.
We changed as “cell numbers”.
282) Please replace "Relative Invaded cell numbers" with "Relative invaded cell numbers" in the y-axis title of the quantitation graph of Figure 5B.
It has been changed as above.
283) Please replace "(FAK, p-FAK, PI3K, p-PI3K, AKT,p-AKT and MMP-9)" with "p-FAK, FAK, p-PI3K, PI3K, p-AKT, AKT, and MMP-9" (page 12).
Replaced.
284) Please change "Western blot analysis (WB)" to "western blot analysis" (page 12).
Changed.
285) Please replace "p-FAK (Try397), p-PI3K(Y607), p-AKT(S472, S473, S474)" with "phospho-FAK (Tyr397), phospho-PI3K (Tyr607), and phospho-AKT (Ser472, Ser473, Ser474)," (page 12).
Replaced.
286) Please change "in a dose-dependent manner after scabertopin treatment (Fig.5C)" to "after scabertopin treatment in a dose-dependent manner (Fig. 5C)" (page 12).
Changed.
287) It is not clear what the authors refer to by "AKT1-3" in Figure 5C? Please explain in the respective figure legend.
This is because the regeant we used is “Anti-AKT1 + AKT2 + AKT3 antibody (ab179463)”. However, according to other reports (PMID: 33263944) which used this one, we have uniformly changed it to AKT (it also revised in Materials and Methods).
288) Please replace "p-FAK (Y397)" with "p-FAK (Y397)", "PI3K (Y607)" with "PI3K (Y607)", "AKT1-3" with "Akt1–3", "p-AKT (S472, S473,S474)" with "p-Akt (S472, S473, S474)", "kDA" with "kDa" 8x in Figure 5C.
Replaced.
289) Please horizontally align "FAK", "p-FAK (Y397)", "PI3K p85"," p-PI3K (Y607)", "AKT1-3", and "MMP-9" captions closer to the WB panels in Figure 5C.
Revised.
290) Please change "p-FAK (Y397)", "p-AKT (S472, S473,S474)" to "p-Akt (S472, S473, S474)", "kDA" to "kDa" 5x in Figure 5D.
Changed.
291) Please horizontally align "SA" and "NAC" closer to the WB panels in Figure 5D.
Changed.
292) Please replace "scabertopin can down-regulate" with "scabertopin can downregulate" (page 12).
Replaced.
293) Please change "activation of FAK/PI3K-AKT signaling pathway, and" to "activation of the FAK/PI3K/Akt signaling axis and" (page 12).
Changed.
294) Please replace "p-FAK, p-PI3K, p-Akt" with "phospho-FAK, phospho-PI3K, phospho-Akt," (page 12).
Replaced.
295) Please change "inhibited after treatment with NAC (Fig.5D)" to "inhibited by NAC (Fig. 5D)" (page 12).
Changed.
296) Please increase the font size of all captions of Figure 5D to match that of Figure 5C.
Revised.
297) Please replace "Scabertopin promoting the invasion" with "scabertopin-mediated invasion" (page 12).
Replaced.
298) Please change "migratory and invasive abilities" to "migratory and invasive ability" (page 13).
Changed.
299) Please replace "FAK /PI3K-Akt /MMP-9" with "FAK/PI3K/Akt/MMP-9" (page 13).
Replaced.
300) From "Scabertopin significantly inhibited the migration ability (n =3) (A) and invasion ability (n =5) (B) of J82 cells treated with gradient concentrations of scabertopin for 24 h" (page 13) is not clear what metric was used to represent the data and whether the experiments were independent?
The wound healing assay is conducted by comparing the healing area after treatment, and the transwell assay is a comparing invasive cells. We have added the illustration in the legends.
301) Please change "ability (n =3)" to "(n = 3)" (page 13 2x).
Changed.
302) Please replace "ability (n =5) (B)" with "(n = 5) (B) ability" (page 13).
Changed.
303) From "C. p-FAK, p-PI3K, p-AKT and MMP-9 expression levels significantly decreased after 24 h of scabertopin treatment" (page 13) is not explicitly clear which cells were used for the experiment?
We updated as “C. phospho-FAK, phospho-PI3K, phospho-Akt and MMP-9 expression levels significantly in J82 cells decreased after 24 h of scabertopin treatment.”
304) Please change "p-FAK, p-PI3K, p-AKT" to "phospho-FAK, phospho-PI3K, phospho-Akt," (page 13).
Changed.
305) From "And it can be reduced after NAC treatment. Data are presented as the mean ± s.e.m. of three independent experiments (n =3)" (page 13) is not clear what metric was used to represent the data and whether the experiments were independent?
We added illustration as “The grayscale values were normalized to GAPDH. Data are presented as the mean ± s.e.m. of three independent experiments (n = 3)”.
306) Please change "D. And it can be reduced after NAC treatment" to "D. Scabertopin-induced phosphorylation of p-FAK, p-PI3K, p-Akt and MMP-9 expression could be reversed by NAC" (page 13).
Changed.
307) Please replace "AKT, Akt-protein kinase B; p-AKT, phosphorylated Akt-protein kinase B; FAK, focal adhesion kinase; p-FAK, phosphorylated FAK; MMP-9, matrix metalloprotease-9; PI3K, phosphoinositide 3-kinase; p-PI3K, phosphorylated phosphoinositide 3-kinase; NAC, N-acetylcysteine" with "AKT, Akt-protein kinase B; FAK, focal adhesion kinase; MMP-9, matrix metalloprotease-9; NAC, N-acetylcysteine; p-, phosphorylated; PI3K, phosphoinositide 3-kinase" (page 13).
308) Please change "tumors.Natural" to "tumors. Natural" (page 13).
Changed.
309) Please replace "treated with scabertopin had" with "treated with scabertopin displayed" (page 13).
Replaced.
310) Please change "superoxide anion, a type of ROS" to "superoxide anion production" (page 13).
Changed.
311) Please replace "However, the further" with "However, further" (page 13).
Replaced.
312) Please change "cell death[33,34]" to "cell death [33,34]" (page 13).
Changed.
313) Please replace "For example: Isoalantolactone" with "For example, isoalantolactone" (page 13).
Replaced.
314) Please change "arrests cells in the S phase, thereby inhibiting cell proliferation and inducing apoptosis[35]" to "arrests these cells in the S phase, thereby inhibits cell proliferation and induces apoptosis [35]" (page 13).
Changed.
315) Please replace "migration of breast cancer cells[37]" with "migration capacity of breast cancer cells [37]" (page 14).
Replaced.
316) Please change "Try397 is the major" to "Tyr397 is the major" (page 14).
Changed.
317) Please replace "site of FAK, and p-FAK" with "site of FAK and phosphorylation of FAK" (page 14).
Replaced.
318) Please change "migration and invasion[38]" to "migration and invasion [38]" (page 14).
Changed.
319) Please replace "FAK/PI3K-AKT signaling pathway, and" with "FAK/PI3K/Akt signaling pathway and" (page 14).
Replaced.
320) Please change "invasive and metastatic[39]" to "invasive and metastatic [39]" (page 14).
Changed.
321) Please replace "pathway of FAK/PI3K-Akt/MMP-9" with "FAK/PI3K/Akt/MMP-9 axis" (page 14).
Replaced.
322) Please change "anion-dominated mitochondrial ROS (Fig2)" to "anion-dominated mitochondrial ROS (Fig. 2)" (page 14).
Changed.
323) From "However, in this study, although we found an increase in ROS, ferroptosis-related signaling molecules did not respond (Fig3)" (page 14) is not explicitly clear to what stimuli "ferroptosis-related signaling molecules" failed to respond?
We revised the wording as “However, in this study, although we found an increase in ROS, there is no difference in the expression of ferroptosis related molecules (Fig. 3).”
324) "In fact, ROS is a class of molecules including peroxides, superoxides, singlet oxygen, and free radicals" (page 14) seems to share the same meaning with "They include peroxides, superoxides, singlet oxygen, and free radicals. ROS levels are higher in different types of cancer cells than in normal cells." (page 13). Please remove this redundancy.
Removed.
325) Please replace "did not respond (Fig3)" with "did not respond (Fig. 3)" (page 14).
Replaced.
326) Please change "RNA and lipid molecules[41]During" to "RNA, and lipid molecules [41]. During" (page 14).
Changed.
327) Please replace "ferroptosis[42]" with "ferroptosis [42]" (page 14).
Replaced.
328) Please change "arachidonoyl(AA)- phosphatidylethanolamine-(PE) and adrenoyl(AdA)-PE(PE)" to "arachidonoyl (AA)-phosphatidylethanolamine (PE) and adrenoyl (AdA)-PE" (page 14).
Changed.
329) Please replace "[43].But this" with "[43]. But this" (page 14).
Replaced.
330) Please change "lipid peroxides[44]" to "lipid peroxides [44]" (page 14).
Changed.
331) Please replace "rupture and perforation" with "rupture, and perforation" (page 14).
Replaced.
332) Please change "cause various cell death" to "induce different forms of cell death" (page 14).
Changed.
333) It is not exactly clear what the authors mean by "damage of mitochondrial" and "reduction of MMPs" in "Although the mechanism of ROS in necroptosis is not fully understood, there is a lot of evidence that ROS plays a crucial role in many drugs-induced necroptosis [50,51], and is accompanied by damage of mitochondrial and reduction of MMPs[52]" (page 14)?
We revised the wording as “Although the mechanism of ROS in necroptosis is not fully understood, there is a lot of evidence that ROS plays a crucial role in many drugs-induced necroptosis [50,51], which is accompanied by mitochondrial injury and expression decreasing of MMPs [52]”.
334) Please replace "in many drugs-induced necroptosis [50,51], and" with "in drug-induced necroptosis [50,51], which" (page 14).
Replaced.
335) Please change "MMPs[52]" to "MMPs [52]" (page 14).
Changed.
336) Please replace "in turn, forming" with "via" (page 14).
Replaced.
337) It is not exactly clear what the authors mean by "anti-apoptotic cells" in "Because the occurrence of necroptosis not only bypasses apoptosis [55,56], but also causes death in anti-apoptotic cells" (page 15)?
We revised it as “Because the occurrence of necroptosis not only bypasses apoptosis [55,56], but also causes death in apoptosis-resistant cancer cells”.
338) Please change "we observed, such as" to "we observed such as" (page 15).
Changed.
339) The authors claim that "Flow cytometry showed that the treatment of scabertopin could increase the number of late apoptotic cells in the Q2 region" (page 15), however this region is specific for the mixture of late apoptotic, necrotic, and dead J82 cells. Please revise.
Revised.
340) Please replace "apoptosis-related Caspase protein, Bcl-2 and Bax had" with "apoptosis-related caspase proteins, Bcl-2, and Bax displayed" (page 15).
Replaced.
341) Please change "was the same as ferroptosis" to "was the same as for ferroptosis" (page 15).
Changed.
342) Please replace "p-RIP1, p-RIP3, and p-MLKL were significantly up-regulated" with "phospho-RIP1, phospho-RIP3, and phospho-MLKL were significantly upregulated" (page 15).
Changed.
343) Please change "RIP1/RIP3/MLKL, and finally causing" to "RIP1/RIP3/MLKL and finally triggering" (page 15).
Changed.
344) Please replace "activating RIP /RIP3 /MLKL phosphorylation, mediating J82 cell necroptosis, and inhibiting FAK /PI3K-Akt /MMP-9" with "activate RIP1/RIP3/MLKL phosphorylation, mediate J82 cell necroptosis, and inhibit the FAK/PI3K/Akt/MMP-9" (page 15).
345) Please change "This in turn inhibited the migration and invasion" to "This in turn inhibits the migration and invasion potential" (page 15).
346) Please replace "cells.AKT, Akt-protein kinase B; p-AKT, phosphorylated Akt-protein kinase B; FAK, focal adhesion kinase; p-FAK, phosphorylated FAK; MMP-9, matrix metalloprotease-9; PI3K, phosphoinositide 3-kinase; p-PI3K, phosphorylated phosphoinositide 3-kinase; RIP, receptor-interacting protein ; p-, phosphorylated. MLKL, mixed lineage kinase like; NAC, N-acetylcysteine;" with "cells. Akt, Akt-protein kinase B; FAK, focal adhesion kinase; MLKL, mixed lineage kinase like; MMP-9, matrix metalloprotease 9; NAC, N-acetylcysteine; PI3K, phosphoinositide 3-kinase; p-, phosphorylated; RIP, receptor-interacting protein" (page 15).
Changed.
347) Please change "In fact, some previous" to "In fact, previous" (page 15).
Changed.
348) Please replace "[57]. found that isodeoxyelephantopin and deoxyelephantopin can inhibit the activation of NF-κB by inducing the production of ROS and inhibit the growth of breast cancer" with "found that isodeoxyelephantopin and deoxyelephantopin can inhibit the activation of NF-κB by inducing the production of ROS and inhibit the growth of breast cancer [57]" (page 15).
Replaced.
349) Please replace "mitochondrial damage and apoptosis" with "mitochondrial damage, and apoptosis" (page 15).
It replaced as “mitochondrial injury, and apoptosis”
350) Please change "generation may come from" to "generation may stem from" (page 15).
Changed.
351) Please replace "Fig.1A" with "Fig. 1A" (page 15).
Replaced.
352) Please change "voltage-dependent anion channels (VDAC) on mitochondria, resulting" to "mitochondrial voltage-dependent anion channel resulting" (page 15).
Changed.
353) Please replace "through a certain role and" with "and" (page 16).
Replaced.
354) Please change "not only has the possible conditions to trigger this mechanism in structure, but" to something like "not only has the capacity to trigger this mechanism owing to its unique structure but" (page 16).
Changed.
355) Please replace "Lipinski et al. [64] proposed that most of the compounds that can be druggable should satisfy the molecular weight ≤500, LogP≤5, hydrogen bond donor count =0, hydrogen bond acceptor count =6, rotatable bond count =3 (PubChem CID: 93959111), and there is no evidence that SA is carcinogenic or genotoxic at present, which further supports that scabertopin can be a good drug candidate" with "To this end, Lipinski et al. proposed that most of druggable compounds should satisfy the condition of molecular weight ≤ 500, LogP ≤ 5, hydrogen bond donor count = 0, hydrogen bond acceptor count = 6, rotatable bond count = 3 (PubChem CID: 93959111), and there is no evidence that SA is carcinogenic or genotoxic at present, which further supports that scabertopin can be a promising drug candidate [64]" (page 16).
It has been revised as “Since the molecular weight (MW) of SA is 358.4 Da, the logarithm of lipid water partition coefficient (LogP) is 2.6, the hydrogen bond donor count is 0, the hydrogen bond acceptor count is 6, and the rotatable bond count is 3 [64]. These characteristics of SA well meet the Lipinski rules, which requires MW is less than 500 Da, LogP is between -2 and 5, the hydrogen bond donor count is less than 5, and the count of hydrogen bond acceptor and the rotatable bond is less than 10, respectively [65].”
Updated reference:
- PubChem [Internet]. Bethesda (MD), P.I.B. National Library of Medicine (US), National Center for Biotechnology Information; 2004-. PubChem Compound Summary for CID 93959111. Available online: https://pubchem.ncbi.nlm.nih.gov/compound/93959111
356) It is not clear what the authors mean by "LogP", " hydrogen bond donor count", "hydrogen bond acceptor count", and "rotatable bond count" in "Lipinski et al. [64] proposed that most of the compounds that can be druggable should satisfy the molecular weight ≤500, LogP≤5, hydrogen bond donor count =0, hydrogen bond acceptor count =6, rotatable bond count =3 (PubChem CID: 93959111), and there is no evidence that SA is carcinogenic or genotoxic at present, which further supports that scabertopin can be a good drug candidate" (page 16)?
We have revised as above, and we have define abbreviation of LogP as logarithm of lipid water partition coefficient
357) From "Lipinski et al. [64] proposed that most of the compounds that can be druggable should satisfy the molecular weight ≤500, LogP≤5, hydrogen bond donor count =0, hydrogen bond acceptor count =6, rotatable bond count =3 (PubChem CID: 93959111), and there is no evidence that SA is carcinogenic or genotoxic at present, which further supports that scabertopin can be a good drug candidate" (page 16) is also not clear what is the unit for the molecular weight limit and why the authors are mentioning "PubChem CID: 93959111"?
We have revised as above. We changed (PubChem CID: 93959111) to [64].
358) Lastly, from "Lipinski et al. [64] proposed that most of the compounds that can be druggable should satisfy the molecular weight ≤500, LogP≤5, hydrogen bond donor count =0, hydrogen bond acceptor count =6, rotatable bond count =3 (PubChem CID: 93959111), and there is no evidence that SA is carcinogenic or genotoxic at present, which further supports that scabertopin can be a good drug candidate" (page 16) is also not clear against which disease scabertopin represents a "good drug candidate"?
We have revised as above, this imprecise statement has been deleted.
359) "Lipinski et al. [64] proposed that most of the compounds that can be druggable should satisfy the molecular weight ≤500, LogP≤5, hydrogen bond donor count =0, hydrogen bond acceptor count =6, rotatable bond count =3 (PubChem CID: 93959111), and there is no evidence that SA is carcinogenic or genotoxic at present, which further supports that scabertopin can be a good drug candidate" (page 16) is too long. Please split into two sentences.
Revisde.
360) Please change "likely to be associated to the decreased of" to "likely to be associated with decreased" (page 16).
Revised.
361) Please replace "[67]NAC-treated" with "[67]. NAC-treated" (page 16).
Changed.
362) Please change "function and reduce ROS" to "function, and reduce ROS" (page 16).
Changed.
363) Please replace "FAK/ PI3K-AK/ MMP-9" with "the FAK/PI3K/Akt/MMP-9 axis" (page 16).
Replaced.
364) Please change "It can mediate the" to something like "Furthermore, scabertopin can mediate" (page 16).
Changed.
365) Please replace "RIP1/ RIP3/ MLKL" with "RIP1/RIP3/MLKL" (page 16).
Replaced.
366) Please change "phosphorylation, and inhibit" to "phosphorylation and inhibit" (page 16).
Changed.
367) Please replace "ROS scavenger, NAC, can" with "the ROS scavenger NAC can" (page 16).
Replaced.
368) Please change "scabertopin, but also" to "scabertopin but also" (page 16).
Changed.
369) Please replace "inhibited the abilities of invasion and migration" with "limit their invasive and migratory ability" (page 16).
Replaced.
370) Please change "the above two" to "the two abovementioned" (page 16).
Changed.
371) Please replace "Our study provided" with "In conclusion, our study provides" (page 16).
Replaced.
372) Please change "and scabertopin may be" to "suggesting that scabertopin may be" (page 16).
Changed.
373) Please replace "YX and DH" with "YX, and DH" (page 16).
Replaced.
374) Please change "ZN, all" to "ZN. All" (page 16).
Changed.
375) Please replace "(grant numbers 820QN423, 821QN424 and ZDYF2021SHFZ096).the" with "(grant numbers 820QN423, 821QN424, and ZDYF2021SHFZ096), the" (page 16).
Replaced.
376) Please change "(grant No. 82160531).Health" to "(grant number 82160531), and the Health" (page 16).
Changed.
- Please replace "grant No.21A200412" with "grant numbers 21A200412" (page 16).
Changed.
Once again, we appreciate your positive comments and valuable inputs. We hope the revised manuscript could meet your standard and the considered for final acceptance for publication in Cancers.
Sincerely
Zhenyu Nie, Yuanhui Gao, Shufang Zhang
Nov. 15. 2022

Round 3
Reviewer 2 Report
Major points:
1) Despite the authors claim in Figure 2A,B that scabertopin increased mitochondrial ROS in J82 cells, this conclusion is rather indirect as the probe they used was not specific for the detection of mitochondrial ROS. Please either deploy a probe specific for evaluating mitochondrial superoxide such as MitoSOX or demonstrate that a mitochondria-targeted antioxidant such as MitoTEMPO elicits an effect in the original assay. This shortcoming seems not to have been addressed from the last revision and resolving it will bring an extra layer of validation to the derived conclusion. Alternatively, the authors might want to reformulate the text to refer to (scabertopin-induced) "ROS" (general) instead of "mitochondrial ROS".
2) The claim that scabertopin induces necrosis in J82 cells in Figure 3C necessitates a more solid experimental support since Annexin V and PI double-staining is not a specific marker for necrosis. Please provide new data utilizing a more necrosis-specific assay such as the lactate dehydrogenase (LDH) release technique (Methods Mol Biol 979:65) to complement the rather generalized Annexin V and PI result. This issue seems not to have been corrected from the last round of revision and providing this data will strengthen the claim that, indeed, scabertopin acts as a potent inducer of necrosis. Alternatively, the authors could reformulate the text to comprise (scabertopin-induced) "cell death" (general) instead of "necrosis".
3) Although the authors claim that "We have re-made all the figure in high quality for the tiff format, We uploaded these pictures to the MDPI system when we uploaded the manuscript, we have made every effort to solve this problem. However, we speculate that the quality of images embedded in reviewed-manuscript may be slightly impaired." in their response, this persistent issue needs to be then fixed by resizing individual panels and at the level of the uploaded manuscript. Please provide expanded panels of high pixel-density resolution depth so that everyone including the peer-reviewers can clearly see the presented datasets when reading not only the online but also the printed version of the manuscript. Alternatively, these panels can be presented with sufficient size and pixel resolution as part of newly incorporated supplementary figures.
Minor points:
1) Please change "we also demonstrated that the half-inhibition" to "we also demonstrate that the half-inhibition" (page 1).
2) Please replace "human ureteral epithelial immortalized cells" with "immortalized human ureteral epithelial cells" (pages 1, 7, 14).
3) Please replace "The above observations indicated" with "The observations indicate" (page 1).
4) Please change "sesquiterpene lactone compound that is" to "sesquiterpene lactone compound, which is" (page 2).
5) "It is a kind of herbaceous plant belonging to phylum Angiosperm, class Dicotyledonous, order Campanulaceae, family Asteraceae, genus Elephantopus [8], which is widely distributed world-wide, particularly in East Asia, Southeast Asia, Africa, Australia, India, and South America, and has been reported various pharmacological properties such as anti-bacterial, anti-diabetic, anti-inflammatory, and anti-cancer efficacy" (page 2) is too long. Please split into at least two sentences.
6) Please replace "It is a kind of herbaceous plant" with "It is a herbaceous plant species" (page 2).
7) Please change "and has been reported various" to "and has been reported to elicit various" (page 2).
8) Please replace "anti-inflammatory, and anti-cancer efficacy" with "anti-inflammatory, and anti-cancer effects" (page 2).
9) Please change "Elephantopus scaber L. can significantly increase" to "Elephantopus scaber L., can significantly increase" (page 2).
10) Please replace "and down-regulate the NF-κB regulated" with "and downregulate the NF-κB-regulated" (page 2).
11) Please replace "effect and mechanism of scabertopin" with "effect and mechanism of action of scabertopin" (page 2).
12) Please change "triggering inflammatory response" to "triggering an inflammatory response" (page 2).
13) Please replace "mixed lineage kinase-like" with "mixed lineage kinase domain-like" (page 2).
14) Please change "are three key factors" to "are three key factors involved" (page 2).
15) Please change "Phosphorylated-MLKL (p-MLKL)" to "Phosphorylated-MLKL (phospho-MLKL)" (page 2).
16) Please change "and thus promote RIP1-dependent" to "and thus promoting RIP1-dependent" (page 2).
17) "potential strategy for cancer therapy" could be replaced with "potential general strategy for cancer therapy" (page 2).
18) Please change "we speculate that scabertopin also" to "we speculated that scabertopin also" (page 3).
19) "bladder cancer cell lines" appears twice in "Therefore, we determined our research object to study the effect of scabertopin in bladder cancer cell lines and explore its way to promote the death of bladder cancer cell lines" (page 3). Please fix.
20) Please replace "determined our research object" with "aimed" (page 3).
21) Please move "DMEM basal medium (10569010) and trypsin-EDTA (25200056) were purchased from Gibco (Grand Island, NY, USA). RPMI 1640 (PM150110) was purchased from Procell Life Science & Technology Co.,Ltd(Wuhan, China). Fetal bovine serum (FBS) (1907301) was purchased from Biological Industries (BI) (Kibbutz Beit Haemek, Israel)" (page 3) to the "2.5. Cell lines and culture" section.
22) Please cahnge "Co.,Ltd(Wuhan, China)" to "Co., Ltd (Wuhan, China)" (page 3).
23) The authors state that "Fetal bovine serum (FBS) (1907301) was purchased from Biological Industries (BI) (Kibbutz Beit Haemek, Israel)" (page 3), however at the same time they also state that "All cell culture media and FBS (30044333) were obtained from Gibco" (page 4). Please resolve this dichotomy.
24) Please replace "KeyGen BioTECH" with "KeyGEN BioTECH" (page 5).
25) Please change "(ab179463)、phospho-AKT" to "(ab179463), phospho-AKT" (page 3).
26) Please replace "1 ml dimethyl sulfoxide DMSO" with "1 ml DMSO" (page 3).
27) Please change "In the subsequent experiment, in" to "In" (page 3).
28) Please replace "control group media as control" with "control group media as a control" (page 3).
29) Please move "KBr (P116274-100g) was purchased from Aladdin Biochemical Technology Co., Ltd (Shanghai, China)." into the "2.1. Main reagents" section.
30) Please replace "operation is as follows" with "operation was as follows" or "procedure was as follows" (page 3).
31) Please change "its with UV maximum absorption" to "its UV maximum absorption" (page 4).
32) "There was no change in 0, 24 and 48 h and it can be inferred from Beer-Lambert law that when the length of the absorption cell, the light source, and the type of the substance to be tested are the same, the absorbance is strictly proportional to the substance concentration" (page 4) is puzzling due to the two following reasons:
a) I is not explicitly clear to what quantity change are the authors referring to? Please specify int the text.
b) It is not exactly clear what the authors mean by "0, 24 and 48 h"? Does this refer to cell treatment? If so, please report in the text.
33) Please replace "Abs: Absorbance" with "Abs: absorbance" (page 4).
34) Please change "in DMEM(10% FBS) medium" to "in DMEM (10% FBS) medium" (page 4).
35) From "Human bladder cancer cells (J82 and T24) and human ureteral epithelial immortalized cells SV-HUC-1 were inoculated into 96-well plates (100 μl per well)" (page 4) is not clear the seeding density used for "human bladder cancer cells (J82 and T24) and human ureteral epithelial immortalized cells SV-HUC-1"?
36) Please replace "(J82 and T24) and human ureteral epithelial immortalized cells SV-HUC-1" with either "(J82 and T24) and human ureteral epithelial immortalized cells (SV-HUC-1)" or "J82 and T24, and human ureteral epithelial immortalized cells SV-HUC-1" (page 4).
37) Please change "after being added with" to "after being supplemented with" (page 4).
38) Please replace "well, and incubation was" with "well and incubation was" (page 4).
39) Please change "Dose–response curves were generated using GraphPad Prism 7 (GraphPad Software, San Diego, CA, USA) software, and the absolute 50% inhibitory concentration (IC50) was determined" to "The absolute 50% inhibitory concentration (IC50) was determined from dose–response curves generated using the GraphPad Prism 7 (GraphPad Software, San Diego, CA, USA) software." (page 4).
40) From "Subsequently, the cells were washed with phosphate buffered saline (PBS) in accordance with the instructions of the LIVE/DEAD cell vitality assay kit (Invitrogen), and each group was incubated for 30 min at room temperature with 5 μl of the supplied 4 mM calcein AM stock solution to the 10 ml ethidium dimers (EthD-1)" (page 4) is not clear whether "4 mM calcein AM" refers to the final or stock concentration? Please specify in the text.
41) Please replace "(Invitrogen), and each group" with "(Invitrogen) and each group" (page 4).
42) Please change "for 30 min at room temperature with 5 μl of the supplied 4 mM calcein AM stock solution to the 10 ml ethidium dimers (EthD-1)" to "with 10 ml of ethidium dimers (EthD-1) supplied with 5 μl of 4 mM calcein AM stock solution for 30 min at room temperature" (page 4).
43) From "The human bladder cancer J82 cells were inoculated at the density of 2×104 cells/ml on slides attached to a 24-well plate, then treated with the drug at the concentration of 10 μM" (page 4) is not explicitly clear which drug are the authors referring to?
44) Please replace "24-well plate, then treated with" with "24-well plate and then treated with" (page 4).
45) From "After being cultured in the incubator for 24 h, the cells were taken out and washed with precooled PBS twice" (page 4) is not evident the temperature to which PBS was precooled?
46) "J82 cells were inoculated into 6-well plates then treated with gradient concentrations of scabertopin for 24 h" (page 5) does not indicate that the cells were seeded for TEM. Please fix.
47) Furthermore, from "J82 cells were inoculated into 6-well plates then treated with gradient concentrations of scabertopin for 24 h" (page 5) is not clear at what density were J82 cells inoculated?
48) Please provide model and manufacturer including its city and state headquarters for the "transmission electron microscope mentioned" in "After further wrapping, postfixation, dehydration, permeabilization, embedding, sectioning, and staining, the cells were observed under a transmission electron microscope (TEM) and images were collected for analysis" (page 5).
49) Please specify the density at which J82 cells were seeded for the "Measurement of reactive oxygen species", "Mitochondrial membrane potential assay", "GSH assay", "Cell cycle assays", "Cell apoptosis analysis", and "Western blot analysis" in the respective Materials and Methods section.
50) Please change "The J82 cells were treated with" to "J82 cells were treated with "(page 5).
51) Please replace "2 h with 5 mM ROS scavenger" with "2 h with 5 mM of the ROS scavenger" (page 5).
52) Please change "with 10 μM dichlorofluorescein" to "with 10 μM of the dichlorofluorescein" (page 5).
53) Please replace "BioTek, Vermont, USA, Synergy H1" with "Synergy H1, BioTek, Vermont, USA" (page 5).
54) Please replace "with dihydroethidium (DHE) probe" with "with the dihydroethidium (DHE) probe" (page 5).
55) Please change "BioTek, Synergy H1" to "Synergy H1, BioTek" (page 5 2x).
56) Please change "Cells were treat with gradient" to "Cells were treated with gradient" (page 5).
57) Please replace "and mixed with the JC-1 solution" with "mixed with the JC-1 solution" (page 5).
58) Please change "incubated at 37°C" to "incubated at 37 °C" (page 5).
59) Please replace "When ΔΨ at higher levels" with "When ΔΨ is high" (page 5).
60) Please change "When ΔΨ at lower level" to "When ΔΨ is low" (page 5).
61) Please replace "JC-1 is monomer" with "JC-1 is in its monomeric form" (page 5).
62) Please change "The GSH (reduced glutathione) and GSSG (oxidized glutathione disulfide) assay" to "The reduced glutathione (GSH) and oxidized glutathione disulfide (GSSG)" (page 5).
63) Please remove italic formatting from "μ" in "70 μl of J82 cells were" (page 6).
64) Please change "Etaluma, Inc., San Diego, CA, USA, LS720" to "LS720, Etaluma, Inc., San Diego, CA, USA" (page 6).
65) From "The cells were fixed by adding pre-cooled 70% ethanol and then washed with PBS to remove the fixative" is not evident the temperature to which 70% ethanol was precooled? (page 6).
66) Please provide model for the "flow cytometer" mentioned in "The samples were analyzed by flow cytometry and the results were analyzed using Modfit software" (page 6).
67) Please change "analyzed using Modfit" to "analyzed using the Modfit" (page 6).
68) "Add 10 μl ProteinSafe™ Phosphatase inhibitor cocktail (DI201, TransGen Biotech, Beijing, China) and ProteinSafe™ Protease inhibitor cocktail (DI101, TransGen Biotech) into every 1ml ProteinExt® Mammalian total protein extraction kit (TPEB) (DE101, TransGen Biotech)" does not seem to be principally correct as it is hard to imagine that a measured volume of an "inhibitor cocktail" can be meaningfully added into a kit. For a similar reason, the expression "1ml ProteinExt® Mammalian total protein extraction kit" does not make sense.
69) Please remove italic formatting from "μ" and "™" in "Add 10 μl ProteinSafe™" (page 6).
70) Please replace "DI201, TransGen Biotech, Beijing, China" with "TransGen Biotech, Beijing, China, DI201" (page 6).
71) Please change "DI101, TransGen Biotech" to "TransGen Biotech, DI101" (page 6).
72) Please replace "into every 1ml" with "into 1 ml of" (page 6).
73) Please change "DE101, TransGen Biotech" to "TransGen Biotech, DE101" (page 6).
74) "TPEB mixed reagents are made by mixing the above three reagents according to the protocols" (page 6) is puzzling for the two following reasons:
a) It is not exactly clear which TPEB reagents are the authors referring to? If it is "ProteinSafe™ Phosphatase inhibitor cocktail", "ProteinSafe™ Protease inhibitor cocktail", and ProteinExt® Mammalian total protein extraction kit", than the mixing of these reagents has been already mentioned in the previous sentence "Add 10 μl ProteinSafe™ Phosphatase inhibitor cocktail (DI201, TransGen Biotech, Beijing, China) and ProteinSafe™ Protease inhibitor cocktail (DI101, TransGen Biotech) into every 1ml ProteinExt® Mammalian total protein extraction kit (TPEB) (DE101, TransGen Biotech)". Moreover, the name "ProteinExt® Mammalian total protein extraction kit" does not seem to refer to a reagent that can be mixed as a liquid.
b) It is not explicitly clear what cells were "mixed" with TPEB mixed reagents"?
c) It is not clear whether the cells were harvested prior to their mixing with "TPEB mixed reagents"? If so, what was the means by which the cells were harvested?
d) It is not clear to which "protocols" are the authors referring to?
75) It is not exactly clear what the authors mean by "TPEB mixed reagents" in "The cells were mixed with TPEB mixed reagents on ice after 24 h of scabertopin treatment" (page 6)? Are they referring to the reagents from the "ProteinExt® Mammalian total protein extraction kit" mentioned in "Add 10 μl ProteinSafe™ Phosphatase inhibitor cocktail (DI201, TransGen Biotech, Beijing, China) and ProteinSafe™ Protease inhibitor cocktail (DI101, TransGen Biotech) into every 1ml ProteinExt® Mammalian total protein extraction kit (TPEB) (DE101, TransGen Biotech)" (page 6)? If so, from this sentence is not clear what is the third component in addition to the "ProteinSafe™ Phosphatase" and "ProteinSafe™ Protease" inhibitor cocktails?
76) Please replace "P0010, Beyotime Biotechnology" with "Beyotime Biotechnology, P0010" (page 6).
77) Please change "M00664, GenScript Biotech, Nanjing, China" to "GenScript Biotech, Nanjing, China, M00664" (page 6).
78) Please define abbreviation for "TBST" (page 7), "DCF" (page 8), "WB" (page 10).
79) Please provide catalog numbers for all antibodies used in the "2.16. Western blot analysis" section.
80) Please replace "The target protein lane were" with "The target protein lanes were" (page 7).
81) Please change "UV absorption peak of scabertopin" to "UV absorption peaks of scabertopin" (page 7).
82) Please replace "DMEM medium at 0, 24, and 48h" with "DMEM medium at 0, 24, and 48 h of treatment" (page 7).
83) Please change "Fig. 1C.According" to "Fig. 1C. According" (page 7).
84) "According to Beer-Lambert law, when the length of the absorption cell, the light source, and the type of the substance to be measured are the same, the absorbance is strictly proportional to the concentration of the substance" (page 7) seems to convey the same meaning as "There was no change in 0, 24 and 48 h and it can be inferred from Beer-Lambert law that when the length of the absorption cell, the light source, and the type of the substance to be tested are the same, the absorbance is strictly proportional to the substance concentration" (page 4). Please remove this unnecessary redundancy.
85) "Since the intensity of the UV maximum absorption peaks of scabertopin at 0, 24, and 48 h were all around 0.55, the concentration of scabertopin did not change" does not seem to make sense as "he intensity of the UV maximum absorption peaks" is not by any means expected to change "the concentration of scabertopin". Please revise.
86) Please replace "Since the intensity of the UV" with "Since the intensities of the UV" (page 7).
87) Please change "scabertopin can maintain stability" to "scabertopin maintains its stability" (page 7).
88) Please replace "The viability of scabertopin-treated on" with "The viability of scabertopin-treated human" (page 7).
89) Please change "T24, J82, RT4, and 5637" to "J82, T24, 5637, and RT4" (page 7).
90) Please replace "human bladder cancer cells (J82,T24, RT4, and 5637)" with "bladder cancer cell lines" (page 7).
91) Please change "cell lines is approximately 20 μM, and" to "cells is approximately 20 μM and" (page 7).
92) Please replace "lower (approximately 18 μM)" with "lower, approximately 18 μM" (page 7).
93) Please change "55.84 μM, which" to "55.84 μM, respectively, which" (page 7).
94) Although the authors state that "However, the IC50 values of scabertopin for SV-HUC-1 cells at 24 and 48 h were 59.42 and 55.84 μM, which were considerably higher than those for bladder cancer cells (Fig. 1D)" (page 7), the prior statement that scabertopin had an effect on the viability of SV-HUC-1 cells seems to be missing. Please fix.
95) Please replace "In the following experiment" with "In all subsequent experiments" (page 7).
96) Please change "is not explicitly mentioned, it defaults to 24 hours" to "defaults to 24 h unless explicitly mentioned otherwise" (page 7).
97) Please replace "λmax=214 Abs =0.55" with either "λmax=214 nm, Abs=0.55" or "λmax = 214 nm, Abs = 0.55" in Figure 1C.
98) Please change "3080 cm-1, and" to "3,080 cm-1 and" (page 8).
99) Please replace "1760 cm-1" with "1,760 cm-1" (page 8).
100) Please change "1710 cm-1" to "1,710 cm-1" (page 8).
101) Please replace "1650 cm-1" with "1,650 cm-1" (page 8).
102) Please change "medium at 0, 24 and 48h" to "medium at 0, 24, and 48 h" (page 8).
103) Please reorder the sequence of Figures or their citation order by rephrasing the text in the "3.2. Increased mitochondrial ROS levels in J82 cells treated with scabertopin." chapter so that they are consecutively presented in an alphabetical order: Fig. 2A, Fig. 2B, Fig. 2C,D.
104) Please replace "3.2. Increased mitochondrial ROS levels in J82 cells treated with scabertopin." with "3.2. Increased mitochondrial ROS levels in J82 cells treated with scabertopin" (page 8).
105) Please change "Fig. 2C-D" to "Fig. 2C,D" (page 8).
106) Please remove bold formatting from "ΔΨ" in "ΔΨ of scabertopin treatment group" (page 8).
107) Please replace "superoxide anions (O2·-)" with "superoxide anion (O2·-)" (page 8).
108) Please change "However, H2O2 is hardly" to "However, while H2O2 is not able" (page 8).
109) It is not clear whether DCF or DHE was used for the measurement of ROS in Figure 2B since this is indicated differently between the y-axis title, the figure legend, and the main text?
110) Despite the authors mention "scabertopin treatment group" in "The results showed that the level of superoxide anion in the scabertopin treatment group was significantly higher than that in the control group(Fig. 2B)" (page 8), two different "scabertopin treatment" groups (10 and 15) are, in fact, portrayed in Fig. 2B.
111) Please replace "group(Fig. 2B)" with "group (Fig. 2B)" (page 8).
112) Although the authors mention that the ability of scabertopin to "deplete GSH" is "one of the reasons for the accumulation of intracellular ROS" in "The results showed that scabertopin treatment can deplete GSH, which is one of the reasons for the accumulation of intracellular ROS, and the efficacy of scabertopin depleting GSH could be blocked by NAC" (page 9), the other reasons "for the accumulation of intracellular ROS" are not evident.
113) Please change "probe assay showed that superoxide anion" to something like "probe assay showed that superoxide anion generation" (page 9 2x).
114) Please replace "NAC treatment" with either "NAC pretreatment" or "NAC cotreatment" (pages 9, 10, 11).
115) Please replace "After 24 h treatment with 10μM" with "After 24 h treatment with 10 μM" (page 10).
116) Please remove italic formatting from "μ" in "After 24 h treatment with 10μM" (page 10).
117) The authors claim that "scabertopin was efficient in depleting GSH through a dose-dependent mechanism" in "After 24 h treatment with 10μM scabertopin was efficient in depleting GSH through a dose-dependent mechanism in J82 cells, which could be reversed with NAC treatment" (page 10), however only single dose of scabertopin (10μM) is shown in Figure 2E. Please provide at least one more dose of scabertopin for this assay or rephrase the sentence accordingly.
118) Please remove italic formatting from "μ" in "5 μM of NAC was used" (page 10).
119) Please change "SA: scabertopin" to "SA, scabertopin" (page 10).
120) Please briefly introduce the concept of ferroptosis and pyroptosis in the "3.3. Cell death induced by scabertopin treatment was not apoptosis and ferroptosis" section since this theoretical background is missing.
121) It is not clear why the authors mention apoptosis and ferroptosis but not pyroptosis in the context of scabertopin effects in the "3.3. Cell death induced by scabertopin treatment was not apoptosis and ferroptosis" section heading (page 10)?
122) Please replace "was not apoptosis and ferroptosis" with "was not apoptosis nor ferroptosis" (page 10).
123) "potential" appears twice in "To preliminarily elucidate the potential mechanism of scabertopin cytotoxicity in bladder cancer cells, we analyzed the effect of scabertopin treatment on the potential disruption of cell cycle phases by using flow cytometry" (page 10). Please rephrase.
124) Please change "on the apoptosis and necrosis" to "on the apoptosis and necrosis rates" (page 10).
125) Please change "The transfer of the cell membrane" to "Translocation of the" (page 10).
126) Please replace "apoptosis, necrosis, or are" with "apoptosis, necrosis, or" (page 10).
127) The use of "PE-A, phycoerythrin area" is not clear in the x-axis of Figure 3B. Please either explain in the Materials and Methods section or modify.
128) Similarly, the deployment of "PerCP-Cy5-5-A, Peridinin-Chlorophyll-Protein Complex-Cyanine5.5 area" in the y-axis of Figure 3C is also not clear. Please explain in the Materials and Methods or relabel.
129) Please remove "Date analyzed: 29-Jan-2022 Model: 1DA0n_DSD Analysis type: Automatic analysis Auto Linearity: Yes Ploidy Mode: First cycle is diploid Diploid: 100.00 % Dip G1: 52.64 % at 87.69 Dip G2: 11.36 % at 169.24 Dip S: 36.00 % G2/G1: 1.93 %CV: 4.48 Total S-Phase: 36.00 % Total B.A.D.: 0.52 % Debris: 1.25 % Aggregates: 1.70 % Modeled events: 1590 All cycle events: 1543 Cycle events per channel: 19 RCS: 0.865", "Date analyzed: 29-Jan-2022 Model: 1DA0n_DSD Analysis tpe: Manual analysis Auto Linearity: No Ploidy Mode: First cycle is diploid Diploid: 100.00 % Dip G1: 43.46 % at 83.29 Dip G2: 16.39 at 165.74 Dip S: 40.15% G2/G1: 1.99 %CV: 7.40 Total S-Phase: 40.15 % Total B.A.D.: 0.00 % Debris: 0.05 % Aggregates: 0.00 % Modeled events: 4808 All cycle events: 4805 Cycle events per channel: 58 RCS: 0.864", and "Date analyzed: 29-Jan-2022 Model: 1D0n_DSD Analysis type: Manual analysis Auto Linearity: No Ploidy Mode: First cycle is diploid Diploid: 100.00 % Dip G1: 30.99 % at 86.49 Dip G2: 28.65 % at 167.80 Dip S: 40.36 % G2/G1: 1.94 %CV: 5.94 Total S-Phase: 40.36 % Total B.A.D.: 0.00 % Debris: 0.50 % Aggregates: 0.10 % Modeled events: 3603 All cycle events: 3581 Cycle events per channel: 44 RCS: 0.644" from Figure 3B.
130) The "B" panel symbol is cut through in Figure 3B. Please fix.
131) From the legend to Figure 3B is not clear what does the black trace, the blue-shaded peak area, and the plain red peaks indicate? More importantly, provide a clue as to how the cell cycle phases were deduced from the flow cytometry data in the figure legend.
132) The authors claim that "The results showed that, in contrast to the control treatment, scabertopin increased cytotoxicity in a concentration- and time-de-pendent manner, thus resulting in cell death" (page 10), despite the time-dependent effect was not observed with 10 uM scabertopin in Figure 3D. Please correct.
133) Please change "on the other hand by WB" to "on the other hand, by WB" (page 10).
134) From "A. SEM images show that in J82 cells treated for 24 h with 10μM scabertopin, the cell spreading area increased and the cell pseudopods shortened and thinned out. 24 h scabertopin treatment induced cell cycle arrest (B) and death (C) in J82 cells in a concentration-dependent manner (the black arrowheads in Fig. 3B indicate the values of the peaks)" (page 11) is not clear what "values" are the authors referring to?
135) Please replace "10μM scabertopin" with "10 μM scabertopin" (page 11).
136) Please remove italic formatting from "μ" in "10μM scabertopin" (page 11).
137) Please change "black arrowheads in Fig. 3B" to "black arrowheads in panel B" (page 11).
138) Please replace "Live/death cell assay shows scabertopin inhibited" with "Live/dead cell assay shows that scabertopin inhibited" (page 11).
139) Please change "(D), and this effect" to "(D) and this effect" (page 11).
140) Please replace "caspase-9, caspase-8, caspase-3" with "caspase-3, -8, -9" (page 11).
141) Please change "* means" to "Asterisk denotes" (page 11).
142) Please replace "Peridinin-Chlorophyll-Protein Complex-Cyanine5.5" with "peridinin-chlorophyll-protein complex-cyanine5.5" (page 11).
143) Please change "further determine the manner in" to something like "delineate the mechanism by" (page 11).
144) Please replace "showed that in contrast" with "showed that, in contrast" (page 11).
145) Please change "lack of membrane blebbing, partially dissolved or absent organelles and perforated of cell membranes" to "partially dissolved or absent organelles, perforated cell membranes but without the presence of membrane blebbing" (page 11).
146) Please replace "and the expressions of phosphorylated" with "and the expression of phosphorylated" (page 11).
147) Please change "RIP1, RIP3 and MLKL" to "RIP1, RIP3, and MLKL" (page 11).
148) Please format "Grayscale value" using consistent formatting between the y-axis titles of Figure 4B and C.
149) Please replace "control group (CON) and J82 cells" with "control J82 cells (CON) and cells" (page 12).
150) Please change "right is the enlarged" to "right is the enlarged view of the" (page 12).
151) Please replace "red arrow points out" with "red arrow depicts" (page 12).
152) Please change "(n = 3) after 24 h treatment" to "after 24 h treatment (n = 3)" (page 13).
153) Please remove italic formatting from "μ" in "10 μM scabertopin" (page 13).
154) Please replace "mixed lineage kinase like" with "mixed lineage kinase domain-like" (pages 13, 16).
155) Please format "N" in "N-acetylcysteine" using italics (page 13).
156) Please replace "24h treatment with scabertopin" with "24 h treatment with scabertopin" (page 13).
157) Please change "Fig.5A" to "Fig. 5A" (page 13).
158) Please replace "results of the transwell assays also" with "results of the transwell assay" (page 13).
159) Please replace "Fig.5B" with "Fig. 5B" (page 13).
160) Please change "p-FAK, FAK, p-PI3K, PI3K, p-AKT" to "phospho-FAK, FAK, phospho-PI3K, PI3K, phospho-AKT" (page 13).
161) Please replace "and phospho-AKT (Ser472, Ser473, Ser474)" with "phospho-AKT (Ser472, Ser473, and Ser474)," (page 13).
162) Please replace "a very important" with "a key" (page 13).
163) Please change "scabertopin-mediated the invasion" to "scabertopin-mediated invasion" (page 13).
164) Please replace "These results suggested" with "These results suggest" (page 13).
165) Please provide panel symbols for Figure 5C,D.
166) Please change "Scabertopin treatment inhibited" to "Scabertopin treatment reduced" (page 13).
167) Please replace "between red lines" with "between the red lines" (page 13).
168) Please change "with (n = 5) (B) of J82 cells treated with gradient concentrations of scabertopin for 24 h" to "of J82 cells treated with gradient concentrations of scabertopin for 24 h (n = 5) (B)" (page 13).
169) Please change "phospho-FAK, phospho-PI3K, phospho-Akt and MMP-9" to "phospho-FAK, phospho-PI3K, phospho-Akt, and MMP-9" (page 13).
170) Please replace "p-FAK, p-PI3K, p-Akt and MMP-9" with "phospho-FAK, phospho-PI3K, phospho-Akt, and MMP-9" (page 13).
171) Please remove italic formatting from "μ" in "5 μM of NAC" (page 13).
172) Please remove italic formatting from "μ" in "0 μM scabertopin-treated group" (page 14).
173) Please incorporate a brief comment on the effects of scabertopin on pyroptosis into the Discussion section.
174) "drug discovery and development" appears twice in "Natural herbs are an important source of potential anticancer compounds in the field of drug discovery and development [33]. Natural compounds themselves contain unique structurally diverse molecules with multiple targets, making them ideal candidates for drug discovery and development" (page 14). Please fix.
175) Please change "in the field of" to "for" (page 14).
176) Please replace "structurally diverse molecules" with "structurally diverse functional groups" (page 14).
177) Please change "thereby inhibits" to "thereby inhibiting" (page 14).
178) Please replace "in the drug-induced inhibition" with "in drug-induced inhibition" (page 14).
179) Please replace "are key phenotypes" with "are key processes" (page 14).
180) Please change "hyperphosphorylation of FAK expression" to "hyperphosphorylation of FAK" (page 14).
181) Please replace "including gastric" with "including that of gastric" (page 14).
182) Please change "FAK activity significantly reduces" to "FAK activity significantly reduced" (page 14).
183) Please replace "FAK and phosphorylation of FAK" with "FAK whose phosphorylation" (page 14).
184) Please change "the pathway of FAK/PI3K/Akt/MMP-9 axis" to "the FAK/PI3K/Akt/MMP-9 signaling axis" (page 14).
185) Please replace "in cells, and the level of intracellular" with "in cells and the level of intracellular" (page 14).
186) Please change "there is no difference" to something like "there was no difference" or "there was no appreciable difference" (page 15).
187) Please replace "ferroptosis related molecules" with "ferroptosis-related molecules" (page 15).
188) Please replace "RNA and lipid molecules" with "RNA, and lipid molecules" (page 15).
189) Please change "of ferroptosis and also a prerequisite" to "and a prerequisite" (page 15).
190) "Accordingly, some scholars refer to lipid peroxides that can specifically cause ferroptosis, such as arachidonoyl (AA)-phosphatidylethanolamine (PE) and adrenoyl (AdA)-PE, as ferroptosis-specific lipid peroxidation" (page 15) does not seem to make sense as "scholars refer to lipid peroxides" as "lipid peroxidation"?
191) It is not exactly clear what the authors refer to by "this" in "But this is not rigid, because hydrogen peroxide in the presence of iron ions can be converted into hydroxyl radicals through the Fenton reaction, which in turn oxidizes lipids to form lipid peroxides" (page 15)?
192) It is not exactly clear what the authors mean by "lipidic ROS" in "While levels of iron, iron-containing proteins [46], and lipid peroxides [47] also promote necroptosis, GPX4 can prevent RIP3-dependent necroptosis in erythroid precursor cells by avoiding lipidic ROS accumulation" (page 15)?
193) It is not exactly clear what the authors mean by "cellular morphology of ferroptosis" in "Another major difference between ferroptosis and necroptosis is that the cellular morphology of ferroptosis is very unique" (page 15)? Do they refer to cellular morphology changes associated with ferroptosis?
194) Please replace "rupture, and perforation of" with "rupture, and perforation seen in" or "rupture, and perforation observed in" (page 15).
195) "lot of evidence" could be changed to "significant body of evidence" (page 15).
196) Please change "in many drugs-induced" to "in many types of drug-induced" (page 15).
197) Please replace "expression decreasing" with "decreased expression" (page 15).
198) Please change "necroptosis, but also increased ROS" to "necroptosis but also increased ROS" (page 15).
199) "Because the occurrence of necroptosis not only bypasses apoptosis [55,56], but also causes death in apoptosis-resistant cancer cells" (page 15) is not grammatically correct. Do the authors actually mean to say "The occurrence of necroptosis not only bypasses apoptosis [55,56], but also causes death in apoptosis-resistant cancer cells"?
200) Please replace "apoptosis [55,56], but also" with "apoptosis [55,56] but also" (page 15).
201) It is not clear why the authors mention "Second" in "Second, this mechanism of necroptosis induced by ROS can occur in most common cancers" (page 15) when there seems to be no "First" in the preceding text?
202) Please change "the phenomenon we observed such as" to "the observed" (page 15).
203) Please replace "apoptosis-related caspase protein" with "apoptosis-related caspase proteins" (page 15).
204) Please change "scabertopine" to "scabertopin" (page 16).
205) Please replace "activate RIP1/RIP3/MLKL phosphorylation, mediate J82 cell necroptosis, and inhibit" with "activating RIP1/RIP3/MLKL phosphorylation, mediating J82 cell necroptosis, and inhibiting" (page 16).
206) Please change "inhibit the growth of breast cancer" to "suppressing the growth of breast cancer" (page 16).
207) Please replace "Xanthium strumarium L" with "Xanthium strumarium L." (page 16).
208) Please replace "molecular weight (MW) of SA" with "molecular weight (MW) of scabertopin" (page 16).
209) Please change "These characteristics of SA" to "These characteristics of scabertopin" (page 16).
210) Please replace "which requires" with "which require that" (page 16).
211) Please change "anti-ROS effect" to "an antioxidant effect" (page 16).
212) Please change "through multiple mechanisms, maintain mitochondrial function, and reduce ROS production" to "maintain mitochondrial function, and reduce ROS production by multiple mechanisms" (page 17).
213) Please replace "scabertopin in vivo, and" with "scabertopin in vivo, and" (page 17).
214) Please remove bold formatting from "ΔΨ" in "accumulation by reducing ΔΨ" (page 17).
215) Please change "invasion by targeting" to "invasion by targeting the" (page 17).
216) Please replace "Using with ROS scavenger" with "Using the ROS scavenger" (page 17).
217) Please change "two abovementioned" to "FAK/PI3K/Akt/MMP-9 and RIP1/RIP3/MLKL" (page 17).
218) Please replace "grant No." with "grant number" (page 17).